# LONG-TAILED LEARNING REQUIRES FEATURE LEARNING

**Thomas Laurent[1], James H. von Brecht, Xavier Bresson[2]**
[1] Loyola Marymount University, `tlaurent@lmu.edu`
[2] National University of Singapore, `xaviercs@nus.edu.sg`

## ABSTRACT

We propose a simple data model inspired from natural data such as text or images, and use it to study the importance of learning features in order to achieve good generalization. Our data model follows a long-tailed distribution in the sense that some rare subcategories have few representatives in the training set. In this context we provide evidence that a learner succeeds if and only if it identifies the correct features, and moreover derive non-asymptotic generalization error bounds that precisely quantify the penalty that one must pay for not learning features.

## 1 INTRODUCTION

Part of the motivation for deploying a neural network arises from the belief that algorithms that learn features/representations generalize better than algorithms that do not. We try to give some mathematical ballast to this notion by studying a data model where, at an intuitive level, a learner succeeds if and only if it manages to learn the correct features. The data model itself attempts to capture two key structures observed in natural data such as text or images. First, it is endowed with a latent structure at the patch or word level that is directly tied to a classification task. Second, the data distribution has a long-tail, in the sense that rare and uncommon instances collectively form a significant fraction of the data. We derive non-asymptotic generalization error bounds that quantify, within our framework, the penalty that one must pay for not learning features.

We first prove a two part result that quantifies precisely the necessity of learning features within the context of our data model. The first part shows that a trivial nearest neighbor classifier performs perfectly when given knowledge of the correct features. The second part shows it is impossible to *a priori* craft a feature map that generalizes well when using a nearest neighbor classification rule. In other words, success or failure depends only on the ability to identify the correct features and not on the underlying classification rule. Since this cannot be done *a priori*, the features must be learned.

Our theoretical results therefore support the idea that algorithms cannot generalize on long-tailed data if they do not learn features. Nevertheless, an algorithm that does learn features can generalize well. Specifically, the most direct neural network architecture for our data model generalizes almost perfectly when using either a linear classifier or a nearest neighbor classifier on the top of the *learned* features. Crucially, designing the architecture requires knowing only the meta structure of the problem, but no *a priori* knowledge of the correct features. This illustrates the built-in advantage of neural networks; their ability to learn features significantly eases the design burden placed on the practitioner.

Subcategories in commonly used visual recognition datasets tend to follow a long-tailed distribution (Salakhutdinov et al., 2011; Zhu et al., 2014; Feldman & Zhang, 2020). Some common subcategories have a wealth of representatives in the training set, whereas many rare subcategories only have a few representatives. At an intuitive level, learning features seems especially important on a long-tailed dataset since features learned from the common subcategories help to properly classify test points from a rare subcategory. Our theoretical results help support this intuition.

We note that when considering complex visual recognition tasks, datasets are almost unavoidably long-tailed (Liu et al., 2019) — even if the dataset contains millions of images, it is to be expected that many subcategories will have few samples. In this setting, the classical approach of deriving asymptotic performance guarantees based on a large-sample limit is not a fruitful avenue. General-

ization must be approached from a different point of view (c.f. Feldman (2020) for very interesting work in this direction). In particular, the analysis must be non-asymptotic. One of our main contribution is to derive, within the context of our data model, generalization error bounds that are non-asymptotic and relatively tight — by this we mean that our results hold for small numbers of data samples and track reasonably well with empirically evaluated generalization error.

In Section 2 we introduce our data model and in Section 3 we discuss our theoretical results. For the simplicity of exposition, both sections focus on the case where each rare subcategory has a *single* representative in the training set. Section 4 is concerned with the general case in which each rare subcategory has *few* representatives. Section 5 provides an overview of our proof techniques. Finally, in Section 6, we investigate empirically a few questions that we couldn't resolve analytically. In particular, our error bounds are restricted to the case in which a nearest neighbor classification rule is applied on the top of the features — we provide empirical evidence in this last section that replacing the nearest neighbor classifier by a linear classifier leads to very minimal improvement. This further support the notion that, on our data model, it is the ability to learn features that drives success, not the specific classification rule used on the top of the features.

**Related work.** By now, a rich literature has developed that studies the generalization abilities of neural networks. A major theme in this line of work is the use of the PAC learning framework to derive generalization bounds for neural networks (e.g. Bartlett et al. (2017); Neyshabur et al. (2017); Golowich et al. (2018); Arora et al. (2018); Neyshabur et al. (2018)), usually by proving a bound on the difference between the finite-sample empirical loss and true loss. While powerful in their generality, such approaches are usually task independent and asymptotic; that is, they are mostly agnostic to any idiosyncrasies in the data generating process and need a statistically meaningful number of samples in the training set. As such, the PAC learning framework is not well-tailored to our specific aim of studying generalization on long-tailed data distributions; indeed, in such setting, a rare subcategory might have only a handful of representatives in the training set.

After breakthrough results (e.g. Jacot et al. (2018); Du et al. (2018); Allen-Zhu et al. (2019); Ji & Telgarsky (2019)) showed that vastly over-parametrized neural networks become kernel methods (the so-called Neural Tangent Kernel or NTK) in an appropriate limit, much effort has gone toward analyzing the extent to which neural networks outperform kernel methods (Yehudai & Shamir, 2019; Wei et al., 2019; Refinetti et al., 2021; Ghorbani et al., 2019; 2020; Karp et al., 2021; Allen-Zhu & Li, 2019; 2020; Li et al., 2020; Malach et al., 2021). Our interest lies not in proving such a gap for its own sake, but rather in using the comparison to gain some understanding on the importance of learning features in computer vision and NLP contexts.

Analyses that shed theoretical light onto learning with long-tailed distributions (Feldman, 2020; Brown et al., 2021) or onto specific learning mechanisms (Karp et al., 2021) are perhaps closest to our own. The former analyses (Feldman, 2020; Brown et al., 2021) investigate the necessity of memorizing rare training examples in order to obtain near-optimal generalization error when the data distribution is long-tailed. Our analysis differs to the extent that we focus on the necessity of learning features and sharing representations in order to properly classify rare instances. Like us, the latter analysis (Karp et al., 2021) also considers a computer vision inspired task and uses it to compare a neural network to a kernel method, with the ultimate aim of studying the learning mechanism involved. Their object of study (finding a sparse signal in the presence of noise), however, markedly differs from our own (learning with long-tailed distributions).

## 2 THE DATA MODEL

We begin with a simple example to explain our data model and to illustrate, at an intuitive level, the importance of learning features when faced with a long-tailed data distribution. For the sake of exposition we adopt NLP terminology such as 'words' and 'sentences,' but the image-based terminology of 'patches' and 'images' would do as well.

The starting point is a very standard mechanism for generating observed data from some underlying collection of latent variables. Consider the data model depicted in Figure 1. We have a vocabulary of $n_w = 12$ words and a set of $n_c = 3$ concepts:

$$\mathcal{V} = \{\text{potato, cheese, carrots, chicken}, \dots\} \qquad \text{and} \qquad \mathcal{C} = \{\text{vegetable, dairy, meat}\}.$$

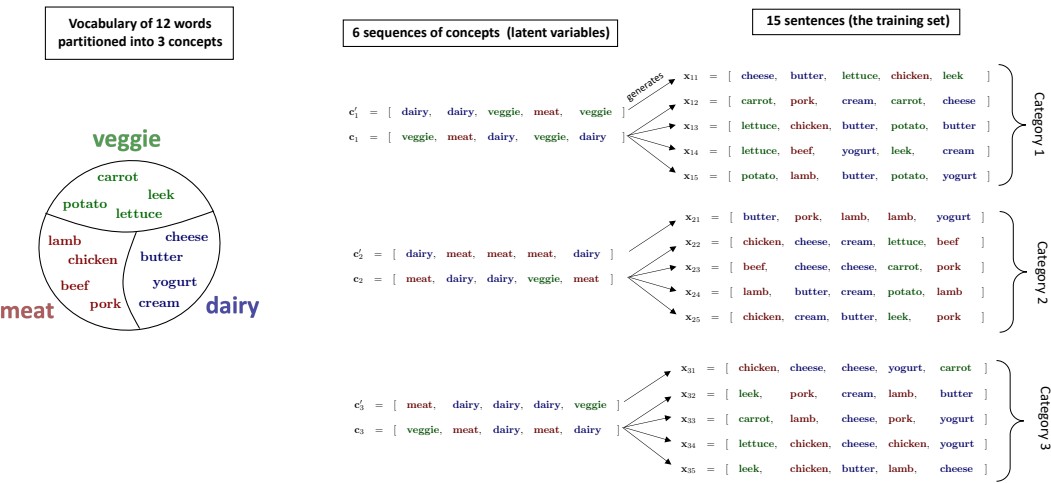

Figure 1: Data model with parameters set to $L = 5$, $n_w = 12$, $n_c = 3$, $R = 3$, and $n_{\text{spl}} = 5$.

The 12 words are partitioned into the 3 concepts as shown on the left of Figure 1. We also have 6 sequences of concepts of length $L = 5$. They are denoted by $\mathbf{c}_1, \mathbf{c}_2, \mathbf{c}_3$ and $\mathbf{c}'_1, \mathbf{c}'_2, \mathbf{c}'_3$. Sequences of concepts are latent variables that generate sequences of words. For example

$$[\text{dairy}, \text{dairy}, \text{veggie}, \text{meat}, \text{veggie}] \xrightarrow{\text{generates}} [\text{cheese}, \text{butter}, \text{lettuce}, \text{chicken}, \text{leek}]$$

The sequence of words on the right was obtained by sampling each word uniformly at random from the corresponding concept. For example, the first word was randomly chosen out of the dairy concept (*butter, cheese, cream, yogurt*), and the last word was randomly chosen out of the vegetable concept (*potato, carrot, leek, lettuce*.) Sequences of words will be referred to as sentences.

The non-standard aspect of our model comes from how we use the 'latent-variable → observed-datum' process to form a training distribution. The training set in Figure 1 is made of 15 sentences split into $R = 3$ categories. The latent variables $\mathbf{c}'_1, \mathbf{c}'_2, \mathbf{c}'_3$ each generate a single sentence, whereas the latent variables $\mathbf{c}_1, \mathbf{c}_2, \mathbf{c}_3$ each generate 4 sentences. We will refer to $\mathbf{c}_1, \mathbf{c}_2, \mathbf{c}_3$ as the **familiar** sequences of concepts since they generate most of the sentences encountered in the training set. On the other hand $\mathbf{c}'_1, \mathbf{c}'_2, \mathbf{c}'_3$ will be called **unfamiliar**. Similarly, a sentence generated by a familiar (resp. unfamiliar) sequence of concepts will be called a familiar (resp. unfamiliar) sentence. The former represents a datum sampled from the head of a distribution while the latter represents a datum sampled from its tail. We denote by $\mathbf{x}_{r,s}$ the $s^{th}$ sentence of the $r^{th}$ category, indexed so that the first sentence of each category is an unfamiliar sentence and the remaining ones are familiar.

Suppose now that we have trained a learning algorithm on the training set described above and that at inference time we are presented with a previously unseen sentence generated by the **unfamiliar** sequence of concept $\mathbf{c}'_1 = [\text{dairy}, \text{dairy}, \text{veggie}, \text{meat}, \text{veggie}]$. To fix ideas, let's say that sentence is:

$$\mathbf{x}^{\text{test}} = [\text{butter}, \text{yogurt}, \text{carrot}, \text{beef}, \text{lettuce}] \qquad (1)$$

This sentence is hard to classify since there is a single sentence in the training set that has been generated by the same sequence of concepts, namely

$$\mathbf{x}_{1,1} = [\text{cheese}, \text{butter}, \text{lettuce}, \text{chicken}, \text{leek}] . \qquad (2)$$

Moreover these two sentences do not overlap at all (i.e. the $i^{th}$ word of $\mathbf{x}^{\text{test}}$ is different from the $i^{th}$ word of $\mathbf{x}_{1,1}$ for all $i$.) To properly classify $\mathbf{x}^{\text{test}}$, the algorithm *must* have learned the equivalences *butter* ↔ *cheese*, *yogurt* ↔ *butter*, *carrot* ↔ *lettuce*, and so forth. In other words, the algorithm must have learned the underlying concepts.

Nevertheless, a neural network with a well-chosen architecture can easily succeed at such a classification task. Consider, for example, the network depicted on Figure 2. Each word of the input sentence, after being encoded into a one-hot-vector, goes through a multi-layer perceptron (MLP 1 on the figure) shared across words. The output is then normalized using LayerNorm (Ba et al., 2016) to produce a representation of the word. The word representations are then concatenated into a single vector that goes through a second multi-layer perceptron (MLP 2 on the figure).

This network, if properly trained, will learn to give similar representations to words that belong to the same concept. Therefore, if it correctly classifies the train point $\mathbf{x}_{1,1}$ given by (2), it will necessarily correctly classify the test point $\mathbf{x}^{\text{test}}$ given by (1). So the neural network is able to classify the previously unseen sentence $\mathbf{x}^{\text{test}}$ despite the fact that the training set contains a single example with the same underlying sequence of concepts. This comes from the fact that the neural network learns features and representations from the familiar part of the training set (generated by the head of the distribution), and uses these, at test time, to correctly classify the unfamiliar sentences (generated by the tail of the distribution). In other words, because it learns features, the neural network has no difficulty handling the long-tailed nature of the distribution.

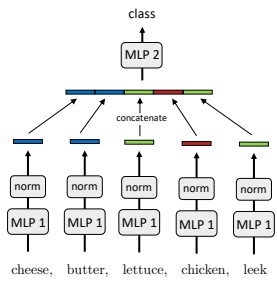

Figure 2: A simple neural net.

To summarize, the variables $L, n_w, n_c, R$ and $n_{\text{spl}}$ parametrize instances of our data model. They denote, respectively, the length of the sentences, the number of words in the vocabulary, the number of concepts, the number of categories, and the number of samples per category. So in the example presented in Figure 1 we have $L = 5, n_w = 12, n_c = 3, R = 3$ and $n_{\text{spl}} = 5$ (four familiar sentences and one unfamiliar sentence per category). The vocabulary $\mathcal{V}$ and set of concepts $\mathcal{C}$ are discrete sets with $|\mathcal{V}| = n_w$ and $|\mathcal{C}| = n_c$, rendered as $\mathcal{V} = \{1, \ldots, n_w\}$ and $\mathcal{C} = \{1, \ldots, n_c\}$ for concreteness. A partition of the vocabulary into concepts, like the one depicted at the top of Figure 1, is encoded by a function $\varphi : \mathcal{V} \to \mathcal{C}$ that assigns words to concepts. We require that each concept contains the same number of words, so that $\varphi$ satisfies

$$|\varphi^{-1}(\{c\})| = |\{w \in \mathcal{V} : \varphi(w) = c\}| = n_w/n_c \qquad \text{for all } c \in \mathcal{C}, \tag{3}$$

and we refer to such a function $\varphi : \mathcal{V} \to \mathcal{C}$ satisfying (3) as equipartition of the vocabulary. The set

$$\Phi = \{\text{All functions } \varphi \text{ from } \mathcal{V} \text{ to } \mathcal{C} \text{ that satisfy (3) }\}$$

denotes the collection of all such equipartitions, while the data space and latent space are denoted

$$\mathcal{X} = \mathcal{V}^L \qquad \text{and} \qquad \mathcal{Z} = \mathcal{C}^L,$$

respectively. Elements of $\mathcal{X}$ are sentences of $L$ words and they take the form $\mathbf{x} = [x_1, x_2, \ldots, x_L]$, while elements of $\mathcal{Z}$ take the form $\mathbf{c} = [c_1, c_2, \ldots, c_L]$ and correspond to sequences of concepts.

In the context of this work, a feature map refers to any function $\psi : \mathcal{X} \mapsto \mathcal{F}$ from data space to feature space. The feature space $\mathcal{F}$ can be any Hilbert space (possibly infinite dimensional) and we denote by $\langle \cdot, \cdot \rangle_{\mathcal{F}}$ the associated inner product. Our analysis applies to the case in which a nearest neighbor classification rule is applied on the top of the extracted features. Such rule works as follow: given a test point $\mathbf{x}$, the inner products $\langle \psi(\mathbf{x}), \psi(\mathbf{y}) \rangle_{\mathcal{F}}$ are evaluated for all $\mathbf{y}$ in the training set; the test point $\mathbf{x}$ is then given the label of the training point $\mathbf{y}$ that led to the highest inner product.

## 3 STATEMENT AND DISCUSSION OF MAIN RESULTS

Our main result states that, in the context of our data model, features must be tailored (i.e. learned) to each specific task. Specifically, it is not possible to find a universal feature map $\psi : \mathcal{X} \to \mathcal{F}$ that performs well on a collection of *tasks* like the one depicted on Figure 1. In the context of this work, a *task* refers to a tuple

$$\mathcal{T} = (\ \varphi \ ; \ \mathbf{c}_1, \ldots, \mathbf{c}_R \ ; \ \mathbf{c}'_1, \ldots, \mathbf{c}'_R \ ) \ \in \ \Phi \times \mathcal{Z}^{2R} \tag{4}$$

that prescribes a partition of the vocabulary into concepts, $R$ familiar sequences of concepts, and $R$ unfamiliar sequences of concepts. Given such a task $\mathcal{T}$ we generate a training set $S$ as described in the previous section. This training set contains $R \times n_{spl}$ sentences split over $R$ categories, and each category contains a single unfamiliar sentence. Randomly generating the training set $S$ from the task $\mathcal{T}$ corresponds to sampling

$$S \sim \mathcal{D}_{\mathcal{T}}^{\text{train}}$$

from a distribution $\mathcal{D}_{\mathcal{T}}^{\text{train}}$ defined on the space $\mathcal{X}^{R \times n_{spl}}$ and parametrized by the variables in (4) (the appendix provides an explicit formula for this distribution). We measure performance of an

algorithm by its ability to generalize on previously unseen unfamiliar sentences. Generating an unfamiliar sentence amounts to drawing a sample

$$\mathbf{x} \sim \mathcal{D}_{\mathcal{T}}^{\text{test}}$$

from a distribution $\mathcal{D}_{\mathcal{T}}^{\text{test}}$ on the space $\mathcal{X}$ parametrized by the variables $\varphi, \mathbf{c}_1', \ldots, \mathbf{c}_R'$ in (4) that determine unfamiliar sequences of concepts. Finally, associated with every task $\mathcal{T}$ we have a labelling function $f_{\mathcal{T}} : \mathcal{X} \to \{1, \ldots, R\}$ that assigns the label $r$ to sentences generated by either $\mathbf{c}_r$ or $\mathbf{c}_r'$ (this function is ill-defined if two sequences of concepts from different categories are identical, but this issue is easily resolved by formal statements in the appendix). Summarizing our notations, for every task $\mathcal{T} \in \Phi \times \mathcal{Z}^{2R}$ we have a distribution $\mathcal{D}_{\mathcal{T}}^{\text{train}}$ on the space $\mathcal{X}^{R \times n_{spl}}$, a distribution $\mathcal{D}_{\mathcal{T}}^{\text{test}}$ on the space $\mathcal{X}$, and a labelling function $f_{\mathcal{T}}$.

Given a feature space $\mathcal{F}$, a feature map $\psi : \mathcal{X} \to \mathcal{F}$, and a task $\mathcal{T} \in \Phi \times \mathcal{Z}^{2R}$, the expected generalization error of the nearest neighbor classification rule on unfamiliar sentences is given by:

$$\text{err}(\mathcal{F}, \psi, \mathcal{T}) = \mathop{\mathbb{E}}_{S \sim \mathcal{D}_{\mathcal{T}}^{\text{train}}} \left[ \mathop{\mathbb{P}}_{\mathbf{x} \sim \mathcal{D}_{\mathcal{T}}^{\text{test}}} \left[ f_{\mathcal{T}} \left( \arg\max_{\mathbf{y} \in S} \langle \psi(\mathbf{x}), \psi(\mathbf{y}) \rangle_{\mathcal{F}} \right) \neq f_{\mathcal{T}}(\mathbf{x}) \right] \right]. \quad (5)$$

For simplicity, if the test point has multiple nearest neighbors with inconsistent labels in the training set (and so the $\arg\max$ returns multiple training points $\mathbf{y}$), we will count the classification as a failure for the nearest neighbor classification rule. We therefore replace (5) by the more formal (but more cumbersome) formula

$$\text{err}(\mathcal{F}, \psi, \mathcal{T}) = \mathop{\mathbb{E}}_{S \sim \mathcal{D}_{\mathcal{T}}^{\text{train}}} \left[ \mathop{\mathbb{P}}_{\mathbf{x} \sim \mathcal{D}_{\mathcal{T}}^{\text{test}}} \left[ \exists \mathbf{y} \in \arg\max_{\mathbf{y} \in S} \langle \psi(\mathbf{x}), \psi(\mathbf{y}) \rangle_{\mathcal{F}} \text{ such that } f_{\mathcal{T}}(\mathbf{y}) \neq f_{\mathcal{T}}(\mathbf{x}) \right] \right] \quad (6)$$

to make this explicit. Our main theoretical results concern performance of a learner not on a single task $\mathcal{T}$ but on a collection of tasks $\mathfrak{T} = \{\mathcal{T}_1, \mathcal{T}_2, \ldots, \mathcal{T}_{N_{\text{tasks}}}\}$, and so we define

$$\overline{\text{err}}(\mathcal{F}, \psi, \mathfrak{T}) = \frac{1}{|\mathfrak{T}|} \sum_{\mathcal{T} \in \mathfrak{T}} \text{err}(\mathcal{F}, \psi, \mathcal{T}) \quad (7)$$

as the expected generalization error on such a collection $\mathfrak{T}$ of tasks. As a task refers to an element of the discrete set $\Phi \times \mathcal{Z}^{2R}$, any subset $\mathfrak{T} \subset \Phi \times \mathcal{Z}^{2R}$ defines a collection of tasks. Our main result concerns the case where the collection of tasks $\mathfrak{T} = \Phi \times \mathcal{Z}^{2R}$ consists in *all possible tasks* that one might encounter. For concreteness, we choose specific values for the model parameters and state the following special case of our main theorem (Theorem 3 at the end of this section) —

**Theorem 1.** *Let $L = 9$, $n_w = 150$, $n_c = 5$, $R = 1000$ and $n_{spl} \geq 2$. Let $\mathfrak{T} = \Phi \times \mathcal{Z}^{2R}$. Then*

$$\overline{\text{err}}(\mathcal{F}, \psi, \mathfrak{T}) > 98.4\%$$

*for all feature spaces $\mathcal{F}$, and all feature maps $\psi : \mathcal{X} \mapsto \mathcal{F}$.*

In other words, for the model parameters specified above, it is not possible to design a 'task-agnostic' feature map $\psi$ that works well if we are uniformly uncertain about which specific task we will face. Indeed, the best possible feature map will fail at least $98.4\%$ of the time at classifying unfamiliar sentences (with a nearest-neighbor classification rule), where the probability is with respect to the random choices of the task, of the training set, and of the unfamiliar test sentence.

**Interpretation:** Our desire to understand *learning* demands that we consider a collection of tasks rather than a single one, for if we consider only a single task then the problem, in our setting, becomes trivial. Indeed, assume $\mathfrak{T} = \{\mathcal{T}_1\}$ with $\mathcal{T}_1 = (\varphi; \mathbf{c}_1, \ldots, \mathbf{c}_R; \mathbf{c}_1', \ldots, \mathbf{c}_R')$ consists only of a single task. With knowledge of this task we can easily construct a feature map $\psi : \mathcal{X} \to \mathbb{R}^{L n_c}$ that performs perfectly. Indeed, the map

$$\psi([x_1, \ldots, x_L]) = [\mathbf{e}_{\varphi(x_1)}, \ldots, \mathbf{e}_{\varphi(x_L)}] \quad (8)$$

that simply 'replaces' each word $x_\ell$ of the input sentence by the one-hot-encoding $\mathbf{e}_{\varphi(x_\ell)}$ of its corresponding concept will do.[1] A bit of thinking reveals that the nearest neighbor classification rule associated with feature map (8) perfectly solves the task $\mathcal{T}_1$. This is due to the fact that sentences

---

[1] We use $\mathbf{e_i}$ to denote the $i^{th}$ basis vector of $\mathbb{R}^{n_c}$. So $\mathbf{e}_{\varphi(x_\ell)}$ is a one-hot vector coding for the concept $\varphi(x_\ell)$.

generated by the same sequence of concepts are mapped by $\psi$ to the exact same location in feature space. As a consequence, the nearest neighbor classification rule will match the unfamiliar test sentence $\mathbf{x}$ to the unique training sentence $\mathbf{y}$ that occupies the same location in feature space, and this training sentence has the correct label by construction (assuming that sequences of concepts from different categories are distinct). To put it formally:

**Theorem 2.** *Given a task $\mathcal{T} \in \Phi \times \mathcal{Z}^{2R}$ satisfying $\mathbf{c}'_r \neq \mathbf{c}'_s$ and $\mathbf{c}'_r \neq \mathbf{c}_s$ for all $r \neq s$, there exists a feature space $\mathcal{F}$ and a feature map $\psi : \mathcal{X} \mapsto \mathcal{F}$ such that $\mathrm{err}(\mathcal{F}, \psi, \mathcal{T}) = 0$.*

Consider now the case where $\mathfrak{T} = \{\mathcal{T}_1, \mathcal{T}_2\}$ consists of two tasks. According to Theorem 2 there exists a $\psi$ that perfectly solves $\mathcal{T}_1$, but this $\psi$ might perform poorly on $\mathcal{T}_2$, and vice versa. So, it might not be possible to design good features if we do not know *a priori* which of these tasks we will face. Theorem 1 states that, in the extreme case where $\mathfrak{T}$ contains all possible tasks, this is indeed the case — the best possible 'task-agnostic' features $\psi$ will perform catastrophically on average. In other words, features must be task-dependent in order to succeed.

To draw a very approximate analogy, imagine once again that $\mathfrak{T} = \{\mathcal{T}_1\}$ and that $\mathcal{T}_1$ represents, say, a hand-written digit classification task. A practitioner, after years of experience, could hand-craft a very good feature map $\psi$ that performs almost perfectly for this task. If we then imagine the case $\mathfrak{T} = \{\mathcal{T}_1, \mathcal{T}_2\}$ where $\mathcal{T}_1$ represents a hand-written digit classification task and $\mathcal{T}_2$ represents, say, an animal classification task, then it becomes more difficult for a practitioner to handcraft a feature map $\psi$ that works well for *both* tasks. In this analogy, the size of the set $\mathfrak{T}$ encodes the amount of knowledge the practitioner has about the specific tasks she will face. The extreme choice $\mathfrak{T} = \Phi \times \mathcal{Z}^{2R}$ corresponds to the practitioner knowing nothing beyond the fact that *natural images are made of patches*. Theorem 1 quantifies, in this extreme case, the impossibility of hand-crafting a feature map $\psi$ knowing only the range of possible tasks and not the specific task itself. In a realistic setting the collection of tasks $\mathfrak{T}$ is smaller, of course, and the data generative process itself is more coherent than in our simplified setup. Nonetheless, we hope our analysis sheds some light on some of the essential limitations of algorithms that do not learn features.

Finally, our empirical results (see Section 6) show that a simple algorithm that learns features does not face this obstacle. We do *not* need knowledge of the specific task $\mathcal{T}$ in order to design a good neural network architecture, but only of the family of tasks $\mathfrak{T} = \Phi \times \mathcal{Z}^{2R}$ that we will face. Indeed, the architecture in Figure 2 succeeds at classifying unfamiliar test sentences more than 99% of the time. This probability, which we empirically evaluate, is with respect to the choice of the task, the choice of the training set, and the choice of the unfamiliar test sentence (we use the values of $L, n_w, n_c$ and $R$ from Theorem 1, and $n_{\mathrm{spl}} = 6$, for this experiment). Continuing with our approximate analogy, this means our hypothetical practitioner needs no domain specific knowledge beyond the patch structure of natural images when designing a successful architecture. In sum, successful feature design requires task-specific knowledge while successful architecture design requires only knowledge of the task family.

**Main Theorem:** Our main theoretical result extends Theorem 1 to arbitrary values of $L$, $n_w$, $n_c$, $n_{\mathrm{spl}}$ and $R$. The resulting formula involves various combinatorial quantities. We denote by $\binom{n}{k}$ the binomial coefficients and by $\left\{ {n \atop k} \right\}$ the Stirling numbers of the second kind. Let $\mathbb{N} = \{0, 1, 2, \ldots\}$ and let $\gamma, \hat{\gamma} : \mathbb{N}^{L+1} \to \mathbb{N}$ be the functions defined by $\gamma(\mathbf{k}) := \sum_{i=1}^{L+1} (i-1)k_i$ and $\hat{\gamma}(\mathbf{k}) := \sum_{i=1}^{L+1} ik_i$, respectively. We then define, for $0 \leq \ell \leq L$, the sets

$$\mathcal{S}_\ell := \left\{ \mathbf{k} \in \mathbb{N}^{L+1} : \quad \hat{\gamma}(\mathbf{k}) = n_w \quad \text{and} \quad \ell \leq \gamma(\mathbf{k}) \leq L \right\}.$$

We let $\mathcal{S} = \mathcal{S}_0$, and we note that the inclusion $\mathcal{S}_\ell \subset \mathcal{S}$ always holds. Given $\mathbf{k} \in \mathbb{N}^{L+1}$ we denote by

$$\mathcal{A}_{\mathbf{k}} := \left\{ A \in \mathbb{N}^{(L+1) \times n_c} : \quad \sum_{i=1}^{L+1} i A_{ij} = n_w/n_c \text{ for all } j \quad \text{and} \quad \sum_{j=1}^{n_c} A_{ij} = k_i \quad \text{for all } i \right\}$$

the set of $\mathbf{k}$-admissible matrices. Finally, we let $\mathfrak{f}, \mathfrak{g} : \mathcal{S} \to \mathbb{R}$ be the functions defined by

$$\mathfrak{f}(\mathbf{k}) := \left( (n_w/n_c)! \right)^{n_c} \frac{n_c^L}{n_w!} \sum_{A \in \mathcal{A}_k} \left( \prod_{i=1}^{L+1} \frac{k_i!}{A_{i,1}!\, A_{i,2}! \, \cdots \, A_{i,n_c}!} \right) \qquad \text{and}$$

$$\mathfrak{g}(\mathbf{k}) := \frac{\gamma(\mathbf{k})!}{n_w^{2L}} \left( \frac{n_w!}{k_1! k_2! \cdots k_{L+1}!} \prod_{i=2}^{L+1} \left( \frac{i^{(i-2)}}{i!} \right)^{k_i} \right) \left( \sum_{i=\gamma(\mathbf{k})}^{L} \binom{L}{i} \left\{ {i \atop \gamma(\mathbf{k})} \right\} 2^i n_w^{L-i} \right),$$

respectively. With these definitions at hand, we may now state our main theorem.

**Theorem 3** (Main Theorem). *Let* $\mathfrak{T} = \Phi \times \mathcal{Z}^{2R}$. *Then*

$$\overline{err}(\mathcal{F}, \psi, \mathfrak{T}) \geq \left( \sum_{\mathbf{k} \in \mathcal{S}_\ell} \mathfrak{f}(\mathbf{k})\mathfrak{g}(\mathbf{k}) \right) - \frac{1}{R} \left( 1 + \frac{1}{2} \max_{\mathbf{k} \in \mathcal{S}_\ell} \mathfrak{f}(\mathbf{k}) \right) \tag{9}$$

*for all feature spaces* $\mathcal{F}$, *all feature maps* $\psi : \mathcal{X} \mapsto \mathcal{F}$, *and all* $0 \leq \ell \leq L$.

The combinatorial quantities involved appear a bit daunting at a first glance, but, within the context of the proof, they all take a quite intuitive meaning. The heart of the proof involves the analysis of a measure of concentration that we call the permuted moment, and of an associated graph-cut problem. The combinatorial quantities arise quite naturally in the course of analyzing the graph cut problem. We provide a quick overview of the proof in Section 5, and refer to the appendix for full details. For now, it suffices to note that we have a formula (i.e. the right hand side of (9)) that can be exactly evaluated with a few lines code. This formula provides a relatively tight lower bound for the generalization error. Theorem 1 is then a direct consequence — plugging $L = 9$, $n_w = 150$, $n_c = 5$, $R = 1000$ and $\ell = 7$ in the right hand side of (9) gives the claimed $98.4\%$ lower bound.

## 4 MULTIPLE UNFAMILIAR SENTENCES PER CATEGORY

The two previous sections were concerned with the case in which each unfamiliar sequence of concepts has a *single* representative in the training set. In this section we consider the more general case in which each unfamiliar sequence of concepts has $n^*$ representatives in the training set. Figure 3 depicts an example with $n_{\mathrm{spl}} = 6$ and $n^* = 2$. This means that each category contains a total of $n_{\mathrm{spl}} = 6$ sentences, and that $n^* = 2$ of these sentences are generated by the unfamiliar sequence of concepts (the remaining four are generated by the familiar sequence of concepts). The other parameters in this example are $L = 5$, $n_w = 12$, $n_c = 3$ and $R = 3$.

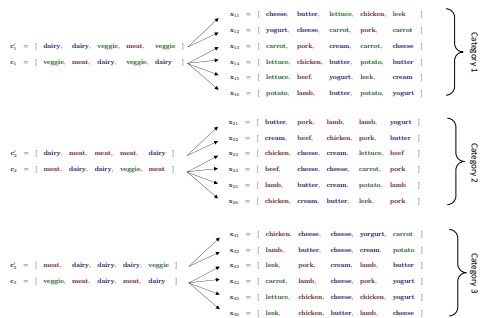

Figure 3: More general version of our data model.

Using a simple union bound, inequality (9) easily extends to this situation — the resulting formula is a bit cumbersome so we present it in the appendix (see Theorem 7). In the concrete case where $L = 9$, $n_w = 150$, $n_c = 5$, $R = 1000$ this formula simplifies to

$$\overline{err}(\mathcal{F}, \psi, \mathfrak{T}) \geq 1 - 0.015 \, n^* - 1/R \qquad \text{for all } \mathcal{F} \text{ and all } \psi, \tag{10}$$

therefore exhibiting an affine relationship between the error rate and the number $n^*$ of unfamiliar sentences per category. Note that choosing $n^* = 1$ in (10) leads to a $98.4\%$ lower bound on the error rate, therefore recovering the result from Theorem 1. This lower bound then decreases by $1.5\%$ with each additional unfamiliar sentence per category in the training set.

We would like to emphasize one more time the importance of non-asymptotic analysis in the long-tailed learning setting. For example, in inequality (10), the difficulty lies in obtaining a value as small as possible for the coefficient in front of $n^*$. We accomplish this via a careful analysis of the graph cut problem associated with our data model.

## 5 PROOF OUTLINE — PERMUTED MOMENT AND OPTIMAL FEATURE MAP

The proof involves two main ingredients. First, the key insight of our analysis is the realization that generalization in our data model is closely tied to the *permuted moment* of a probability distribution. To state this central concept, it will prove convenient to think of probability distributions on $\mathcal{X}$ as vectors $\mathbf{p} \in \mathbb{R}^N$ with $N = |\mathcal{X}|$, together with indices $0 \leq i \leq N - 1$ given by some arbitrary (but fixed) indexing of the elements of data space. Then $p_i$ denotes the probability of the $i^{\mathrm{th}}$ element of

$\mathcal{X}$ in this indexing. We use $S_N$ to denote the set of permutations of $\{0, 1, \ldots, N-1\}$ and $\sigma \in S_N$ to refer to a particular permutation. The $t^{th}$ permuted moment of the probability vector $\mathbf{p} \in \mathbb{R}^N$ is

$$\mathcal{H}_t(\mathbf{p}) \;:=\; \max_{\sigma \in S_N} \; \sum_{i=0}^{N-1} (i/N)^t \; p_{\sigma(i)} \tag{11}$$

Since (11) involves a maximum over all possible permutations, the definition clearly does not depend on the way the set $\mathcal{X}$ was indexed. In order to maximize the sum, the permutation $\sigma$ must match the largest values of $p_i$ with the largest values of $(i/N)^t$, so the maximizing permutation simply orders the entries $p_i$ from smallest to largest. A very peaked distribution that gives large probability to only a handful of elements of $\mathcal{X}$ will have large permuted moment. Because of this, the permuted moment is akin to the negative entropy; it has large values for delta-like distributions and small values for uniform ones. From definition (11) it is clear that $0 \le \mathcal{H}_t(\mathbf{p}) \le 1$ for all probability vectors $\mathbf{p}$, and it is easily verified that the permuted moment is convex. These properties, as well as various useful bounds for the permuted moment, are presented and proven in the appendix.

Second, we identify a specific feature map, $\psi^\star : \mathcal{X} \to \mathcal{F}^\star$, which is optimal for a collection of tasks closely related to the ones considered in our data model. Leveraging the optimality of $\psi^\star$ on these related tasks allows us to derive an error bound that holds for the tasks of interest. The feature map $\psi^\star$ is better understood through its associated kernel, which is given by the formula

$$K^\star(\mathbf{x}, \mathbf{y}) = \langle \psi^\star(\mathbf{x}), \psi^\star(\mathbf{y}) \rangle_{\mathcal{F}^\star} = \frac{n_c^L}{n_w^L} \; \frac{\left| \{ \varphi \in \Phi : \varphi(x_\ell) = \varphi(y_\ell) \text{ for all } 1 \le \ell \le L \} \right|}{|\Phi|}. \tag{12}$$

Up to normalization, $K^\star(\mathbf{x}, \mathbf{y})$ simply counts the number of equipartitions of the vocabulary for which sentences $\mathbf{x}$ and $\mathbf{y}$ have the same underlying sequence of concepts. Intuitively this makes sense, for the *best possible* kernel must leverage the only information we have at hand. We know the general structure of the problem (words are partitioned into concepts) but not the partition itself. So to try and determine if sentences $(\mathbf{x}, \mathbf{y})$ were generated by the same sequence of concepts, the best we can do is to simply try all possible equipartitions of the vocabulary and count how many of them wind up generating $(\mathbf{x}, \mathbf{y})$ from the same underlying sequence of concepts. A high count makes it more likely that $(\mathbf{x}, \mathbf{y})$ were generated by the same sequence of concepts. The optimal kernel $K^\star$ does exactly this, and provides a good (actually optimal, see the appendix) measure of similarity between pairs of sentences.

For fixed $\mathbf{x} \in \mathcal{X}$, the function $\mathbf{y} \mapsto K^\star(\mathbf{x}, \mathbf{y})$ defines a probability distribution on data space. The connection between generalization error, permuted moment, and optimal feature map, come from the fact that

$$\sup_{\mathcal{F}, \psi} \left[ 1 - \overline{\mathrm{err}}(\mathcal{F}, \psi, \mathfrak{T}) \right] \le \frac{1}{|\mathcal{X}|} \sum_{\mathbf{x} \in \mathcal{X}} \mathcal{H}_{2R-1}\left( K^\star(\mathbf{x}, \cdot) \right) + \frac{1}{R}, \tag{13}$$

and so, up to a small error $1/R$, it is the permuted moments of $K^\star$ that determine the success rate. We then obtain the lower bound (9) by studying these moments in great detail. A simple union bound is then used to obtain inequalities such as (10).

## 6  EMPIRICAL RESULTS

We conclude by presenting empirical results that complement our theoretical findings. The full details of these experiments (training procedure, hyperparameter choices, number of experiments ran to estimate the success rates, and standard deviations of these success rates), as well as additional experiments, can be found in Appendix E. Codes are available at `https://github.com/xbresson/Long_Tailed_Learning_Requires_Feature_Learning`.

**Parameter Settings.** We consider five parameter settings for the data model depicted in Figure 3. Each setting corresponds to a column in Table 1. In all five settings, we set the parameters $L = 9$, $n_w = 150$, $n_c = 5$ and $R = 1000$ to the values for which the error bound (10) holds. We choose values for the parameters $n_{\mathrm{spl}}$ and $n^*$ so that the $i^{th}$ column of the table corresponds to a setting in which the training set contains 5 familiar and $i$ unfamiliar sentences per category. Recall that $n_{\mathrm{spl}}$ is the total number of samples per category in the training set. So the first column of the table corresponds to a setting in which each category contains 5 familiar sentences and 1 unfamiliar

Table 1: Success rate on unfamiliar test sentences.

| | $n^* = 1$ $n_{\mathrm{spl}} = 6$ | $n^* = 2$ $n_{\mathrm{spl}} = 7$ | $n^* = 3$ $n_{\mathrm{spl}} = 8$ | $n^* = 4$ $n_{\mathrm{spl}} = 9$ | $n^* = 5$ $n_{\mathrm{spl}} = 10$ |
|---|---|---|---|---|---|
| Neural network in Figure 2 | 99.8% | 99.9% | 99.9% | 99.9% | 100% |
| Nearest neighb. on features learned by neural net | 99.9% | 99.9% | 99.9% | 99.9% | 99.9% |
| Nearest neighb. on features extracted by $\psi^\star$ | 0.7% | 1.1% | 1.5% | 1.8% | 2.2% |
| Nearest neighb. on features extracted by $\psi_{\mathrm{one-hot}}$ | 0.6% | 1.1% | 1.4% | 1.7% | 2.1% |
| Theoretical upper bound ($0.015n^* + 1/1000$) | 1.6% | 3.1% | 4.6% | 6.1% | 7.6% |
| SVM on features extracted by $\psi^\star$ | 0.6% | 1.5% | 2.2% | 3.2% | 4.2% |
| SVM on features extracted by $\psi_{\mathrm{one-hot}}$ | 0.5% | 1.1% | 1.9% | 2.8% | 3.8% |
| SVM with Gaussian kernel | 0.6% | 1.1% | 2.0% | 2.8% | 3.6% |

sentence, whereas the last column corresponds to a setting in which each category contains 5 familiar sentences and 5 unfamiliar sentences.

**Algorithms.** We evaluate empirically seven different algorithms. The first two rows of the table correspond to experiments in which the neural network in Figure 2 is trained with SGD and constant learning rate. At test time, we consider two different strategies to classify test sentences. The first row of the table considers the usual situation in which the trained neural network is used to classify test points. The second row considers the situation in which the trained neural network is only used to extract features (i.e. the concatenated words representation right before MLP2). The classification of test points is then accomplished by running a nearest neighbor classifier on these learned features. The third (resp. sixth) row of the table shows the results obtained when running a nearest neighbor algorithm (resp. SVM) on the features $\psi^\star$ of the optimal feature map. By the kernel trick, these algorithms only require the values of the optimal kernel $\langle \psi^\star(\mathbf{x}), \psi^\star(\mathbf{y}) \rangle_{\mathcal{F}^\star}$, computed via (12), and not the features $\psi^\star$ themselves. The fourth (resp. seventh) row shows results obtained when running a nearest neighbor algorithm (resp. SVM) on features extracted by the simplest possible feature map, that is

$$\psi_{\mathrm{one-hot}}([x_1, \ldots, x_L]) = [\mathbf{e}_{x_1}, \ldots, \mathbf{e}_{x_L}]$$

where $\mathbf{e}_{x_\ell}$ denotes the one-hot-encoding of the $\ell^{th}$ word of the input sentence. Finally, the last row considers a SVM with Gaussian Kernel (also called RBF kernel).

**Results.** The first two rows of the table correspond to algorithms that *learn* features from the data; the remaining rows correspond to algorithms that use a pre-determined (not learned) feature map. Table 1 reports the success rate of each algorithm on unfamiliar test sentences. A crystal-clear pattern emerges. Algorithms that learn features generalize almost perfectly, while algorithms that do not learn features catastrophically fail. Moreover, the specific classification rule matters little. For example, replacing MLP2 by a nearest neighbor classifier on the top of features learned by the neural network leads to equally accurate results. Similarly, replacing the nearest neighbor classifier by a SVM on the top of features extracted by $\psi^\star$ or $\psi_{\mathrm{one-hot}}$ leads to almost equally poor results. The only thing that matters is whether or not the features are learned. Finally, inequality (10) gives an upper bound of $0.015n^* + 1/1000$ on the success rate of the nearest neighbor classification rule applied on the top of *any possible feature map* (including $\psi^\star$ and $\psi_{\mathrm{one-hot}}$). The fifth row of Table 1 compares this bound against the empirical accuracy obtained with $\psi^\star$ and $\psi_{\mathrm{one-hot}}$, and the comparison shows that our theoretical upper bound is relatively tight.

When $n^* = 1$ our main theorem states that no feature map can succeed more than $1.6\%$ of the time on unfamiliar test sentences (fifth row of the table). At first glance this appears to contradict the empirical performance of the feature map extracted by the neural network, which succeeds $99\%$ of the time (second row of the table). The resolution of this apparent contradiction lies in the order of operations. The point here is to separate *hand crafted* or *fixed* features from *learned* features via the order of operations. If we choose the feature map *before* the random selection of the task then the algorithm performs poorly since it uses unlearned, task-independent features. By contrast, the neural network learns a feature map from the training set, and since the training set is generated by the task, this process takes place *after* the random selection of the task. It therefore uses task-dependent features, and the network performs almost perfectly for the specific task that generated its training set. But by our main theorem, it too must fail if the task changes but the features do not.

**Acknowledgements.** Xavier Bresson is supported by NUS-R-252-000-B97-133 and A*STAR Grant ID A20H4g2141.

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

# Appendix

In Section A we prove a few elementary properties of the permuted moment (11). Section B is devoted to the proof of inequality (13), which we restate here for convenience:

$$\sup_{\mathcal{F}, \psi} \left[ 1 - \overline{\mathrm{err}}(\mathcal{F}, \psi, \mathfrak{T}) \right] \leq \frac{1}{|\mathcal{X}|} \sum_{\mathbf{x} \in \mathcal{X}} \mathcal{H}_{2R-1} \left( K^{\star}(\mathbf{x}, \cdot) \right) + \frac{1}{R} \tag{14}$$

where the collection of tasks $\mathfrak{T} = \Phi \times \mathcal{Z}^{2R}$ consists in *all possible tasks* that one might encounter. Inequality (14) plays a central role in our work as it establishes the connection between the generalization error, the permuted moment, and the optimal kernel $K^{\star}$ defined by (12). The proof is non-technical and easily accessible. In Section C we provide the following upper bound on the permuted moment of the optimal kernel:

$$\frac{1}{|\mathcal{X}|} \sum_{\mathbf{x} \in \mathcal{X}} \mathcal{H}_{2R-1} \left( K^{\star}(\mathbf{x}, \cdot) \right) \leq \left( 1 - \sum_{\mathbf{k} \in \mathcal{S}_\ell} \mathfrak{f}(\mathbf{k}) \mathfrak{g}(\mathbf{k}) \right) + \frac{1}{2R} \left( \max_{\mathbf{k} \in \mathcal{S}_\ell} \mathfrak{f}(\mathbf{k}) \right) \tag{15}$$

for all $0 \leq \ell \leq L$. The proof is combinatorial in nature, and involves the analysis of a graph-cut problem. Combining (14) and (15) establishes Theorem 3. In Section D we consider the case in which each unfamiliar sequence of concepts has $n^*$ representatives in the training set. A simple union bound shows that, in this situation, inequality (14) becomes

$$\sup_{\mathcal{F}, \psi} \left[ 1 - \overline{\mathrm{err}}(\mathcal{F}, \psi, \mathfrak{T}) \right] \leq \frac{n^*}{|\mathcal{X}|} \sum_{\mathbf{x} \in \mathcal{X}} \mathcal{H}_{2R-1} \left( K^{\star}(\mathbf{x}, \cdot) \right) + \frac{1}{R} \tag{16}$$

Combining (16) and (15) then provides our most general error bound, see Theorem 7. Inequality (10) in the main body of the paper is just a special case of Theorem 7. Finally, in Section E, we provide the full details of the experiments.

## A  PROPERTIES OF THE PERMUTED MOMENT

The permuted moment, in Section 5, was defined for probability vectors only. It will prove convenient to consider the permuted moment of nonnegative vectors as well. We denote by $\mathbb{R}_+ = [0, +\infty)$ the nonnegative real numbers, and by $\mathbb{R}_+^N$ the vectors with $N$ nonnegative real entries indexed from $i = 0$ to $i = N - 1$. The permuted moment of $\mathbf{u} \in \mathbb{R}_+^N$ is then given by

$$\mathcal{H}_t(\mathbf{u}) \; := \; \max_{\sigma \in S_N} \sum_{i=0}^{N-1} (i/N)^t \, u_{\sigma(i)}. \tag{17}$$

where $S_N$ denote the set of permutations of $\{0, 1, \ldots, N-1\}$. The concept of an *ordering permutation* will prove useful in the next lemma.

**Definition 1.** $\sigma \in S_N$ *is said to be an ordering permutation of* $\mathbf{u} \in \mathbb{R}^N$ *if*

$$u_{\sigma(0)} \leq u_{\sigma(1)} \leq \ldots \leq u_{\sigma(N-1)}. \tag{18}$$

The lemma below shows that the permutation maximizing (17) is the one that sorts the entries $u_i$ from smallest to largest.

**Lemma 1.** *Let* $\mathbf{u} \in \mathbb{R}_+^N$ *and let* $\sigma^*$ *be an ordering permutation of* $\mathbf{u}$. *Then*

$$\sigma^* \; \in \; \arg\max_{\sigma \in S_N} \; \sum_{i=0}^{N-1} (i/N)^t \, u_{\sigma(i)}. \tag{19}$$

*Proof.* The optimization problem (19) can be formulated as finding a pairing between the $u_i$'s and the $(i/N)^t$'s that maximizes the sum of the product of the pairs. An ordering permutation of $\mathbf{u}$ corresponds to pairing the smallest entry of $\mathbf{u}$ to $(0/N)^t$, the second smallest entry to $(1/N)^t$, the third smallest entry to $(2/N)^t$, and so forth. This pairing is clearly optimal. $\qquad \square$

In light of the previous lemma, we see that computing the permuted moment of a vector $\mathbf{u}$ can be accomplished as follow: 1) sort the entries of $\mathbf{u}$ from smallest to largest; 2) compute the dot product between this sorted vector and the vector

$$\left[\left(\tfrac{0}{N}\right)^t \quad \left(\tfrac{1}{N}\right)^t \quad \left(\tfrac{2}{N}\right)^t \quad \cdots \quad \left(\tfrac{N-1}{N}\right)^t\right]. \tag{20}$$

Let us now focus on the case where $\mathbf{u}$ is a probability distribution. If $\mathbf{u}$ is very peaked, it must have a large permuted moment since, after sorting, most of the mass concentrates on the high values of (20) located on the right. On the contrary, if $\mathbf{u}$ is very spread, it must have small permuted moment since it 'wastes' its mass on small values of (20). Because of this, the permuted moment is akin to the negative entropy; it has large values for delta-like distributions and small values for uniform ones.

We now show that the permuted moment is subadditive and one-homogeneous on $\mathbb{R}_+^N$ (as a consequence it is convex on the set of probability vectors) and we derive some elementary $\ell_1$ and $\ell_\infty$ bounds. We denote by $\|\mathbf{u}\|_p$ the $\ell_p$-norm of a vector $\mathbf{u}$. In particular, if $\mathbf{u} \in \mathbb{R}_+^N$, we have

$$\|\mathbf{u}\|_1 := \sum_{i=0}^{N-1} u_i \qquad \text{and} \qquad \|\mathbf{u}\|_\infty := \max_{0 \le i \le N-1} u_i.$$

With this notation in hand, we can now state our lemma:

**Lemma 2.**  *(i)* $\mathcal{H}_t(\mathbf{u} + \mathbf{v}) \le \mathcal{H}_t(\mathbf{u}) + \mathcal{H}_t(\mathbf{v})$ *for all* $\mathbf{u}, \mathbf{v} \in \mathbb{R}_+^N$.

*(ii)* $\mathcal{H}_t(c\,\mathbf{u}) = c\,\mathcal{H}_t(\mathbf{u})$ *for all* $\mathbf{u} \in \mathbb{R}_+^N$ *and all* $c \ge 0$.

*(iii)* $\mathcal{H}_t(\mathbf{u}) \le \|\mathbf{u}\|_1$ *for all* $\mathbf{u} \in \mathbb{R}_+^N$.

*(iv)* $\mathcal{H}_t(\mathbf{u}) \le \frac{N}{t+1}\|\mathbf{u}\|_\infty$ *for all* $\mathbf{u} \in \mathbb{R}_+^N$.

*Proof.* Properties (i) and (ii) are obvious. To prove (iii) and (iv), define $w_i = (i/N)^t$ and note that

$$\|\mathbf{w}\|_\infty \le 1 \qquad \text{and} \qquad \|\mathbf{w}\|_1 = N\left(\frac{1}{N}\sum_{i=0}^{N-1}(i/N)^t\right) \le N\int_0^1 x^t dt = \frac{N}{t+1}$$

Then (iii) comes from $\mathcal{H}_t(\mathbf{u}) \le \|\mathbf{w}\|_\infty \|\mathbf{u}\|_1$ whereas (iv) comes from $\mathcal{H}_t(\mathbf{u}) \le \|\mathbf{w}\|_1 \|\mathbf{u}\|_\infty$. $\quad\square$

We conclude this section with a slightly more sophisticated bound that holds for probability vectors — this bound will play a central role in Section C.

**Lemma 3.** *Suppose* $\mathbf{p} \in \mathbb{R}_+^N$, *and suppose* $\sum_{i=1}^N p_i = 1$. *Then*

$$\mathcal{H}_t(\mathbf{p}) \le \left(1 - \sum_{i=0}^{N-1} \min\{p_i, \lambda\}\right) + \frac{\lambda N}{t+1} \qquad \text{for all } \lambda \ge 0.$$

*Proof.* Fix a $\lambda \ge 0$ and define the vectors $\mathbf{u}$ and $\mathbf{v}$ as follow:

$$u_i = \min\{p_i, \lambda\} \qquad \text{and} \qquad v_i = p_i - \min\{p_i, \lambda\} \qquad \text{for all } 0 \le i \le N-1$$

Note that this two vectors are non-negative and sum to $\mathbf{p}$. We can therefore use Lemma 2 to obtain

$$\mathcal{H}_t(\mathbf{p}) = \mathcal{H}_t(\mathbf{u} + \mathbf{v}) \le \mathcal{H}_t(\mathbf{u}) + \mathcal{H}_t(\mathbf{v}) \le \frac{N}{t+1}\|\mathbf{u}\|_\infty + \|\mathbf{v}\|_1$$

To conclude, we note that $\|\mathbf{u}\|_\infty \le \lambda$, and $\|\mathbf{v}\|_1 = 1 - \sum_{i=0}^{N-1}\min\{p_i, \lambda\}$. $\quad\square$

## B    PERMUTED MOMENT OF $K^\star$ AND GENERALIZATION ERROR

This section is devoted to the proof of inequality (14). We start by recalling a few definitions. The vocabulary, set of concepts, data space, and latent space are

$$\mathcal{V} = \{1, \ldots, n_w\}, \qquad \mathcal{C} = \{1, \ldots, n_c\}, \qquad \mathcal{X} = \mathcal{V}^L \qquad \text{and} \qquad \mathcal{Z} = \mathcal{C}^L$$

respectively. Elements of $\mathcal{X}$ are sentences of $L$ words and they take the form $\mathbf{x} = [x_1, x_2, \ldots, x_L]$, while elements of $\mathcal{Z}$ take the form $\mathbf{c} = [c_1, c_2, \ldots, c_L]$ and correspond to sequences of concepts. We also recall that the collection of all equipartitions of the vocabulary is denoted by

$$\Phi = \big\{ \text{All functions } \varphi \text{ from } \mathcal{V} \text{ to } \mathcal{C} \text{ that satisfy } |\varphi^{-1}(\{c\})| = s_c \text{ for all } c \big\}$$

where $s_c := n_w/n_c$ denote the size of the concepts. Given $\varphi \in \Phi$, we denote by $\mathring{\varphi} : \mathcal{X} \to \mathcal{C}$ the function

$$\mathring{\varphi}\big([x_1, x_2, \ldots, x_L]\big) := \big[\varphi(x_1), \varphi(x_2), \ldots, \varphi(x_L)\big]$$

that operates on sentences element-wise. The informal statement "the sentence $\mathbf{x}$ is randomly generated by the sequence of concepts $\mathbf{c}$" means that $\mathbf{x}$ is sampled uniformly at random from the set

$$\mathring{\varphi}^{-1}(\{\mathbf{c}\}) = \{\mathbf{x} \in \mathcal{X} : \mathring{\varphi}(\mathbf{x}) = \mathbf{c}\}. \tag{21}$$

We will often do the abuse of notation of writing $\mathring{\varphi}^{-1}(\mathbf{c})$ instead of $\mathring{\varphi}^{-1}(\{\mathbf{c}\})$. We now formally define the sampling process associated with our main data model.

**Sampling Process DM:**

---

   (i)  Sample $\mathcal{T} = ( \varphi \,;\, \mathbf{c}_1, \ldots, \mathbf{c}_R \,;\, \mathbf{c}'_1, \ldots, \mathbf{c}'_R )$ uniformly at random in $\mathfrak{T} = \Phi \times \mathcal{Z}^{2R}$.

   (ii)  For $r = 1, \ldots, R$:
   - Sample $(\mathbf{x}_{r,1}, \ldots, \mathbf{x}_{r,n_{\text{unf}}})$ uniformly at random in $\mathring{\varphi}^{-1}(\mathbf{c}'_r) \times \ldots \times \mathring{\varphi}^{-1}(\mathbf{c}'_r)$.
   - Sample $(\mathbf{x}_{r,n_{\text{unf}}+1}, \ldots, \mathbf{x}_{r,n_{\text{spl}}})$ uniformly at random in $\mathring{\varphi}^{-1}(\mathbf{c}_r) \times \ldots \times \mathring{\varphi}^{-1}(\mathbf{c}_r)$.

   (iii)  Sample $\mathbf{x}^{\text{test}}$ uniformly at random in $\mathring{\varphi}^{-1}(\mathbf{c}'_1)$.

---

Step (i) of the above sampling process consists in selecting at random a task $\mathcal{T}$ among all possible tasks. Step (ii) consists in generating a training set[2] $S \in \mathcal{X}^{R \times n_{spl}}$ exactly as depicted on Figure 1: each unfamiliar sequence of concept $\mathbf{c}'_r$ generates $n_{\text{unf}}$ sentences, whereas each familiar sequence of concept $\mathbf{c}_r$ generates $n_{\text{fam}}$ sentences (recall that the number of samples per category is $n_{\text{spl}} = n_{\text{unf}} + n_{\text{fam}}$). Finally step (iii) consists in randomly generating an unfamiliar test sentence $\mathbf{x}^{\text{test}} \in \mathcal{X}$. Without loss of generality we assume that this test sentence is generated by the unfamiliar sequence of concept $\mathbf{c}'_1$.

We denote by $\varrho_{\text{DM}}$ the p.d.f. of the sampling process DM. This function is defined on the sample space

$$\Omega_{\text{DM}} := \big(\Phi \times \mathcal{C}^{2R}\big) \times \mathcal{X}^{R \times n_{spl}} \times \mathcal{X} \,.$$

A sample from $\Omega_{\text{DM}}$ takes the form

$$\omega = \Big( \underbrace{\varphi \,;\, \mathbf{c}_1, \ldots, \mathbf{c}_R \,;\, \mathbf{c}'_1, \ldots, \mathbf{c}'_R}_{\text{The task}} \,;\, \underbrace{\mathbf{x}_{1,1}, \ldots, \mathbf{x}_{1,n_{\text{spl}}} \,;\, \ldots \,;\, \mathbf{x}_{R,1}, \ldots, \mathbf{x}_{R,n_{\text{spl}}}}_{\text{The training sentences}} \,;\, \underbrace{\mathbf{x}^{\text{test}}}_{\text{The test sentence}} \Big)$$

and we have the following formula for $\varrho_{\text{DM}}$

$$\varrho_{\text{DM}}(\omega) := \frac{1}{|\Phi||\mathcal{C}|^{2R}} \prod_{r=1}^{R} \left( \frac{\mathbf{1}_{\mathring{\varphi}^{-1}(\mathbf{c}'_r)}(\mathbf{x}_{r,1})}{|\mathring{\varphi}^{-1}(\mathbf{c}'_r)|} \prod_{s=2}^{n_{\text{spl}}} \frac{\mathbf{1}_{\mathring{\varphi}^{-1}(\mathbf{c}_r)}(\mathbf{x}_{r,s})}{|\mathring{\varphi}^{-1}(\mathbf{c}_r)|} \right) \frac{\mathbf{1}_{\mathring{\varphi}^{-1}(\mathbf{c}'_1)}(\mathbf{x}^{\text{test}})}{|\mathring{\varphi}^{-1}(\mathbf{c}'_1)|} \tag{22}$$

where $\mathbf{1}_{\mathring{\varphi}^{-1}(\mathbf{c}_r)}$ and $\mathbf{1}_{\mathring{\varphi}^{-1}(\mathbf{c}'_r)}$ denote the indicator functions of the set $\mathring{\varphi}^{-1}(\mathbf{c}_r)$ and $\mathring{\varphi}^{-1}(\mathbf{c}_r)$ respectively. Let us compute a few marginals of $\varrho_{\text{DM}}$ in order to verify that it is indeed the p.d.f. of the

---

[2]We refer to $S$ as the 'training set'. In our formalism however it is not a set, but an element of $\mathcal{X}^{R \times n_{spl}}$.

sampling process DM. Writing $\omega = (\mathcal{T}, S, \mathbf{x}^{\text{test}})$, summing over the variables $S$ and $\mathbf{x}^{\text{test}}$, and using the fact that $\sum_{\mathbf{x} \in \mathcal{X}} \mathbf{1}_{\mathring{\varphi}^{-1}(\mathbf{c})}(\mathbf{x}) = |\mathring{\varphi}^{-1}(\mathbf{c})|$, we obtain

$$\sum_{S \in \mathcal{C}^{2R}} \sum_{\mathbf{x}^{\text{test}} \in \mathcal{X}} \varrho_{\text{DM}}(\mathcal{T}, S, \mathbf{x}^{\text{test}}) = \frac{1}{|\Phi||\mathcal{C}|^{2R}}.$$

This shows that each task $\mathcal{T}$ is equiprobable. Summing over the variable $S$ gives

$$\sum_{S \in \mathcal{C}^{2R}} \varrho_{\text{DM}}(\mathcal{T}, S, \mathbf{x}^{\text{test}}) = \frac{1}{|\Phi||\mathcal{C}|^{2R}} \frac{\mathbf{1}_{\mathring{\varphi}^{-1}(\mathbf{c}_1')}(\mathbf{x}^{\text{test}})}{|\mathring{\varphi}^{-1}(\mathbf{c}_1')|}$$

This shows that, given a task $\mathcal{T}$, the test sentence $\mathbf{x}^{\text{test}}$ is obtained by sampling uniformly at random from $\mathring{\varphi}^{-1}(\mathbf{c}_1')$. A similar calculation shows that, given a task $\mathcal{T}$, the train sentence $\mathbf{x}_{r,s}$ is obtained by sampling uniformly at random from $\mathring{\varphi}^{-1}(\mathbf{c}_r)$ if $s \geq 1$, and from $\mathring{\varphi}^{-1}(\mathbf{c}_r')$ if $s = 1$.

Given a feature space $\mathcal{F}$ and a feature map $\psi : \mathcal{X} \to \mathcal{F}$, we define the events $E_{\mathcal{F}, \psi} \subset \Omega_{\text{DM}}$ as follow:

$$E_{\mathcal{F}, \psi} = \Big\{ \omega \in \Omega_{\text{DM}} : \text{ There exists } 1 \leq s^* \leq n_{\text{spl}} \text{ such that}$$

$$\langle \psi(\mathbf{x}^{\text{test}}), \psi(\mathbf{x}_{1,s^*}) \rangle_{\mathcal{F}} > \langle \psi(\mathbf{x}^{\text{test}}), \psi(\mathbf{x}_{r,s}) \rangle_{\mathcal{F}} \text{ for all } 2 \leq r \leq R \text{ and all } 1 \leq s \leq n_{\text{spl}} \Big\}. \quad (23)$$

Note that this event consists in all the outcomes $\omega = (\mathcal{T}, S, \mathbf{x}^{\text{test}})$ for which the feature map $\psi$ associates the test sentence $\mathbf{x}^{\text{test}}$ to a train point $\mathbf{x}_{r,s}$ from the first category. Since by construction, $\mathbf{x}^{\text{test}}$ belongs to the first category, $E_{\mathcal{F}, \psi}$ consists in all the outcomes for which the nearest neighbor classification rule is 'successful'. As a consequence, when $\mathfrak{T} = \Phi \times \mathcal{Z}^{2R}$, the generalization error can be expressed as

$$\overline{\text{err}}(\mathcal{F}, \psi, \mathfrak{T}) = 1 - \mathbb{P}_{\text{DM}} \Big[ E_{\mathcal{F}, \psi} \Big] \quad (24)$$

where $\mathbb{P}_{\text{DM}}$ denote the probability measure on $\Omega_{\text{DM}}$ induced by $\varrho_{\text{DM}}$. Equation (24) should be viewed as our 'fully formal' definition of the quantity $\overline{\text{err}}(\mathcal{F}, \psi, \mathfrak{T})$, as opposed to the more informal definition given earlier by equations (6) and (7).

The goal of this section is to prove inequality (14), which, in light of (24) is equivalent to

$$\sup_{\mathcal{F}, \psi} \mathbb{P}_{\text{DM}} \Big[ E_{\mathcal{F}, \psi} \Big] \leq \frac{1}{|\mathcal{X}|} \sum_{\mathbf{x} \in \mathcal{X}} \mathcal{H}_{2R-1} \left( K^\star(\mathbf{x}, \cdot) \right) + \frac{1}{R}. \quad (25)$$

which in turn equivalent to

$$\sup_{\substack{K : \mathcal{X} \times \mathcal{X} \to \mathbb{R} \\ K \text{ pos. semi-def.}}} \mathbb{P}_{\text{DM}} \Big[ E_K \Big] \leq \frac{1}{|\mathcal{X}|} \sum_{\mathbf{x} \in \mathcal{X}} \mathcal{H}_{2R-1} \left( K^\star(\mathbf{x}, \cdot) \right) + \frac{1}{R} \quad (26)$$

where the event $E_K$ is defined by

$$E_K = \Big\{ \omega \in \Omega_{\text{DM}} : \text{ There exists } 1 \leq s^* \leq n_{\text{spl}} \text{ such that}$$

$$K(\mathbf{x}^{\text{test}}, \mathbf{x}_{1,s^*}) > K(\mathbf{x}^{\text{test}}, \mathbf{x}_{r,s}) \text{ for all } 2 \leq r \leq R \text{ and all } 1 \leq s \leq n_{\text{spl}} \Big\} \quad (27)$$

and where the supremum is taken over all kernels $K : \mathcal{X} \times \mathcal{X} \to \mathbb{R}$ which are symmetric positive semidefinite. We will actually prove a slightly stronger result, namely

$$\sup_{\substack{K : \mathcal{X} \times \mathcal{X} \to \mathbb{R} \\ K \text{ is symmetric}}} \mathbb{P}_{\text{DM}} \Big[ E_K \Big] \leq \frac{1}{|\mathcal{X}|} \sum_{\mathbf{x} \in \mathcal{X}} \mathcal{H}_{2R-1} \left( K^\star(\mathbf{x}, \cdot) \right) + \frac{1}{R} \quad (28)$$

where the supremum is taken over all functions $K : \mathcal{X} \times \mathcal{X} \to \mathbb{R}$ that satisfy $K(\mathbf{x}, \mathbf{y}) = K(\mathbf{y}, \mathbf{x})$ for all $(\mathbf{x}, \mathbf{y}) \in \mathcal{X} \times \mathcal{X}$. The rest of the section is devoted to proving (28).

In Subsection B.1 we start by considering a simpler data model — for this simpler data model we are able to show that the function $\psi^\star$ implicitly defined by (12) is the best possible feature map (we actually only work with the associated kernel $K^\star$, and never need $\psi^\star$ itself). We also show that the success rate is *exactly equal* to the permuted moment of $K^\star$ — see Theorem 4, which is is the central result of this section. In the remaining subsections, namely Subsection B.2 and Subsection B.3, we leverage the bound obtained for the simpler data model in order to obtain bound (28) for the main data model. These two subsections are mostly notational. The core of the analysis takes place in Subsection B.1.

### B.1    A SIMPLER DATA MODEL

We start by presenting the sampling process associated with our simpler datamodel.

**Sampling Process SDM:**

(i) Sample $\varphi$ uniformly at random in $\Phi$. Sample $\mathbf{c}_1, \mathbf{c}_2, \ldots, \mathbf{c}_{t+1}$ uniformly at random in $\mathcal{Z}$.

(ii) For $1 \leq r \leq t+1$: Sample $\mathbf{x}_r$ uniformly at random in $\mathring{\varphi}^{-1}(\mathbf{c}_r)$.

(iii) Sample $\mathbf{x}^{\text{test}}$ uniformly at random in $\mathring{\varphi}^{-1}(\mathbf{c}_1)$.

The function

$$\varrho_{\text{SDM}}(\varphi; \mathbf{c}_1, \ldots, \mathbf{c}_{t+1}; \mathbf{x}_1, \ldots, \mathbf{x}_{t+1}; \mathbf{x}^{\text{test}}) := \frac{1}{|\Phi||\mathcal{C}|^{t+1}} \left( \prod_{r=1}^{t+1} \frac{\mathbf{1}_{\mathring{\varphi}^{-1}(\mathbf{c}_r)}(\mathbf{x}_r)}{|\mathring{\varphi}^{-1}(\mathbf{c}_r)|} \right) \frac{\mathbf{1}_{\mathring{\varphi}^{-1}(\mathbf{c}_1)}(\mathbf{x}^{\text{test}})}{|\mathring{\varphi}^{-1}(\mathbf{c}_1)|} \tag{29}$$

on $\Omega_{\text{SDM}} := \Phi \times \mathcal{C}^{t+1} \times \mathcal{X}^{t+2}$ is the p.d.f. of the above sampling process. We use $\mathbb{P}_{\text{SDM}}$ to denote the probability measure on $\Omega_{\text{SDM}}$ induced by this function. The identity in the next theorem is the central result of this section.

**Theorem 4.** *Let $\mathcal{K}$ denote the set of all symmetric functions from $\mathcal{X} \times \mathcal{X}$ to $\mathbb{R}$. Then*

$$\sup_{K \in \mathcal{K}} \mathbb{P}_{\text{SDM}}\left[ K(\mathbf{x}^{\text{test}}, \mathbf{x}_1) > K(\mathbf{x}^{\text{test}}, \mathbf{x}_r) \text{ for all } 2 \leq r \leq t+1 \right] = \frac{1}{|\mathcal{X}|} \sum_{\mathbf{x} \in \mathcal{X}} \mathcal{H}_t(K_{\mathbf{x}}^{\star}) \tag{30}$$

In (30), $K_{\mathbf{x}}^{\star}$ stands for the function $K(\mathbf{x}, \cdot)$. Theorem 4 establishes an intimate connection between the permuted moment and the ability of any fixed feature map (or equivalently, any fixed kernel) to generalize well in our framework. The sampling process considered in this theorem involves two points, $\mathbf{x}^{\text{test}}$ and $\mathbf{x}_1$, generated by the same sequence of concepts $\mathbf{c}_1$, and $t$ 'distractor' points $\mathbf{x}_2, \ldots, \mathbf{x}_{t+1}$ generated by different sequences of concepts. Success for the kernel $K$ means correctly recognizing that $\mathbf{x}^{\text{test}}$ is more 'similar' to $\mathbf{x}_1$ than any of the distractors, and the success rate in (30) precisely quantifies its ability to do so as a function of the number $t$ of distractors. The theorem shows that the probability of success for the *best possible kernel* at this task is *exactly equal* to the averaged $t^{th}$-permuted moment of $K_{\mathbf{x}}^{\star}$, so it elegantly quantifies the generalization ability of the best possible fixed feature map in term of the permuted moment. We also provide an explicit construction for a kernel $K(\mathbf{x}, \mathbf{y})$ that achieves the supremum in (30) — First, choose a kernel $\varepsilon(\mathbf{x}, \mathbf{y})$ that satisfies

(i) $\varepsilon(\mathbf{x}, \mathbf{y}) \neq \varepsilon(\mathbf{x}, \mathbf{z})$ for all $\mathbf{x}, \mathbf{y}, \mathbf{z} \in \mathcal{X}$ with $\mathbf{y} \neq \mathbf{z}$.

(ii) $0 \leq \varepsilon(\mathbf{x}, \mathbf{y}) \leq 1$ for all $\mathbf{x}, \mathbf{y} \in \mathcal{X}$.

and then define the following perturbation

$$K(\mathbf{x}, \mathbf{y}) = K^{\star}(\mathbf{x}, \mathbf{y}) + \varepsilon(\mathbf{x}, \mathbf{y})/(2s_c^L |\Phi|) \tag{31}$$

of $K^{\star}$. Any such kernel is a maximizer of the optimization problem in (30), so we may think of perturbations of $K^{\star}$ as bona-fide optimal.

The rest of this subsection is devoted to the proof of Theorem 4, and we also show, in the course of the proof, that (31) is a maximizer of the optimization problem in (30). We use $\mathcal{K}$ to denote the set of all symmetric functions from $\mathcal{X} \times \mathcal{X}$ to $\mathbb{R}$. We will refers to such functions as 'kernel' despite the fact that these functions are not necessarily positive semi-definite.

Proving Theorem 4 requires that we study the following optimization problem:

$$\text{Maximize} \quad \mathfrak{E}(K) := \mathbb{P}_{SDM}\left[ K(\mathbf{x}^{\text{test}}, \mathbf{x}_1) > K(\mathbf{x}^{\text{test}}, \mathbf{x}_r) \text{ for all } 2 \leq r \leq t+1 \right] \tag{32}$$

$$\text{over all kernels } K \in \mathcal{K}. \tag{33}$$

We recall the definition of the optimal kernel,

$$K^{\star}(\mathbf{x}, \mathbf{y}) = \frac{1}{s_c^L} \frac{\left| \{ \varphi \in \Phi : \varphi(x_\ell) = \varphi(y_\ell) \text{ for all } 1 \leq \ell \leq L \} \right|}{|\Phi|} \tag{34}$$

where $s_c = n_w/s_c$ denotes the size of a concept. We start with the following simple lemma:

**Lemma 4.** *The function $K_{\mathbf{x}}^\star(\cdot) = K^\star(\mathbf{x}, \cdot)$ is a probability distribution on $\mathcal{X}$.*

*Proof.* First note that $K^\star$ can be written as

$$K^\star(\mathbf{x}, \mathbf{y}) = \frac{1}{s_c^L} \frac{|\{\varphi \in \Phi : \mathring{\varphi}(\mathbf{x}) = \mathring{\varphi}(\mathbf{y})\}|}{|\Phi|} = \frac{1}{s_c^L |\Phi|} \sum_{\varphi \in \Phi} \mathbf{1}_{\{\mathring{\varphi}(\mathbf{x}) = \mathring{\varphi}(\mathbf{y})\}} \tag{35}$$

Since $\varphi$ maps exactly $s_c$ words to each concept $c \in \{1, \ldots, n_c\}$, we have that

$$|\{\mathbf{x} \in \mathcal{X} : \mathring{\varphi}(\mathbf{x}) = \mathbf{c}\}| = s_c^L \qquad \text{for all } \mathbf{c} \in \mathcal{C}. \tag{36}$$

Therefore

$$\sum_{\mathbf{y} \in \mathcal{X}} K^\star(\mathbf{x}, \mathbf{y}) = \frac{1}{s_c^L |\Phi|} \sum_{\varphi \in \Phi} \sum_{\mathbf{y} \in \mathcal{X}} \mathbf{1}_{\{\mathring{\varphi}(\mathbf{x}) = \mathring{\varphi}(\mathbf{y})\}} = \frac{1}{s_c^L |\Phi|} \sum_{\varphi \in \Phi} |\{\mathbf{y} \in \mathcal{X} : \mathring{\varphi}(\mathbf{y}) = \mathring{\varphi}(\mathbf{x})\}| = 1$$

$\square$

We now show that the marginal of the p.d.f. $\varrho_{\text{SDM}}$ is related to $K^\star$.

**Lemma 5.** *For all $\mathbf{x}_1, \ldots, \mathbf{x}_{t+1}$ and $\mathbf{x}^{test}$ in $\mathcal{X}$ we have*

$$\sum_{\varphi \in \Phi} \sum_{\mathbf{c}_1 \in \mathcal{C}} \cdots \sum_{\mathbf{c}_{t+1} \in \mathcal{C}} \varrho_{\text{SDM}}(\varphi; \mathbf{c}_1, \ldots, \mathbf{c}_{t+1}; \mathbf{x}_1, \ldots, \mathbf{x}_{t+1}; \mathbf{x}^{test}) = \frac{1}{|\mathcal{X}|^{t+1}} K^\star(\mathbf{x}_1, \mathbf{x}^{test}).$$

*Proof.* Identity (36) can be expressed as $\left|\mathring{\varphi}^{-1}(\mathbf{c})\right| = s_c^L$ for all $\mathbf{c} \in \mathcal{C}$. As a consequence, definition (29) of $\varrho_{\text{SDM}}(\omega)$ simplifies to

$$\varrho_{\text{SDM}}(\omega) = \alpha \left( \prod_{r=1}^{t+1} \mathbf{1}_{\mathring{\varphi}^{-1}(\mathbf{c}_r)}(\mathbf{x}_r) \right) \mathbf{1}_{\mathring{\varphi}^{-1}(\mathbf{c}_1)}(\mathbf{x}^{test}) \tag{37}$$

where the constant $\alpha$ is given by

$$\alpha = \frac{1}{|\Phi||\mathcal{C}|^{t+1} s_c^{L(t+2)}} = \frac{1}{|\Phi| n_c^{L(t+1)} s_c^{L(t+2)}} = \frac{1}{|\Phi||\mathcal{X}|^{t+1} s_c^L}$$

In the above we have used the fact that $|\mathcal{C}| = n_c^L$ and $|\mathcal{X}| = n_w^L$. We then note that the identity $\mathbf{1}_{\mathring{\varphi}^{-1}(\mathbf{c})}(\mathbf{x}) = \mathbf{1}_{\{\mathring{\varphi}(\mathbf{x}) = \mathbf{c}\}}$ implies

$$\sum_{\mathbf{c} \in \mathcal{C}} \mathbf{1}_{\mathring{\varphi}^{-1}(\mathbf{c})}(\mathbf{x}) = \sum_{\mathbf{c} \in \mathcal{C}} \mathbf{1}_{\{\mathring{\varphi}(\mathbf{x}) = \mathbf{c}\}} = 1 \tag{38}$$

$$\sum_{\mathbf{c} \in \mathcal{C}} \left( \mathbf{1}_{\mathring{\varphi}^{-1}(\mathbf{c})}(\mathbf{x}) \, \mathbf{1}_{\mathring{\varphi}^{-1}(\mathbf{c})}(\mathbf{y}) \right) = \sum_{\mathbf{c} \in \mathcal{C}} \left( \mathbf{1}_{\{\mathring{\varphi}(\mathbf{x}) = \mathbf{c}\}} \, \mathbf{1}_{\{\mathring{\varphi}(\mathbf{y}) = \mathbf{c}\}} \right) = \mathbf{1}_{\{\mathring{\varphi}(\mathbf{x}) = \mathring{\varphi}(\mathbf{y})\}} \tag{39}$$

for all $\mathbf{x}, \mathbf{y} \in \mathcal{X}$. Summing (37) over the variables $\mathbf{c}_1, \ldots, \mathbf{c}_{t+1}$ we obtain

$$\sum_{\mathbf{c}_1 \in \mathcal{C}} \cdots \sum_{\mathbf{c}_{t+1} \in \mathcal{C}} \varrho_{\text{SDM}}(\omega) = \alpha \sum_{\mathbf{c}_1 \in \mathcal{C}} \cdots \sum_{\mathbf{c}_{t+1} \in \mathcal{C}} \left( \mathbf{1}_{\mathring{\varphi}^{-1}(\mathbf{c}_1)}(\mathbf{x}_1) \, \mathbf{1}_{\mathring{\varphi}^{-1}(\mathbf{c}_1)}(\mathbf{x}^{test}) \prod_{r=2}^{t+1} \mathbf{1}_{\mathring{\varphi}^{-1}(\mathbf{c}_r)}(\mathbf{x}_r) \right)$$

$$= \alpha \sum_{\mathbf{c}_1 \in \mathcal{C}} \left( \mathbf{1}_{\mathring{\varphi}^{-1}(\mathbf{c}_1)}(\mathbf{x}_1) \, \mathbf{1}_{\mathring{\varphi}^{-1}(\mathbf{c}_1)}(\mathbf{x}^{test}) \right) \sum_{\mathbf{c}_2 \in \mathcal{C}} \cdots \sum_{\mathbf{c}_{t+1} \in \mathcal{C}} \left( \prod_{r=2}^{t+1} \mathbf{1}_{\mathring{\varphi}^{-1}(\mathbf{c}_r)}(\mathbf{x}_r) \right)$$

$$= \alpha \, \mathbf{1}_{\{\mathring{\varphi}(\mathbf{x}_1) = \mathring{\varphi}(\mathbf{x}^{test})\}}$$

where we have used (38) and (39) to obtain the last equality. Summing the above over the variable $\varphi$ gives $K^\star(\mathbf{x}_1, \mathbf{x}^{test})/|\mathcal{X}|^{t+1}$. $\square$

The next lemma provides a purely algebraic (as opposed to probabilistic) formulation for the functional $\mathfrak{E}(K)$ defined in (32).

**Lemma 6.** *The functional* $\mathfrak{E} : \mathcal{K} \to \mathbb{R}$ *can be expressed as*

$$\mathfrak{E}(K) = \frac{1}{|\mathcal{X}|} \sum_{\mathbf{x} \in \mathcal{X}} \sum_{\mathbf{y} \in \mathcal{X}} K^\star(\mathbf{x}, \mathbf{y}) \left( \frac{|\{\mathbf{z} \in \mathcal{X} : K(\mathbf{x}, \mathbf{z}) < K(\mathbf{x}, \mathbf{y})\}|}{|\mathcal{X}|} \right)^t. \tag{40}$$

*Proof.* Let $g : \mathcal{X}^{t+2} \times \mathcal{K} \to \{0, 1\}$ be the indicator function defined by

$$g(\mathbf{x}_1, \ldots, \mathbf{x}_{t+1}, \mathbf{x}^{\text{test}}, K) = \begin{cases} 1 & \text{if } K(\mathbf{x}^{\text{test}}, \mathbf{x}_1) > K(\mathbf{x}^{\text{test}}, \mathbf{x}_r) \text{ for all } 2 \leq r \leq t+1 \\ 0 & \text{otherwise} \end{cases}$$

Let $\omega$ denote the sample $(\varphi; \mathbf{c}_1, \ldots, \mathbf{c}_{t+1}; \mathbf{x}_1, \ldots, \mathbf{x}_{t+1}; \mathbf{x}^{\text{test}})$. Since $g$ only depends on the last $t+2$ variables of $\omega$, we have

$$\mathfrak{E}(K) = \mathbb{P}_{SDM}\left[ K(\mathbf{x}^{\text{test}}, \mathbf{x}_1) > K(\mathbf{x}^{\text{test}}, \mathbf{x}_r) \text{ for all } 2 \leq r \leq t+1 \right] \tag{41}$$

$$= \sum_{\varphi \in \Phi} \sum_{\mathbf{c}_1 \in \mathcal{C}} \cdots \sum_{\mathbf{c}_{t+1} \in \mathcal{C}} \sum_{\mathbf{x}_1 \in \mathcal{X}} \cdots \sum_{\mathbf{x}_{t+1} \in \mathcal{X}} \sum_{\mathbf{x}^{\text{test}} \in \mathcal{X}} g(\mathbf{x}_1, \ldots, \mathbf{x}_{t+1}, \mathbf{x}^{\text{test}}, K) \, \varrho_{\text{SDM}}(\omega) \tag{42}$$

$$= \sum_{\mathbf{x}_1 \in \mathcal{X}} \cdots \sum_{\mathbf{x}_{t+1} \in \mathcal{X}} \sum_{\mathbf{x}^{\text{test}} \in \mathcal{X}} g(\mathbf{x}_1, \ldots, \mathbf{x}_{t+1}, \mathbf{x}^{\text{test}}, K) \left( \sum_{\varphi \in \Phi} \sum_{\mathbf{c}_1 \in \mathcal{C}} \cdots \sum_{\mathbf{c}_{t+1} \in \mathcal{C}} \varrho_{\text{SDM}}(\omega) \right) \tag{43}$$

$$= \sum_{\mathbf{x}_1 \in \mathcal{X}} \cdots \sum_{\mathbf{x}_{t+1} \in \mathcal{X}} \sum_{\mathbf{x}^{\text{test}} \in \mathcal{X}} g(\mathbf{x}_1, \ldots, \mathbf{x}_{t+1}, \mathbf{x}^{\text{test}}, K) \, \frac{1}{|\mathcal{X}|^{t+1}} \, K^\star(\mathbf{x}_1, \mathbf{x}^{\text{test}}) \tag{44}$$

$$= \frac{1}{|\mathcal{X}|} \sum_{\mathbf{x}_1 \in \mathcal{X}} \sum_{\mathbf{x}^{\text{test}} \in \mathcal{X}} K^\star(\mathbf{x}_1, \mathbf{x}^{\text{test}}) \left( \frac{1}{|\mathcal{X}|^t} \sum_{\mathbf{x}_2 \in \mathcal{X}} \cdots \sum_{\mathbf{x}_{t+1} \in \mathcal{X}} g(\mathbf{x}_1, \ldots, \mathbf{x}_{t+1}, \mathbf{x}^{\text{test}}, K) \right) \tag{45}$$

where we have used Lemma 5 to go from (43) to (44). Writing the indicator function $g$ as a product of indicator functions,

$$g(\mathbf{x}_1, \ldots, \mathbf{x}_{t+1}, \mathbf{x}^{\text{test}}, K) = \prod_{r=2}^{t+1} \mathbf{1}_{\{K(\mathbf{x}^{\text{test}}, \mathbf{x}_1) > K(\mathbf{x}^{\text{test}}, \mathbf{x}_r)\}}$$

we obtain the following expression for the term appearing between parentheses in (45):

$$\frac{1}{|\mathcal{X}|^t} \sum_{\mathbf{x}_2 \in \mathcal{X}} \cdots \sum_{\mathbf{x}_{t+1} \in \mathcal{X}} g(\mathbf{x}_1, \ldots, \mathbf{x}_{t+1}, \mathbf{x}^{\text{test}}, K) = \frac{1}{|\mathcal{X}|^t} \prod_{r=2}^{t+1} \left( \sum_{\mathbf{x}_r \in \mathcal{X}} \mathbf{1}_{\{K(\mathbf{x}^{\text{test}}, \mathbf{x}_1) > K(\mathbf{x}^{\text{test}}, \mathbf{x}_r)\}} \right)$$

$$= \frac{1}{|\mathcal{X}|^t} \left( \sum_{\mathbf{z} \in \mathcal{X}} \mathbf{1}_{\{K(\mathbf{x}^{\text{test}}, \mathbf{x}_1) > K(\mathbf{x}^{\text{test}}, \mathbf{z})\}} \right)^t$$

$$= \left( \frac{|\{\mathbf{z} \in \mathcal{X} : K(\mathbf{x}^{\text{test}}, \mathbf{x}_1) > K(\mathbf{x}^{\text{test}}, \mathbf{z})\}|}{|\mathcal{X}|} \right)^t$$

Changing the name of variables $\mathbf{x}^{\text{test}}, \mathbf{x}_1$ to $\mathbf{x}, \mathbf{y}$ gives (40). $\square$

We now use expression (40) for $\mathfrak{E}(K)$ and reformulate optimization problem (32)-(33) into an equivalent optimization problem over symmetric matrices. Putting an arbitrary ordering on the set $\mathcal{X}$ (starting with $i = 0$) and denoting by $K_{ij}^\star$ the value of the kernel $K^\star$ on the pair that consists of the $i^{th}$ and $j^{th}$ element of $\mathcal{X}$, we see that optimization problem (32)-(33) can be written as

$$\text{Maximize} \quad \mathfrak{E}(K) := \frac{1}{N} \sum_{i=0}^{N-1} \sum_{j=0}^{N-1} K_{ij}^\star \left( \frac{|\{j' \in [N] : K_{ij'} < K_{ij}\}|}{N} \right)^t \tag{46}$$

$$\text{over all symmetric matrices } K \in \mathbb{R}^{N \times N} \tag{47}$$

In the above we have used the letter $N$ to denote the cardinality of $\mathcal{X}$, that is $N = n_w^L$, and we have used the notation $[N] = \{0, 1, \ldots, N-1\}$. Before solving the matrix optimization problem (46)-(47), we start with a simpler vector optimization problem. Let $\mathbf{p}^\star$ be a probability vector, that is $\mathbf{p}^\star \in \mathbb{R}_+^N$ with $\sum_{i=1}^N p_i = 1$, and consider the optimization problem:

$$\text{Maximize} \quad \mathfrak{e}(\mathbf{v}) := \sum_{j=0}^{N-1} p_j^\star \left( \frac{|\{j' \in [N] : v_{j'} < v_j\}|}{N} \right)^t \tag{48}$$

$$\text{over all vector } \mathbf{v} \in \mathbb{R}^N. \tag{49}$$

Recall from Definition 1 that an ordering permutation of a vector $\mathbf{v}$ is a permutation that sorts its entries from smallest to largest. We will say that two vectors $\mathbf{v}, \mathbf{w} \in \mathbb{R}^N$ have *the same ordering* if there exist $\sigma \in S_N$ which is ordering for both $\mathbf{v}$ and $\mathbf{w}$. The following lemma is key — it shows that the optimization problem (48)–(49) has a simple solution.

**Lemma 7.** *The following identity*

$$\sup_{\mathbf{v} \in \mathbb{R}^N} \mathfrak{e}(\mathbf{v}) = \mathcal{H}_t(\mathbf{p}^\star)$$

*holds. Moreover, the supremum is achieved by any vector $\mathbf{v} \in \mathbb{R}^N$ that has mutually distinct entries[3] and that has the same ordering than $\mathbf{p}^\star$.*

*Proof.* Let $\text{Distinct}(\mathbb{R}^N)$ denote the vectors of $\mathbb{R}^N$ that have mutually distinct entries. We will first show that

$$\sup_{\mathbf{v} \in \mathbb{R}^N} \mathfrak{e}(\mathbf{v}) = \sup_{\mathbf{v} \in \text{Distinct}(\mathbb{R}^N)} \mathfrak{e}(\mathbf{v}). \tag{50}$$

To do this we show that for any $\mathbf{v} \in \mathbb{R}^N$, there exists $\mathbf{w} \in \text{Distinct}(\mathbb{R}^N)$ such that

$$|\{j' \in [N] : v_{j'} < v_j\}| \le |\{j' \in [N] : w_{j'} < w_j\}| \qquad \text{for all } 0 \le j \le N-1. \tag{51}$$

There are many ways to construct such a $\mathbf{w}$. One way is to simply set $w_j = \sigma^{-1}(j)$ for some permutation $\sigma$ that orders $v$. Indeed, note that $\sigma^{-1}(j)$ provides the position of $v_j$ in the sequence of inequality (18). Therefore if $v_{j'} < v_j$ we must have that $\sigma^{-1}(j') < \sigma^{-1}(j)$. This implies

$$\{j' \in [N] : v_{j'} < v_j\} \subset \{j' \in [N] : \sigma^{-1}(j') < \sigma^{-1}(j)\} \qquad \text{for all } j \in [N]$$

which in turn implies (51).

Because of (50) we can now restrict our attention to $\mathbf{v} \in \text{Distinct}(\mathbb{R}^N)$. Note that if $\mathbf{v} \in \text{Distinct}(\mathbb{R}^N)$, then it has a unique ordering permutation $\sigma$,

$$v_{\sigma(0)} < v_{\sigma(1)} < v_{\sigma(2)} < v_{\sigma(3)} < \ldots < v_{\sigma(N-1)}$$

and, recalling that $\sigma^{-1}(j)$ provide the position of $v_j$ in the above ordering, we clearly have that

$$|\{j' \in [N] : v_j' < v_j\}| = \sigma^{-1}(j).$$

Therefore, if $\mathbf{v} \in \text{Distinct}(\mathbb{R}^N)$ and if $\sigma$ denotes its unique ordering permutation, $\mathfrak{e}(\mathbf{v})$ can be expressed as

$$\mathfrak{e}(\mathbf{v}) = \sum_{j=0}^{N-1} p_j^\star \left( \frac{|\{j' \in [N] : v_{j'} < v_j\}|}{N} \right)^t = \sum_{j=0}^{N-1} p_j^\star \left( \frac{\sigma^{-1}(j)}{N} \right)^t = \sum_{j=0}^{N-1} p_{\sigma(j)}^\star \, (j/N)^t \tag{52}$$

Looking at definition (11) of the permuted moment, it is then clear that $\mathfrak{e}(\mathbf{v}) \le \mathcal{H}_t(\mathbf{p}^\star)$ for all $\mathbf{v} \in \text{Distinct}(\mathbb{R}^N)$. We then note that if $\mathbf{v} \in \text{Distinct}(\mathbb{R}^N)$ has the same ordering than $\mathbf{p}^\star$, then its unique ordering permutation $\sigma$ must also be an ordering permutation of $\mathbf{p}^\star$. Then (52) combined with Lemma 1 implies that $\mathfrak{e}(\mathbf{v}) = \mathcal{H}_t(\mathbf{p}^\star)$. This concludes the proof. $\qquad \square$

---

[3]That is, $v_i \ne v_j$ for all $i \ne j$.

Relaxing the symmetric constraint in the optimization problem (46)-(47) gives the following unconstrained problem over all $N$-by-$N$ matrices:

$$\text{Maximize} \quad \mathfrak{E}(K) := \frac{1}{N} \sum_{i=0}^{N-1} \sum_{j=0}^{N-1} K_{ij}^{\star} \left( \frac{|\{j' \in [N] : K_{ij'} < K_{ij}\}|}{N} \right)^{t} \tag{53}$$

$$\text{over all matrices } K \in \mathbb{R}^{N \times N} \tag{54}$$

Let us denote by $K_{i,:}^{\star}$ the $i^{th}$ row of the matrix $K^{\star}$ and remark that $K_{i,:}^{\star}$ is a probability vector (because $K^{\star}(\mathbf{x}, \cdot)$ is a probability distribution on $\mathcal{X}$, see Lemma 4). We then note that the above unconstrained problem decouples into $N$ separate optimization problems of the type (48)-(49) in which the probability vector $\mathbf{p}^{\star}$ must be replaced by the probability vector $K_{i,:}^{\star}$. Using Lemma 7 we therefore have that any $K \in \mathbb{R}^{N \times N}$ that satisfies, for each $0 \le i \le N - 1$,

(a) The entries of $K_{i,:}$ are mutually distinct,

(b) $K_{i,:}$ and $K_{i,:}^{\star}$ have the same ordering,

must be a solution of (53)-(54). Lemma 7 also gives:

$$\sup_{K \in \mathbb{R}^{N \times N}} \mathfrak{E}(K) = \frac{1}{N} \sum_{i=0}^{N-1} \mathcal{H}_t(K_{i,:}^{\star}).$$

Now let $\varepsilon \in \mathbb{R}^{N \times N}$ be a symmetric matrix that satisfies:

(i) $\varepsilon_{ij} \ne \varepsilon_{ij'}$ for all $i, j, j' \in [N]$ with $j \ne j'$,

(ii) $0 \le \varepsilon_{ij} \le 1$ for all $i, j \in [N]$,

and define the following perturbation of the matrix $K^{\star}$:

$$K = K^{\star} + \frac{0.5}{s_c^L |\Phi|} \; \varepsilon \tag{55}$$

Recalling definition (35) of the kernel $K^{\star}$, it is clear that for each $i, j \in [N]$, we have

$$K_{ij}^{\star} = \frac{\ell}{s_c^L |\Phi|} \qquad \text{for some integer } \ell. \tag{56}$$

As a consequence perturbing $K^{\star}$ by adding to its entries quantities smaller than $1/(s_c^L|\Phi|)$ can not change the ordering of its rows. Therefore the kernel $K$ defined by (55) satisfies (b). It also satisfies (a). Indeed, if $K_{ij}^{\star} = K_{ij'}^{\star}$ and $j \ne j'$, then we clearly have that $K_{ij} \ne K_{ij'}$ due to (i). On the other hand if $K_{ij}^{\star} \ne K_{ij'}^{\star}$, then $K_{ij} \ne K_{ij'}$ due to (ii) and (56).

We have therefore constructed a symmetric matrix that is a solution of the optimization problem (53)-(54). As a consequence we have

$$\sup_{K \in \mathcal{K}} \mathfrak{E}(K) = \sup_{K \in \mathbb{R}^{N \times N}} \mathfrak{E}(K) = \frac{1}{N} \sum_{i=0}^{N-1} \mathcal{H}_t(K_{i,:}^{\star})$$

where $\mathcal{K}$ should now be interpreted as the set of $N$-by-$N$ symmetric matrices. The above equality proves Theorem 4, and we have also shown that the perturbed kernel (55) achieves the supremum.

### B.2 CONNECTION BETWEEN THE TWO SAMPLING PROCESSES

In this subsection we show that the p.d.f. of Sampling Process SDM can be obtained by marginalizing the p.d.f. of Sampling Process DM over a subset of the variables. We also compute another marginal of $\varrho_{\text{DM}}$ that will prove useful in the next subsection. Recall that

$$\varrho_{\text{DM}}(\omega) = \frac{1}{|\Phi||\mathcal{C}|^{2R}} \prod_{r=1}^{R} \left( \frac{\mathbf{1}_{\mathring{\varphi}^{-1}(\mathbf{c}_r')}(\mathbf{x}_{r,1})}{|\mathring{\varphi}^{-1}(\mathbf{c}_r')|} \prod_{s=2}^{n_{\text{spl}}} \frac{\mathbf{1}_{\mathring{\varphi}^{-1}(\mathbf{c}_r)}(\mathbf{x}_{r,s})}{|\mathring{\varphi}^{-1}(\mathbf{c}_r)|} \right) \frac{\mathbf{1}_{\mathring{\varphi}^{-1}(\mathbf{c}_1')}(\mathbf{x}^{\text{test}})}{|\mathring{\varphi}^{-1}(\mathbf{c}_1')|} \tag{57}$$

on $\Omega_{\mathrm{DM}} := \Phi \times \mathcal{C}^{2R} \times \mathcal{X}^{R \times n_{\mathrm{spl}}+1}$ is the p.d.f. of the sampling process for our main data model. Samples from $\Omega_{\mathrm{DM}}$ take the form

$$\omega = (\varphi \quad ; \quad \mathbf{c}_1, \mathbf{c}_2, \mathbf{c}_3, \ldots, \mathbf{c}_R \quad ; \quad \mathbf{c}'_1, \mathbf{c}'_2, \mathbf{c}'_3, \ldots, \mathbf{c}'_R \quad ;$$
$$\mathbf{x}_{1,1}, \mathbf{x}_{1,2}, \mathbf{x}_{1,3}, \ldots, \mathbf{x}_{1,n_{\mathrm{spl}}} \quad ; \quad \cdots \quad ; \quad \mathbf{x}_{R,1}, \mathbf{x}_{R,2}, \mathbf{x}_{R,3}, \ldots, \mathbf{x}_{R,n_{\mathrm{spl}}} \quad ; \quad \mathbf{x}^{\mathrm{test}})$$

We separate these variables into two groups, $\omega = (\omega_a, \omega_b)$, where

$$\omega_a = (\varphi \quad ; \quad \mathbf{c}_1, \mathbf{c}_2, \mathbf{c}_3, \ldots, \mathbf{c}_R \quad ; \quad \mathbf{c}'_1, \mathbf{c}'_2, \mathbf{c}'_3, \ldots, \mathbf{c}'_R \quad ; \quad \mathbf{x}_{1,1}, \mathbf{x}_{1,2} \quad ; \quad \cdots \quad ; \quad \mathbf{x}_{R,1}, \mathbf{x}_{R,2} \quad ; \quad \mathbf{x}^{\mathrm{test}})$$
$$(58)$$

$$\omega_b = (\mathbf{x}_{1,3}, \mathbf{x}_{1,4}, \ldots, \mathbf{x}_{1,n_{\mathrm{spl}}} \quad ; \quad \cdots \quad ; \quad \mathbf{x}_{R,3}, \mathbf{x}_{R,4}, \ldots, \mathbf{x}_{R,n_{\mathrm{spl}}}) \tag{59}$$

The variable $\omega_a$ belongs to $\Omega_a = \Phi \times \mathcal{C}^{2R} \times \mathcal{X}^{2R+1}$, and the variable $\omega_b$ belongs to $\Omega_b = \mathcal{X}^{R(n_{\mathrm{spl}}-2)}$. Note that the variables in $\omega_a$ contains, among other, $2R$ sequences of concepts and $2R$ training points (the first and second training points of each category). Each of these $2R$ training points is generated by one of the $2R$ sequences of concepts. So the variables involved in $\omega_a$ are generated by a process similar to the one involved in the simpler data model. The following lemma shows that $p.d.f.$ of $\omega_a$, after marginalizing $\omega_b$, is indeed $\varrho_{SDM}$.

**Lemma 8.** *For all $\omega_a \in \Omega_a$ we have*

$$\sum_{\omega_b \in \Omega_b} \varrho_{\mathrm{DM}}(\omega_a, \omega_b) = \frac{1}{|\Phi||\mathcal{C}|^{2R}} \left( \prod_{r=1}^{R} \frac{\mathbf{1}_{\mathring{\varphi}^{-1}(\mathbf{c}'_r)}(\mathbf{x}_{r,1})}{|\mathring{\varphi}^{-1}(\mathbf{c}'_r)|} \frac{\mathbf{1}_{\mathring{\varphi}^{-1}(\mathbf{c}_r)}(\mathbf{x}_{r,2})}{|\mathring{\varphi}^{-1}(\mathbf{c}_r)|} \right) \left( \frac{\mathbf{1}_{\mathring{\varphi}^{-1}(\mathbf{c}'_1)}(\mathbf{x}^{test})}{|\mathring{\varphi}^{-1}(\mathbf{c}'_1)|} \right)$$

Recalling the definition (29) of $\varrho_{\mathrm{SDM}}$, and letting $t+1 = 2R$, we see that the above lemma states that

$$\sum_{\omega_b \in \Omega_b} \varrho_{\mathrm{DM}}(\omega_a, \omega_b) = \varrho_{\mathrm{SDM}}(\omega_a) \tag{60}$$

and $\Omega_a = \Omega_{\mathrm{SDM}}$.

*Proof of Lemma 8.* We start by reorganizing the terms involved in the product defining $\varrho_{\mathrm{DM}}$ so that the variables in $\omega_a$ and $\omega_b$ are clearly separated:

$$\varrho_{\mathrm{DM}}(\omega) =$$
$$\frac{1}{|\Phi||\mathcal{C}|^{2R}} \left( \prod_{r=1}^{R} \frac{\mathbf{1}_{\mathring{\varphi}^{-1}(\mathbf{c}'_r)}(\mathbf{x}_{r,1})}{|\mathring{\varphi}^{-1}(\mathbf{c}'_r)|} \frac{\mathbf{1}_{\mathring{\varphi}^{-1}(\mathbf{c}_r)}(\mathbf{x}_{r,2})}{|\mathring{\varphi}^{-1}(\mathbf{c}_r)|} \right) \left( \frac{\mathbf{1}_{\mathring{\varphi}^{-1}(\mathbf{c}'_1)}(\mathbf{x}^{\mathrm{test}})}{|\mathring{\varphi}^{-1}(\mathbf{c}'_1)|} \right) \left( \prod_{r=1}^{R} \prod_{s=3}^{n_{\mathrm{spl}}} \frac{\mathbf{1}_{\mathring{\varphi}^{-1}(\mathbf{c}_r)}(\mathbf{x}_{r,s})}{|\mathring{\varphi}^{-1}(\mathbf{c}_r)|} \right)$$

To demonstrate the process, let us start by summing the above formula over the first variable of $\omega_b$, namely $\mathbf{x}_{1,3}$. Since this variable only occurs in the last term of the above product, we have:

$$\sum_{\mathbf{x}_{1,3} \in \mathcal{X}} \varrho_{\mathrm{DM}}(\omega) = \frac{1}{|\Phi||\mathcal{C}|^{2R}} \left( \prod_{r=1}^{R} \frac{\mathbf{1}_{\mathring{\varphi}^{-1}(\mathbf{c}'_r)}(\mathbf{x}_{r,1})}{|\mathring{\varphi}^{-1}(\mathbf{c}'_r)|} \frac{\mathbf{1}_{\mathring{\varphi}^{-1}(\mathbf{c}_r)}(\mathbf{x}_{r,2})}{|\mathring{\varphi}^{-1}(\mathbf{c}_r)|} \right) \left( \frac{\mathbf{1}_{\mathring{\varphi}^{-1}(\mathbf{c}'_1)}(\mathbf{x}^{\mathrm{test}})}{|\mathring{\varphi}^{-1}(\mathbf{c}'_1)|} \right)$$
$$\left( \prod_{\substack{1 \le r \le R \\ 3 \le s \le n_{\mathrm{spl}} \\ (r,s) \ne (1,3)}} \frac{\mathbf{1}_{\mathring{\varphi}^{-1}(\mathbf{c}_r)}(\mathbf{x}_{r,s})}{|\mathring{\varphi}^{-1}(\mathbf{c}_r)|} \right) \left( \sum_{\mathbf{x}_{1,3} \in \mathcal{X}} \frac{\mathbf{1}_{\mathring{\varphi}^{-1}(\mathbf{c}_1)}(\mathbf{x}_{1,3})}{|\mathring{\varphi}^{-1}(\mathbf{c}_1)|} \right)$$

Since $\sum_{\mathbf{x} \in \mathcal{X}} \mathbf{1}_{\mathring{\varphi}^{-1}(\mathbf{c}_1)}(\mathbf{x}) = |\mathring{\varphi}^{-1}(\mathbf{c}_1)|$, the last term of the above product is equal to 1 and can therefore be omitted. Repeating this process for all the $\mathbf{x}_{r,s}$ that constitute $\omega_b$ leads to the desired result. □

In the next subsection we will need the marginal of $\varrho_{\mathrm{DM}}$ with respect to another set of variables. To this aim we write $\omega = (\omega_c, \omega_d)$ where

$$\omega_c = (\varphi \quad ; \quad \mathbf{x}_{1,2}, \mathbf{x}_{1,3}, \ldots, \mathbf{x}_{1,n_{\mathrm{spl}}} \quad ; \quad \cdots \quad ; \quad \mathbf{x}_{R,2}, \mathbf{x}_{R,3}, , \ldots, \mathbf{x}_{R,n_{\mathrm{spl}}} \quad ; \quad \mathbf{x}^{\mathrm{test}}) \tag{61}$$

$$\omega_d = (\mathbf{c}_1, \ldots, \mathbf{c}_R \quad ; \quad \mathbf{c}'_1, \ldots, \mathbf{c}'_R \quad ; \quad \mathbf{x}_{1,1} \quad ; \quad \cdots \quad ; \quad \mathbf{x}_{R,1}) \tag{62}$$

Note that all the unfamiliar training points are contained in $\omega_d$. The test point and the familiar training points are in $\omega_c$. We also let $\Omega_c = \Phi \times \mathcal{X}^{R(n_{\mathrm{spl}}-1)+1}$ and $\Omega_d = \mathcal{C}^{2R} \times \mathcal{X}^R$.

**Lemma 9.** *For all $\omega_c \in \Omega_c$ we have*

$$\sum_{\omega_d \in \Omega_d} \varrho_{\mathrm{DM}}(\omega_c, \omega_d) = \frac{1}{|\Phi||\mathcal{X}|^{R+1} s_c^{LR(n_{spl}-2)}} \prod_{r=1}^{R} \mathbf{1}_{\{\mathring{\varphi}(\mathbf{x}_{r,2})=\mathring{\varphi}(\mathbf{x}_{r,3})=\ldots=\mathring{\varphi}(\mathbf{x}_{r,n_{spl}})\}}$$

*Proof.* We reorganizing the terms involved in the product defining $\varrho_{\mathrm{DM}}$ so that the variables in $\omega_c$ and $\omega_d$ are separated:

$$\varrho_{\mathrm{DM}}(\omega) = \frac{1}{|\Phi||\mathcal{C}|^{2R}} \left( \prod_{r=1}^{R} \prod_{s=2}^{n_{spl}} \frac{\mathbf{1}_{\mathring{\varphi}^{-1}(\mathbf{c}_r)}(\mathbf{x}_{r,s})}{|\mathring{\varphi}^{-1}(\mathbf{c}_r)|} \right) \left( \frac{\mathbf{1}_{\mathring{\varphi}^{-1}(\mathbf{c}_1')}(\mathbf{x}^{\mathrm{test}})}{|\mathring{\varphi}^{-1}(\mathbf{c}_1')|} \right) \left( \prod_{r=1}^{R} \frac{\mathbf{1}_{\mathring{\varphi}^{-1}(\mathbf{c}_r')}(\mathbf{x}_{r,1})}{|\mathring{\varphi}^{-1}(\mathbf{c}_r')|} \right)$$

Summing the above formula over the last variable involved in $\omega_d$, namely $\mathbf{x}_{R,1}$, gives

$$\sum_{\mathbf{x}_{R,1} \in \mathcal{X}} \varrho_{\mathrm{DM}}(\omega) = \frac{1}{|\Phi||\mathcal{C}|^{2R}} \left( \prod_{r=1}^{R} \prod_{s=2}^{n_{spl}} \frac{\mathbf{1}_{\mathring{\varphi}^{-1}(\mathbf{c}_r)}(\mathbf{x}_{r,s})}{|\mathring{\varphi}^{-1}(\mathbf{c}_r)|} \right) \left( \frac{\mathbf{1}_{\mathring{\varphi}^{-1}(\mathbf{c}_1')}(\mathbf{x}^{\mathrm{test}})}{|\mathring{\varphi}^{-1}(\mathbf{c}_1')|} \right)$$
$$\left( \prod_{r=1}^{R-1} \frac{\mathbf{1}_{\mathring{\varphi}^{-1}(\mathbf{c}_r')}(\mathbf{x}_{r,1})}{|\mathring{\varphi}^{-1}(\mathbf{c}_r')|} \right) \left( \sum_{\mathbf{x}_{R,1} \in \mathcal{X}} \frac{\mathbf{1}_{\mathring{\varphi}^{-1}(\mathbf{c}_R')}(\mathbf{x}_{R,1})}{|\mathring{\varphi}^{-1}(\mathbf{c}_R')|} \right)$$

The last term in the above product is equal to 1 and can therefore be omitted. Iterating this process gives

$$\sum_{\mathbf{x}_{1,1} \in \mathcal{X}} \cdots \sum_{\mathbf{x}_{R,1} \in \mathcal{X}} \varrho_{\mathrm{DM}}(\omega) = \frac{1}{|\Phi||\mathcal{C}|^{2R}} \left( \prod_{r=1}^{R} \prod_{s=2}^{n_{spl}} \frac{\mathbf{1}_{\mathring{\varphi}^{-1}(\mathbf{c}_r)}(\mathbf{x}_{r,s})}{|\mathring{\varphi}^{-1}(\mathbf{c}_r)|} \right) \left( \frac{\mathbf{1}_{\mathring{\varphi}^{-1}(\mathbf{c}_1')}(\mathbf{x}^{\mathrm{test}})}{|\mathring{\varphi}^{-1}(\mathbf{c}_1')|} \right)$$

We then use the fact that $\sum_{c_1' \in \mathcal{C}} \mathbf{1}_{\mathring{\varphi}^{-1}(\mathbf{c}_1')}(\mathbf{x}^{\mathrm{test}}) = 1$, see (38), together with $\left|\mathring{\varphi}^{-1}(\mathbf{c}_1')\right| = s_c^L$, see (36), to obtain

$$\sum_{c_1' \in \mathcal{C}} \sum_{\mathbf{x}_{1,1} \in \mathcal{X}} \cdots \sum_{\mathbf{x}_{R,1} \in \mathcal{X}} \varrho_{\mathrm{DM}}(\omega) = \frac{1}{|\Phi||\mathcal{C}|^{2R} s_c^L} \left( \prod_{r=1}^{R} \prod_{s=2}^{n_{spl}} \frac{\mathbf{1}_{\mathring{\varphi}^{-1}(\mathbf{c}_r)}(\mathbf{x}_{r,s})}{|\mathring{\varphi}^{-1}(\mathbf{c}_r)|} \right)$$

We then sum over $\mathbf{c}_2', \ldots, \mathbf{c}_R'$. Since these variables are not involved in the above formula we get

$$\sum_{c_1' \in \mathcal{C}} \cdots \sum_{c_R' \in \mathcal{C}} \sum_{\mathbf{x}_{1,1} \in \mathcal{X}} \cdots \sum_{\mathbf{x}_{R,1} \in \mathcal{X}} \varrho_{\mathrm{DM}}(\omega) = \frac{1}{|\Phi||\mathcal{C}|^{R+1} s_c^L} \left( \prod_{r=1}^{R} \prod_{s=2}^{n_{spl}} \frac{\mathbf{1}_{\mathring{\varphi}^{-1}(\mathbf{c}_r)}(\mathbf{x}_{r,s})}{|\mathring{\varphi}^{-1}(\mathbf{c}_r)|} \right)$$
$$= \frac{1}{|\Phi||\mathcal{C}|^{R}|\mathcal{X}|} \left( \prod_{r=1}^{R} \prod_{s=2}^{n_{spl}} \frac{\mathbf{1}_{\mathring{\varphi}^{-1}(\mathbf{c}_r)}(\mathbf{x}_{r,s})}{|\mathring{\varphi}^{-1}(\mathbf{c}_r)|} \right)$$

where we have used $|\mathcal{C}| s_c^L = n_c^L s_c^L = |\mathcal{X}|$ to obtain the last equality. Summing over $\mathbf{c}_1$ gives

$$\sum_{\mathbf{c}_1 \in \mathcal{C}} \sum_{c_1' \in \mathcal{C}} \cdots \sum_{c_R' \in \mathcal{C}} \sum_{\mathbf{x}_{1,1}} \cdots \sum_{\mathbf{x}_{R,1} \in \mathcal{X}} \varrho_{\mathrm{DM}}(\omega) = \frac{1}{|\Phi||\mathcal{C}|^{R}|\mathcal{X}|} \sum_{\mathbf{c}_1 \in \mathcal{C}} \left( \prod_{r=1}^{R} \prod_{s=2}^{n_{spl}} \frac{\mathbf{1}_{\mathring{\varphi}^{-1}(\mathbf{c}_r)}(\mathbf{x}_{r,s})}{|\mathring{\varphi}^{-1}(\mathbf{c}_r)|} \right)$$
$$= \frac{1}{|\Phi||\mathcal{C}|^{R}|\mathcal{X}|} \left( \prod_{r=2}^{R} \prod_{s=2}^{n_{spl}} \frac{\mathbf{1}_{\mathring{\varphi}^{-1}(\mathbf{c}_r)}(\mathbf{x}_{r,s})}{|\mathring{\varphi}^{-1}(\mathbf{c}_r)|} \right) \left( \sum_{\mathbf{c}_1 \in \mathcal{C}} \prod_{s=2}^{n_{spl}} \frac{\mathbf{1}_{\mathring{\varphi}^{-1}(\mathbf{c}_1)}(\mathbf{x}_{1,s})}{|\mathring{\varphi}^{-1}(\mathbf{c}_1)|} \right)$$
$$= \frac{1}{|\Phi||\mathcal{C}|^{R}|\mathcal{X}|} \left( \prod_{r=2}^{R} \prod_{s=2}^{n_{spl}} \frac{\mathbf{1}_{\mathring{\varphi}^{-1}(\mathbf{c}_r)}(\mathbf{x}_{r,s})}{|\mathring{\varphi}^{-1}(\mathbf{c}_r)|} \right) \left( \frac{\mathbf{1}_{\{\mathring{\varphi}(\mathbf{x}_{1,2})=\mathring{\varphi}(\mathbf{x}_{1,3})=\ldots=\mathring{\varphi}(\mathbf{x}_{1,n_{spl}})\}}}{|\mathring{\varphi}^{-1}(\mathbf{c}_1)|^{n_{spl}-1}} \right)$$

To obtain the last equality we have used (39) but for a product of $n_{spl} - 1$ indicator functions instead of just two. Iterating this process we obtain

$$\sum_{\mathbf{c}_1 \in \mathcal{C}} \cdots \sum_{\mathbf{c}_R \in \mathcal{C}} \sum_{c_1' \in \mathcal{C}} \cdots \sum_{c_R' \in \mathcal{C}} \sum_{\mathbf{x}_{1,1} \in \mathcal{X}} \cdots \sum_{\mathbf{x}_{R,1} \in \mathcal{X}} \varrho_{\mathrm{DM}}(\omega) = \frac{1}{|\Phi||\mathcal{C}|^{R}|\mathcal{X}|} \prod_{r=1}^{R} \frac{\mathbf{1}_{\{\mathring{\varphi}(\mathbf{x}_{r,2})=\mathring{\varphi}(\mathbf{x}_{1,3})=\ldots=\mathring{\varphi}(\mathbf{x}_{r,n_{spl}})\}}}{|\mathring{\varphi}^{-1}(\mathbf{c}_r)|^{n_{spl}-1}}$$

Using one more time that $\left|\mathring{\varphi}^{-1}(\mathbf{c}_r)\right| = s_c^L$ and $|\mathcal{C}| s_c^L = |\mathcal{X}|$ gives to the desired result. $\square$

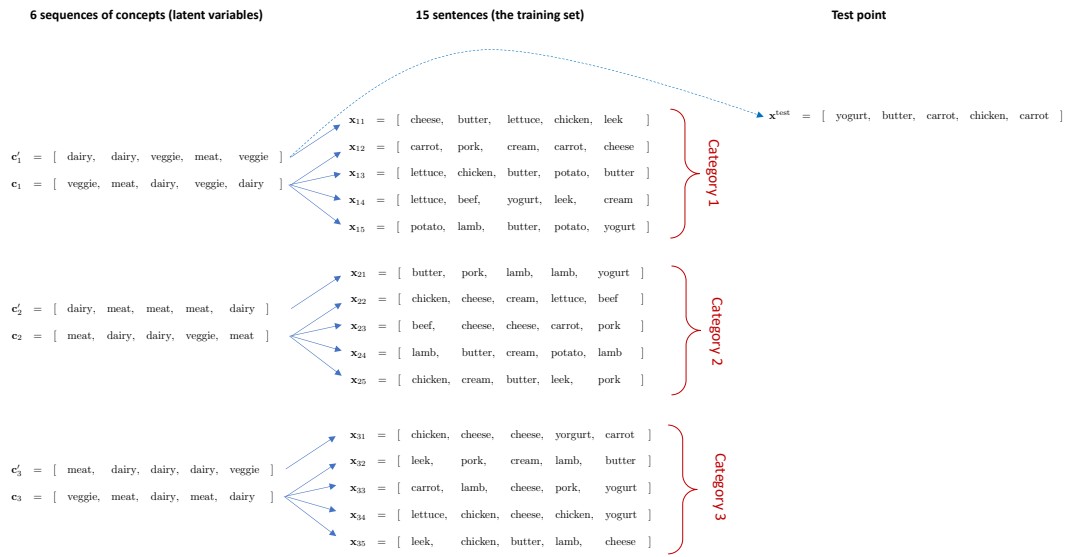

Figure 4: The test point $\mathbf{x}^{\text{test}}$ and the train point $\mathbf{x}_{1,1}$ are generated by the same sequence of concepts.

### B.3 CONCLUSION OF PROOF

We now establish the desired upper bound (28), which we restate below for convenience:

$$\sup_{K \in \mathcal{K}} \mathbb{P}_{\text{DM}}\Big[E_K\Big] \leq \frac{1}{|\mathcal{X}|} \sum_{\mathbf{x} \in \mathcal{X}} \mathcal{H}_{2R-1}\left(K^\star(\mathbf{x}, \cdot)\right) + \frac{1}{R} \tag{63}$$

where

$$E_K = \Big\{\omega \in \Omega_{\text{DM}} : \quad \text{There exists } 1 \leq s^* \leq n_{\text{spl}} \text{ such that}$$

$$K\left(\mathbf{x}^{\text{test}}, \mathbf{x}_{1,s^*}\right) > K\left(\mathbf{x}^{\text{test}}, \mathbf{x}_{r,s}\right) \text{ for all } 2 \leq r \leq R \text{ and all } 1 \leq s \leq n_{\text{spl}}\Big\} \tag{64}$$

We recall that the test point $\mathbf{x}^{\text{test}}$ is generated by the unfamiliar sequence of concepts $\mathbf{c}_1'$ and that it belongs to category 1, see Figure 4. The event $E_K$ describes all the outcomes in which the training point most similar to $\mathbf{x}^{\text{test}}$ (where similarity is measured with respect to the kernel $K$) belongs to the first category. There are two very distinct cases within the event $E_K$: the training point most similar to $\mathbf{x}^{\text{test}}$ can be $\mathbf{x}_{1,1}$ — this corresponds to a 'meaningful success' in which the learner recognizes that $\mathbf{x}_{1,1}$ is generated by the same sequence of concepts than $\mathbf{x}^{\text{test}}$, see Figure 4. Or the training point most similar to $\mathbf{x}^{\text{test}}$ can be one of the points $\mathbf{x}_{1,2}, \ldots, \mathbf{x}_{1,n_{\text{spl}}}$ — this corresponds to a 'lucky success' because $\mathbf{x}_{1,2}, \ldots, \mathbf{x}_{1,n_{\text{spl}}}$ are not related to $\mathbf{x}^{\text{test}}$ (they are generated by a different sequence of concept, see Figure 4). To make this discussion formal, we fix a kernel $K \in \mathcal{K}$, and we partition the event $E_K$ as follow

$$E_K = E_{\text{meaningful}} \cup E_{\text{luck}} \tag{65}$$

where

$$E_{\text{meaningful}} = E_K \cap \Big\{\omega \in \Omega_{\text{DM}} : \quad K\left(\mathbf{x}^{\text{test}}, \mathbf{x}_{1,1}\right) > K\left(\mathbf{x}^{\text{test}}, \mathbf{x}_{1,s}\right) \text{ for all } 2 \leq s \leq n_{\text{spl}}\Big\}$$

$$E_{\text{luck}} = E_K \cap \Big\{\omega \in \Omega_{\text{DM}} : \quad K\left(\mathbf{x}^{\text{test}}, \mathbf{x}_{1,1}\right) \leq K\left(\mathbf{x}^{\text{test}}, \mathbf{x}_{1,s}\right) \text{ for some } 2 \leq s \leq n_{\text{spl}}\Big\}$$

The next two lemmas provide upper bounds for the probability of the events $E_{\text{meaningful}}$ and $E_{\text{luck}}$.

**Lemma 10.** $\mathbb{P}_{\text{DM}}[E_{meaningful}] \leq \dfrac{1}{|\mathcal{X}|} \displaystyle\sum_{\mathbf{x} \in \mathcal{X}} \mathcal{H}_{2R-1}(K_{\mathbf{x}}^\star).$

*Proof.* Define the event

$$A := \left\{ \omega \in \Omega_{\mathrm{DM}} : \quad K\left(\mathbf{x}^{\mathrm{test}}, \mathbf{x}_{1,1}\right) > K\left(\mathbf{x}^{\mathrm{test}}, \mathbf{x}_{r,1}\right) \text{ for all } 2 \le r \le R \right\}$$
$$\cap \left\{ \omega \in \Omega_{\mathrm{DM}} : \quad K\left(\mathbf{x}^{\mathrm{test}}, \mathbf{x}_{1,1}\right) > K\left(\mathbf{x}^{\mathrm{test}}, \mathbf{x}_{r,2}\right) \text{ for all } 1 \le r \le R \right\}.$$

This events involves only the first two training points of each category. On the example depicted on Figure 4, that would be the points $\mathbf{x}_{1,1}$ and $\mathbf{x}_{1,2}$, the points $\mathbf{x}_{2,1}$ and $\mathbf{x}_{2,2}$, and finally the points $\mathbf{x}_{3,1}$ and $\mathbf{x}_{3,2}$. The event $A$ consists in all the outcomes in which, among these $2R$ training points, $\mathbf{x}_{1,1}$ is most similar to $\mathbf{x}^{\mathrm{test}}$. We then make two key remarks. First, these $2R$ points are generated by $2R$ distinct sequences on concepts — so if we restrict our attention to these $2R$ points, we are in a situation very similar to the simpler data model $\varrho_{SDM}$ (i.e. we first generate $2R$ sequences of concepts, then from each sequence of concepts we generate a single training point, and finally we generate a test point from the first sequence of concepts.) We will make this intuition precise by appealing to the fact that $\varrho_{SDM}$ is the marginal of $\varrho_{DM}$, and this will allow us to obtain a bound for $\mathbb{P}_{\mathrm{DM}}[A]$ in term of the permuted moment of $K^{\star}$. The second remark is that $E_{\mathrm{meaningful}}$ is clearly contained in $A$, and therefore we have

$$\mathbb{P}_{\mathrm{DM}}[E_{\mathrm{meaningful}}] \le \mathbb{P}_{\mathrm{DM}}[A] \tag{66}$$

so an upper bound for $\mathbb{P}_{\mathrm{DM}}[A]$ is also an upper bound for $\mathbb{P}_{\mathrm{DM}}[E_{\mathrm{meaningful}}]$.

Let us rename some of the variables. We define $\mathbf{d}_1, \ldots, \mathbf{d}_{2R}$, and $\mathbf{y}_1, \ldots, \mathbf{y}_{2R}$ as follow:

$$\mathbf{d}_{2r-1} = \mathbf{c}'_r \quad \text{and} \quad \mathbf{d}_{2r} = \mathbf{c}_r \qquad \text{for } r = 1, \ldots, R$$
$$\mathbf{y}_{2r-1} = \mathbf{x}_{r,1} \quad \text{and} \quad \mathbf{y}_{2r} = \mathbf{x}_{r,2} \qquad \text{for } r = 1, \ldots, R$$

On the example depicted on Figure 4, that would be:

$$\begin{array}{cccccc} \mathbf{y}_1 = \mathbf{x}_{1,1}, & \mathbf{y}_2 = \mathbf{x}_{1,2}, & \mathbf{y}_3 = \mathbf{x}_{2,1}, & \mathbf{y}_4 = \mathbf{x}_{2,2}, & \mathbf{y}_5 = \mathbf{x}_{3,1}, & \mathbf{y}_6 = \mathbf{x}_{3,2} \\ \mathbf{d}_1 = \mathbf{c}_1, & \mathbf{d}_2 = \mathbf{c}'_1, & \mathbf{d}_3 = \mathbf{c}_2, & \mathbf{d}_4 = \mathbf{c}'_2, & \mathbf{d}_5 = \mathbf{c}_3, & \mathbf{d}_6 = \mathbf{c}'_3 \end{array}$$

In other words, the $\mathbf{y}_r$'s are the first two training points of each category and the $\mathbf{d}_r$'s are their corresponding sequence of concepts. With these notations it is clear that the training points $\mathbf{y}_r$ are generated by distinct sequences of concepts, and that the test point $\mathbf{x}^{\mathrm{test}}$ is generated by the same sequence of concepts than $\mathbf{y}_1$. Moreover the event $A$ can now be conveniently written as

$$A = \{\omega \in \Omega_{\mathrm{DM}} : \quad K\left(\mathbf{x}^{\mathrm{test}}, \mathbf{y}_1\right) > K\left(\mathbf{x}^{\mathrm{test}}, \mathbf{y}_r\right) \text{ for all } 2 \le r \le 2R\}.$$

Let $h : \mathcal{X}^{2R+1} \to \mathbb{R}$ be the indicator function defined by

$$h(\mathbf{y}_1, \ldots, \mathbf{y}_{2R}, \mathbf{x}^{\mathrm{test}}) = \begin{cases} 1 & \text{if } K\left(\mathbf{x}^{\mathrm{test}}, \mathbf{y}_1\right) > K\left(\mathbf{x}^{\mathrm{test}}, \mathbf{x}_r\right) \text{ for all } 2 \le r \le 2R \\ 0 & \text{otherwise} \end{cases}$$

We now recall the splitting $\omega = (\omega_a, \omega_b)$ described in (58)-(59) and note that $\omega_a$ can be written as

$$\omega_a = (\varphi \ ; \ \mathbf{d}_1, \ldots, \mathbf{d}_{2R} \ ; \ \mathbf{y}_1, \ldots, \mathbf{y}_{2R} \ ; \ \mathbf{x}^{\mathrm{test}})$$

Since $h$ only depends on the variables involved in $\omega_a$, and since, according to Lemma 8, $\sum_{\omega_b} \varrho_{\mathrm{DM}}(\omega_a, \omega_b) = \varrho_{\mathrm{SDM}}(\omega_a)$, we obtain

$$\mathbb{P}_{\mathrm{DM}}[A] = \sum_{\omega_a \in \Omega_a} \sum_{\omega_b \in \Omega_b} h(\mathbf{y}_1, \ldots, \mathbf{y}_{2R}, \mathbf{x}^{\mathrm{test}}) \, \varrho_{\mathrm{DM}}(\omega_a, \omega_b)$$
$$= \sum_{\omega_a \in \Omega_a} h(\mathbf{y}_1, \ldots, \mathbf{y}_{2R}, \mathbf{x}^{\mathrm{test}}) \, \varrho_{\mathrm{SDM}}(\omega_a)$$
$$= \mathbb{P}_{\mathrm{SDM}}[\omega_a \in \Omega_a : \quad K\left(\mathbf{x}^{\mathrm{test}}, \mathbf{y}_1\right) > K\left(\mathbf{x}^{\mathrm{test}}, \mathbf{y}_r\right) \text{ for all } 2 \le r \le 2R]$$
$$\le \frac{1}{|\mathcal{X}|} \sum_{\mathbf{x} \in \mathcal{X}} \mathcal{H}_{2R-1}(K^{\star}_{\mathbf{x}})$$

where we have used Theorem 4 in order to get the last inequality (with the understanding that $t + 1 = 2R$.) Combining the above bound with (66) concludes the proof. $\qquad \square$

We now estimate the probability of the event $E_{\text{luck}}$.

**Lemma 11.** $\mathbb{P}_{\text{DM}}[E_{luck}] \leq \dfrac{1}{R}$.

*Proof.* For $1 \leq r \leq R$, we define the events

$$B_r = \bigcap_{\substack{1 \leq r' \leq R \\ r' \neq r}} \left\{ \omega \in \Omega_{\text{DM}} : \max_{2 \leq s \leq n_{\text{spl}}} K(\mathbf{x}^{\text{test}}, \mathbf{x}_{r,s}) > \max_{2 \leq s' \leq n_{\text{spl}}} K(\mathbf{x}^{\text{test}}, \mathbf{x}_{r',s'}) \right\}$$

Note that the events $B_r$ involve only the training points with an index $s \geq 2$: these are the familiar training points. On the example depicted on Figure 4, these are the training points generated by $\mathbf{c}_1, \mathbf{c}_2$ and $\mathbf{c}_3$. Let us pursue with this example. The event $B_1$ consists in all the outcomes in which one of the points generated by $\mathbf{c}_1$ is more similar to $\mathbf{x}^{\text{test}}$ than any of the points generated by $\mathbf{c}_2$ and $\mathbf{c}_3$. The event $B_2$ consists in all the outcomes in which one of the points generated by $\mathbf{c}_2$ is more similar to $\mathbf{x}^{\text{test}}$ than any of the points generated by $\mathbf{c}_1$ and $\mathbf{c}_3$. And finally the event $B_3$ consists in all the outcomes in which one of the points generated by $\mathbf{c}_3$ is more similar to $\mathbf{x}^{\text{test}}$ than any of the points generated by $\mathbf{c}_1$ and $\mathbf{c}_2$. Importantly, the test point $\mathbf{x}^{\text{test}}$ is generated by the unfamiliar sequence of concepts $\mathbf{c}'_1$, and this sequence of concept is unrelated to the sequence $\mathbf{c}_1, \mathbf{c}_2$ and $\mathbf{c}_3$. So one would expect that, from simple symmetry, that

$$\mathbb{P}_{\text{DM}}[B_1] = \mathbb{P}_{\text{DM}}[B_2] = \mathbb{P}_{\text{DM}}[B_3]. \tag{67}$$

We will prove (67) rigorously, for general $R$, using Lemma 9 from the previous subsection. But before to do so, let us show that (67) implies the desired upperbound on the probability of $E_{\text{luck}}$. First, note that $E_{\text{luck}} \subset B_1$ and therefore

$$\mathbb{P}_{\text{DM}}[E_{\text{luck}}] \leq \mathbb{P}_{\text{DM}}[B_1]. \tag{68}$$

Then, note that $B_1$, $B_2$ and $B_3$ are mutually disjoints, and therefore, continuing with the same example,

$$\mathbb{P}_{\text{DM}}[B_1] + \mathbb{P}_{\text{DM}}[B_2] + \mathbb{P}_{\text{DM}}[B_3] = \mathbb{P}_{\text{DM}}[B_1 \cup B_2 \cup B_3] \leq 1$$

which, combined with (67) and (68), gives $\mathbb{P}_{\text{DM}}[E_{\text{luck}}] \leq 1/3$ as desired.

We now provide a formal proof of (67). As in the proof of the previous lemma, it is convenient to rename some of the variables. Let denote by $\mathbf{fam}_r$ the variable that consists in the familiar training points from category $r$:

$$\mathbf{fam}_r = (\mathbf{x}_{r,2}, \ldots, \mathbf{x}_{r,n_{\text{spl}}}) \in \mathcal{X}^{n_{\text{spl}}-1}$$

With this notation we have $\mathbf{fam}_{r,s} = \mathbf{x}_{r,s+1}$. We now recall the splitting $\omega = (\omega_c, \omega_d)$ described in (61)-(62), and note that $\omega_c$ can be written as

$$\omega_c = (\varphi; \ \mathbf{fam}_1 \ ; \ \ldots \ ; \ \mathbf{fam}_R \ ; \ \mathbf{x}^{\text{test}}). \tag{69}$$

Using Lemma 9 we have

$$\sum_{\omega_d \in \Omega_d} \varrho_{\text{DM}}(\omega_c, \omega_d) = \alpha \prod_{r=1}^{R} \mathbf{1}_{\{\check{\varphi}(\mathbf{x}_{r,2}) = \check{\varphi}(\mathbf{x}_{r,3}) = \ldots = \check{\varphi}(\mathbf{x}_{r,n_{\text{spl}}})\}} \tag{70}$$

$$= \alpha \prod_{r=1}^{R} \mathbf{1}_{\{\check{\varphi}(\mathbf{fam}_{r,1}) = \check{\varphi}(\mathbf{fam}_{r,2}) = \ldots = \check{\varphi}(\mathbf{fam}_{r,n_{\text{spl}}-1})\}} \tag{71}$$

$$= \alpha \prod_{r=1}^{R} h(\varphi, \mathbf{fam}_r) \tag{72}$$

where $\alpha$ is the constant appearing in front of the product in Lemma 9 an $h : \Phi \times \mathcal{X}^{n_{\text{spl}}-1} \to \{0, 1\}$ is the indicator function implicitly defined in equality (71)-(72). With the slight abuse of notation of viewing $\mathbf{fam}_r$ as a set instead of a tuple, let us rewrite the event $B_r$ as

$$B_r = \bigcap_{\substack{1 \leq r' \leq R \\ r' \neq r}} \left\{ \omega \in \Omega_{\text{DM}} : \max_{\mathbf{x} \in \mathbf{fam}_r} K(\mathbf{x}^{\text{test}}, \mathbf{x}) > \max_{\mathbf{x} \in \mathbf{fam}_{r'}} K(\mathbf{x}^{\text{test}}, \mathbf{x}) \right\}$$

We also define the corresponding indicator function

$$
g(\mathbf{fam}_r, \mathbf{fam}_{r'}, \mathbf{x}^{\text{test}}) = \begin{cases} 1 & \text{if } \max_{\mathbf{x} \in \mathbf{fam}_r} K(\mathbf{x}^{\text{test}}, \mathbf{x}) > \max_{\mathbf{x} \in \mathbf{fam}_{r'}} K(\mathbf{x}^{\text{test}}, \mathbf{x}) \\ \\ 0 & \text{otherwise} \end{cases}
$$

We now compute $\mathbb{P}_{\text{DM}}(B_1)$ (the formula for the other $\mathbb{P}_{\text{DM}}(B_r)$ are obtained in a similar manner.) Recall from (69) that the variables involved in $\mathbf{fam}_r$ only appear in $\omega_c$. Therefore

$$
\mathbb{P}_{\text{DM}}[B_1] = \sum_{\omega_c \in \Omega_c} \sum_{\omega_d \in \Omega_d} \left( \prod_{r=2}^{R} g(\mathbf{fam}_1, \mathbf{fam}_r, \mathbf{x}^{\text{test}}) \right) \varrho_{\text{DM}}(\omega_c, \omega_d) \tag{73}
$$

$$
= \sum_{\omega_c \in \Omega_c} \left( \prod_{r=2}^{R} g(\mathbf{fam}_1, \mathbf{fam}_r, \mathbf{x}^{\text{test}}) \right) \sum_{\omega_d \in \Omega_d} \varrho_{\text{DM}}(\omega_c, \omega_d) \tag{74}
$$

$$
= \alpha \sum_{\omega_c \in \Omega_c} \left( \prod_{r=2}^{R} g(\mathbf{fam}_1, \mathbf{fam}_r, \mathbf{x}^{\text{test}}) \right) \left( \prod_{r'=1}^{R} h(\varphi, \mathbf{fam}_{r'}) \right) \tag{75}
$$

where we have used (72) to obtain the last equality. Let us now compare $\mathbb{P}_{\text{DM}}(B_1)$ and $\mathbb{P}_{\text{DM}}(B_2)$. Letting $\mathcal{Z} := \mathcal{X}^{n_{\text{spl}}-1}$, and recalling that $\omega_c = (\varphi; \ \mathbf{fam}_1 \ , \ \dots \ , \ \mathbf{fam}_R \ ; \ \mathbf{x}^{\text{test}})$, we obtain:

$$
\mathbb{P}_{\text{DM}}[B_1] = \sum_{\varphi \in \Phi} \sum_{\mathbf{fam}_1 \in \mathcal{Z}} \cdots \sum_{\mathbf{fam}_R \in \mathcal{Z}} \sum_{\mathbf{x}^{\text{test}} \in \mathcal{X}} \left( \prod_{\substack{1 \le r \le R \\ r \ne 1}} g(\mathbf{fam}_1, \mathbf{fam}_r, \mathbf{x}^{\text{test}}) \right) \left( \prod_{r'=1}^{R} h(\phi, \mathbf{fam}'_r) \right)
$$

$$
\mathbb{P}_{\text{DM}}[B_2] = \sum_{\varphi \in \Phi} \sum_{\mathbf{fam}_1 \in \mathcal{Z}} \cdots \sum_{\mathbf{fam}_R \in \mathcal{Z}} \sum_{\mathbf{x}^{\text{test}} \in \mathcal{X}} \left( \prod_{\substack{1 \le r \le R \\ r \ne 2}} g(\mathbf{fam}_2, \mathbf{fam}_r, \mathbf{x}^{\text{test}}) \right) \left( \prod_{r'=1}^{R} h(\phi, \mathbf{fam}'_r) \right)
$$

From the above it is clear that $\mathbb{P}_{\text{DM}}[B_1] = \mathbb{P}_{\text{DM}}[B_2]$ (as can be seen by exchanging the name of the variables $\mathbf{fam}_1$ and $\mathbf{fam}_2$). Similar reasoning shows that the events $B_r$ are all equiprobable, which concludes the proof. $\square$

Combining Lemma 10 and 11, together with the fact that $E_K = E_{\text{correct}} \cup E_{\text{luck}}$, concludes the proof of (63). Inequality (63) implies inequality (26), which itself is equivalent to inequality (14).

## C  Upper bound for the Permuted Moment of $K^\star$

This section is devoted to the proof of inequality (15), which we state below as a theorem for convenience.

**Theorem 5.** *For all $0 \le \ell \le L$, we have the upper bound*

$$
\frac{1}{|\mathcal{X}|} \sum_{\mathbf{x} \in \mathcal{X}} \mathcal{H}_t(K^\star_{\mathbf{x}}) \le \left( 1 - \sum_{\mathbf{k} \in \mathcal{S}_\ell} \mathfrak{f}(\mathbf{k}) \mathfrak{g}(\mathbf{k}) \right) + \frac{1}{t+1} \left( \max_{\mathbf{k} \in \mathcal{S}_\ell} \mathfrak{f}(\mathbf{k}) \right). \tag{76}
$$

The rather intricate formula for the function $\mathfrak{f}$ and $\mathfrak{g}$ can be found in the main body of the paper, but we will recall them as we go through the proof.

We also recall that the optimal kernel is given by the formula:

$$
K^\star(\mathbf{x}, \mathbf{y}) = \frac{1}{s_c^L} \frac{\left| \{ \varphi \in \Phi : \varphi(x_\ell) = \varphi(y_\ell) \text{ for all } 1 \le \ell \le L \} \right|}{|\Phi|} \tag{77}
$$

The key insight to derive the upper bound (76) is to note that each pair of sentences $(\mathbf{x}, \mathbf{y})$ induces a graph on the vocabulary $\{1, 2, \dots, n_w\}$, and that the quantity

$$
\left| \{ \varphi \in \Phi : \varphi(x_\ell) = \varphi(y_\ell) \text{ for all } 1 \le \ell \le L \} \right|
$$

Table 2: First five rows of the Strirling triangle for the Stirling numbers $\left\{ {n \atop k} \right\}$.

| $n \backslash k$ | 1 | 2 | 3 | 4 | 5 |
|---|---|---|---|---|---|
| 1 | 1 | | | | |
| 2 | 1 | 1 | | | |
| 3 | 1 | 3 | 1 | | |
| 4 | 1 | 7 | 6 | 1 | |
| 5 | 1 | 15 | 25 | 10 | 1 |

can be interpreted as the number of equipartitions of the vocabulary that do not sever any of the edges of the graph. This graph-cut interpretation of the optimal kernel is presented in detail in Subsection C.1. In Subsection C.2 we derive a formula for $K^\star$ which is more tractable than (77). To do this we partition $\mathcal{X} \times \mathcal{X}$ into subsets on which $K^\star$ is constant, then provide a formula for the value of $K^\star$ on each of these subsets (c.f. Lemma 16). With this formula at hand, we then appeal to Lemma 3 to derive a first bound for the permuted moment of $K^\star$ (c.f. Lemma 17). This first bound is not fully explicit because it involves the size of the subsets on which $K^\star$ is constant. In section C.3 we appeal to Cayley's formula, a classical result from graph theory, to estimate the size of these subsets (c.f. Lemma 18) and therefore conclude the proof of Theorem 5.

We now introduce the combinatorial notations that will be used in this section, and we recall a few basics combinatorial facts. We denote by $\mathbb{N} = \{0, 1, 2, \ldots\}$ the nonnegative integers. We use the standard notations

$$\binom{n}{k} := \frac{n!}{k!(n-k)!} \qquad \text{and} \qquad \binom{n}{k_1, k_2, \ldots, k_m} := \frac{n!}{k_1! k_2! \cdots k_n!}$$

for the binomial and multinomial coefficients, with the understanding that $0! = 1$. We recall that multinomial coefficients can be interpreted as the number of ways of placing $n$ distinct objects into $m$ distinct bins, with the constraint that $k_1$ objects must go in the first bin, $k_2$ objects must go in the second bin, and so forth.

The Stirling numbers of the second kind $\left\{ {n \atop k} \right\}$ are close relatives of the binomial coefficients. $\left\{ {n \atop k} \right\}$ stands for the number of ways to partition a set of $n$ objects into $k$ nonempty subsets. To give a simple example, $\left\{ {4 \atop 2} \right\} = 7$ because there are 7 ways to partition the set $\{1, 2, 3, 4\}$ into two non-empty subsets, namely:

$$\{1\} \cup \{2, 3, 4\}, \qquad \{2\} \cup \{1, 3, 4\}, \qquad \{3\} \cup \{1, 2, 4\}, \qquad \{4\} \cup \{1, 2, 3\},$$
$$\{1, 2\} \cup \{3, 4\}, \qquad \{1, 3\} \cup \{2, 4\}, \qquad \{1, 4\} \cup \{3, 4\}.$$

Stirling numbers are easily computed via the following variant of Pascal's recurrence formula [4]:

$$\left\{ {n \atop 1} \right\} = 1, \qquad \left\{ {n \atop n} \right\} = 1 \qquad \text{for } n \geq 1,$$
$$\left\{ {n \atop k} \right\} = \left\{ {n-1 \atop k-1} \right\} + k \left\{ {n-1 \atop k} \right\} \qquad \text{for } 2 \leq k \leq n-1.$$

The above formula is easily derived from the definition of the Stirling numbers as providing the number of ways to partition a set of $n$ objects into $k$ nonempty subsets (see for example chapter 6 of **?**). Table 2 shows the first few Stirling numbers.

We recall that an undirected graph is an ordered pair $\mathcal{G} = (V, E)$, where $V$ is the vertex set and

$$E \subset \{\{v, v'\} : v, v' \in V \text{ and } v \neq v'\}$$

is the edge set. Edges are unordered pairs of distinct vertices (so loops are not allowed.) A tree is a connected graph with no cycles. A tree on $n$ vertices has exactly $n - 1$ edges. Cayley's formula states that there are $n^{n-2}$ ways to put $n - 1$ edges on $n$ labeled vertices in order to make a tree. We formally state this classical result below:

**Lemma 12** (Cayley's formula). *There are $n^{n-2}$ trees on $n$ labeled vertices.*

---

[4]Alternatively, Stirling numbers can be defined through the formula $\left\{ {n \atop k} \right\} = \frac{1}{k!} \sum_{i=0}^{k} (-1)^i \binom{k}{k-i} (k-i)^n$

C.1 GRAPH-CUT FORMULATION OF THE OPTIMAL KERNEL

In this section we consider undirected graphs on the vertex set

$$\mathcal{V} := \{1, 2, \ldots, n_w\}.$$

Since the data space $\mathcal{X}$ consists of sentences of length $L$, graphs that have at most $L$ edges will be of particular importance. We therefore define:

$$\mathfrak{G} := \{\text{All graphs on } \mathcal{V} \text{ that have at most } L \text{ edges}\}.$$

In other words, $\mathfrak{G}$ consists in all the graphs $\mathcal{G} = (\mathcal{V}, E)$ whose edge set $E$ has cardinality less or equal to $L$. Since these graphs all have the same vertex set, we will often identify them with their edge set. We now introduce a mapping between pairs of sentences containing $L$ words, and graphs containing at most $L$ edges.

**Definition 2.** *The function* $\zeta : \mathcal{X} \times \mathcal{X} \to \mathfrak{G}$ *is defined by*

$$\zeta(\mathbf{x}, \mathbf{y}) := \bigcup_{\substack{1 \leq \ell \leq L \\ x_\ell \neq y_\ell}} \{\{x_\ell, y_\ell\}\}. \tag{78}$$

The right hand side of (78) is a set of at most $L$ edges. Since graphs in $\mathfrak{G}$ are identified with their edge set, $\zeta$ indeed define a mapping from $\mathcal{X} \times \mathcal{X}$ to $\mathfrak{G}$. Let us give a few examples illustrating how the map $\zeta$ works. Suppose we have a vocabulary of $n_w = 10$ words and sentences of length $L = 6$. Consider the pair of sentences $(\mathbf{x}, \mathbf{y}) \in \mathcal{X} \times \mathcal{X}$ where

$$\begin{aligned} \mathbf{x} &= [\quad 2, \quad 2, \quad 8, \quad 5, \quad 9, \quad 7 \quad] \\ \mathbf{y} &= [\quad 2, \quad 5, \quad 8, \quad 2, \quad 2, \quad 1 \quad] \end{aligned} \tag{79}$$

Then $\zeta(\mathbf{x}, \mathbf{y})$ is the set of 3 edges

$$\zeta(\mathbf{x}, \mathbf{y}) = \Big\{ \quad \{2, 5\}, \quad \{9, 2\}, \quad \{7, 1\} \quad \Big\}.$$

which indeed define a graph on $\mathcal{V}$. Note that position $\ell = 2$ and $\ell = 4$ of $(\mathbf{x}, \mathbf{y})$ 'code' for the same edge $\{2, 5\}$, position 5 codes for the edge $\{9, 2\}$, and position 6 codes for the edge $\{7, 1\}$. On the other hand, position 1 and 3 do not code for any edge: indeed, since $x_1 = y_1$ and $x_3 = y_3$, these two positions do not contribute any edges to the edge set defined by (78). We will say that positions 1 and 3 are *silent*. We make this terminology formal in the definition below:

**Definition 3.** *Let* $(\mathbf{x}, \mathbf{y}) \in \mathcal{X} \times \mathcal{X}$. *If* $x_\ell = y_\ell$ *for some* $1 \leq \ell \leq L$, *we say that position* $\ell$ *of the pair* $(\mathbf{x}, \mathbf{y})$ *is silent. If* $x_\ell \neq y_\ell$ *for some* $1 \leq \ell \leq L$, *we say that position* $\ell$ *of the pair* $(\mathbf{x}, \mathbf{y})$ *codes for the edge* $\{x_\ell, y_\ell\}$.

Note that if $(\mathbf{x}, \mathbf{y})$ has some silent positions, or if multiple positions codes for the same edge, then the graph $\zeta(\mathbf{x}, \mathbf{y})$ will have strictly less than $L$ edges. On the other hand, if none of these take place, then $\zeta(\mathbf{x}, \mathbf{y})$ will have exactly $L$ edges. For example the pair of sentences

$$\begin{aligned} \mathbf{x} &= [\quad 1, \quad 1, \quad 1, \quad 5, \quad 6, \quad 7] \\ \mathbf{y} &= [\quad 2, \quad 3, \quad 4, \quad 6, \quad 7, \quad 1] \end{aligned} \tag{80}$$

does not have silent positions, and all positions code for different edges. The corresponding graph

$$\zeta(\mathbf{x}, \mathbf{y}) = \Big\{ \quad \{1, 2\}, \quad \{1, 3\}, \quad \{1, 4\}, \quad \{5, 6\}, \quad \{6, 7\}, \quad \{7, 1\} \quad \Big\}$$

has the maximal possible number of edges, namely $L = 6$ edges. From the above discussion, it is clear that any graph with $L$ or less edges can be expressed as $\zeta(\mathbf{x}, \mathbf{y})$ for some pair of sentences $(\mathbf{x}, \mathbf{y}) \in \mathcal{X} \times \mathcal{X}$. Therefore $\zeta : \mathcal{X} \times \mathcal{X} \to \mathfrak{G}$ is surjective. On the other hand, different pair of sentences can be mapped to the same graph. Therefore $\zeta$ is not injective. We now introduce the following function.

**Definition 4** (Number of cut-free equipartitions of a graph). *The function* $\mathcal{I} : \mathfrak{G} \to \mathbb{N}$ *is defined by :*

$$\mathcal{I}(\mathcal{G}) = |\{\varphi \in \Phi : \varphi(v) = \varphi(v') \text{ for all edge } \{v, v'\} \text{ of the graph } \mathcal{G}\}| \tag{81}$$

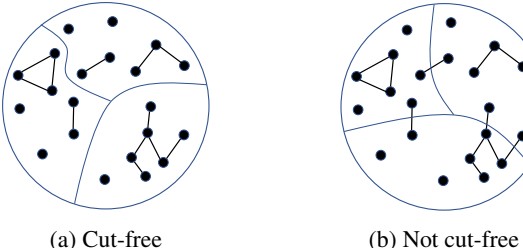

(a) Cut-free                    (b) Not cut-free

Figure 5: Two equipartitions of the same graph (each subsets of the equipartitions contain 7 vertices). The equipartition on the left is cut-free (no edges are severed). The equipartition on the right is not cut-free (4 edges are severed). The optimal kernel $K^\star(\mathbf{x}, \mathbf{y})$ can be interpreted as the number of distinct cut-free equipartitions of the graph $\zeta(\mathbf{x}, \mathbf{y})$ (modulo some scaling factor.)

Recall that $\Phi$ is the set of maps $\varphi : \mathcal{V} \to \{1, \ldots, n_c\}$ that satisfy $|\varphi^{-1}(c)| = s_c$ for all $1 \leq c \leq n_c$. Given a graph $\mathcal{G}$, the quantity $\mathcal{I}(\mathcal{G})$ can therefore be interpreted as the number of ways to partition the vertices into $n_c$ labelled subsets of equal size so that no edges are severed (i.e. two connected vertices must be in the same subset.) In other words, $\mathcal{I}(\mathcal{G})$ is the number of "cut-free" equipartition of the graph $\mathcal{G}$, see Figure 5 for an illustration. Note that if the graph $\mathcal{G}$ is connected, then $\mathcal{I}(\mathcal{G}) = 0$ since any equipartition of the graph will sever some edges. On the other hand, if the graph $\mathcal{G}$ has no edges, then $\mathcal{I}(\mathcal{G}) = |\Phi|$ since there are no edges to be cut (and therefore any equipartition is acceptable.)

The optimal kernel $K^\star$ can be expressed as a composition of the function $\zeta$ and $\mathcal{I}$. Indeed:

$$K^\star(\mathbf{x}, \mathbf{y}) = \frac{1}{|\Phi|s_c^L} \; |\{\varphi \in \Phi : \varphi(x_\ell) = \varphi(y_\ell) \text{ for all } 1 \leq \ell \leq L\}| \tag{82}$$

$$= \frac{1}{|\Phi|s_c^L} \; |\{\varphi \in \Phi : \varphi(v) = \varphi(v') \text{ for all } \{v, v'\} \in \zeta(\mathbf{x}, \mathbf{y})\}| \tag{83}$$

$$= \frac{1}{|\Phi|s_c^L} \; \mathcal{I}(\zeta(\mathbf{x}, \mathbf{y})) \tag{84}$$

where we have simply used that $\zeta(\mathbf{x}, \mathbf{y}) := \bigcup_{\substack{1 \leq \ell \leq L \\ x_\ell \neq y_\ell}} \{\{x_\ell, y_\ell\}\}$ to go from (82) to (83). We will refer to (84) as the *graph-cut formulation* of the optimal kernel.

We have discussed earlier that the function $\zeta : \mathcal{X} \times \mathcal{X} \to \mathfrak{G}$ is surjective but not injective. We conclude this subsection with a lemma that provides an exact count of how many distinct $(\mathbf{x}, \mathbf{y})$ are mapped by $\zeta$ to a same graph $\mathcal{G}$.

**Lemma 13.** *Suppose $\mathcal{G} \in \mathfrak{G}$ has $m$ edges. Then*

$$|\zeta^{-1}(\mathcal{G})| = |\{(\mathbf{x}, \mathbf{y}) \in \mathcal{X} \times \mathcal{X} : \zeta(\mathbf{x}, \mathbf{y}) = \mathcal{G}\}| = m! \sum_{\alpha=m}^{L} \binom{L}{\alpha} \begin{Bmatrix} \alpha \\ m \end{Bmatrix} 2^\alpha n_w^{L-\alpha}.$$

*Proof.* Fix once and for all a graph $\mathcal{G} = (\mathcal{V}, E)$ with edge set $E = \{e_1, \ldots, e_m\}$ where $m \leq L$. Given $0 \leq \alpha \leq L$, define the set

$$\mathcal{O}_\alpha = \{(\mathbf{x}, \mathbf{y}) \in \mathcal{X} \times \mathcal{X} : \zeta(\mathbf{x}, \mathbf{y}) = \mathcal{G} \text{ and } (\mathbf{x}, \mathbf{y}) \text{ has exactly } \alpha \text{ non-silent positions}\}.$$

We start by noting that the set $\mathcal{O}_\alpha$ is empty for all $\alpha < m$: indeed, since $\mathcal{G}$ has $m$ edges, at least $m$ positions of a pair $(\mathbf{x}, \mathbf{y})$ must be coding for edges (i.e., must be non-silent) in order to have $\zeta(\mathbf{x}, \mathbf{y}) = \mathcal{G}$. We therefore have the following partition:

$$\zeta^{-1}(\mathcal{G}) = \bigcup_{\alpha=m}^{L} \mathcal{O}_\alpha \qquad \text{and} \qquad \mathcal{O}_\alpha \cap \mathcal{O}_{\alpha'} = \emptyset \quad \text{if } \alpha \neq \alpha'.$$

To conclude the proof, we need to show that

$$|\mathcal{O}_\alpha| = \binom{L}{\alpha} \begin{Bmatrix} \alpha \\ m \end{Bmatrix} m! \, 2^\alpha \, n_w^{L-\alpha} \qquad \text{for all } m \leq \alpha \leq L. \tag{85}$$

Consider the following process to generate an ordered pair $(\mathbf{x}, \mathbf{y})$ that belong to $\mathcal{O}_\alpha$: we start by deciding which positions of $(\mathbf{x}, \mathbf{y})$ are going to be silent, and which positions are going to code for which edge of the graph $\mathcal{G}$. This is equivalent to choosing a map $\rho : \{1, \ldots, L\} \to \{e_1, \ldots, e_m, s\}$ where $\{1, \ldots, L\}$ denotes the $L$ positions, $e_1, \ldots, e_m$ denote the $m$ edges of the graph $\mathcal{G}$, and $s$ is the silent symbol. Choosing a map $\rho$ correspond to assigning a "role" to each position: $\rho(\ell) = e_i$ means that position $\ell$ is given the role to code for edge $e_i$, and $\rho(\ell) = s$ means that position $\ell$ is given the role of being silent. The map $\rho$ must satisfy:

$$|\rho^{-1}(s)| = L - \alpha \qquad \text{and} \qquad \rho^{-1}(e_i) \neq \emptyset \quad \text{for } 1 \leq i \leq m \tag{86}$$

because $L - \alpha$ position must be silent and each edge must be represented by at least one position. The number of maps $\rho : \{1, \ldots, L\} \to \{e_1, \ldots, e_m, s\}$ that satisfies (86) is equal to

$$\binom{L}{L-\alpha} \left\{ \begin{matrix} \alpha \\ m \end{matrix} \right\} m!$$

Indeed, we need to choose the $L - \alpha$ positions that will be mapped to the silent symbol $s$: there are $\binom{L}{L-\alpha}$ ways of accomplishing this. We then partition the $\alpha$ remaining positions into $m$ non-empty subsets: there are $\left\{ \begin{smallmatrix} \alpha \\ m \end{smallmatrix} \right\}$ ways of accomplishing this. We finally map each of these non-empty subset to a different edge: there are $m!$ ways of accomplishing this.

We have shown that there are $\binom{L}{\alpha} \left\{ \begin{smallmatrix} \alpha \\ m \end{smallmatrix} \right\} m!$ ways to assign roles to the positions. Let say that position $\ell$ is assigned the role of coding for edge $\{v, v'\}$. Then we have two choices to generate entries $x_\ell$ and $y_\ell$: either $x_\ell = v$ and $y_\ell = v'$, or $x_\ell = v'$ and $y_\ell = v$. Since $\alpha$ positions are coding for edges, this lead to the factor $2^\alpha$ in equation (85). Finally, if the position $\ell$ is silent, then we have $n_w$ choices to generate entries $x_\ell$ and $y_\ell$ (because we need to choose $v \in \mathcal{V}$ such that $x_\ell = y_\ell = v$) , hence the factor $n_w^{L-\alpha}$ appearing in (85). □

## C.2 Level Sets of the Optimal Kernel

We recall that a connected component (or simply a component) of a graph is a connected subgraph that is not part of any larger connected subgraph. Since graphs in $\mathfrak{G}$ have at most $L$ edges, their components contain at most $L + 1$ vertices. This comes from the fact that the largest component that can be made with $L$ edges contains $L + 1$ vertices. It is therefore natural, given a vector $\mathbf{k} = [k_1, \ldots, k_{L+1}] \in \mathbb{N}^{L+1}$, to define

$$\mathfrak{G}_\mathbf{k} := \{\mathcal{G} \in \mathfrak{G} : \mathcal{G} \text{ has exactly } k_1 \text{ components of size 1, exactly } k_2 \text{ components of size 2,}$$
$$\ldots, \text{ exactly } k_{L+1} \text{ components of size } L + 1 \} \tag{87}$$

where the size of a component refers to the number of vertices it contains. We recall that $\mathbb{N} = \{0, 1, 2, \ldots\}$ therefore some of the entries of the vector $\mathbf{k}$ can be equal to zero. Note that components of size 1 are simply isolated vertices. The following lemma identify which $\mathbf{k} \in \mathbb{N}^{L+1}$ lead to non-empty $\mathfrak{G}_\mathbf{k}$.

**Lemma 14.** *The set $\mathfrak{G}_\mathbf{k}$ is not empty if and only if $\mathbf{k}$ satisfies*

$$\sum_{i=1}^{L+1} i k_i = n_w \quad \text{and} \quad \sum_{i=1}^{L+1} (i-1) k_i \leq L. \tag{88}$$

*Proof.* Suppose $\mathfrak{G}_\mathbf{k}$ is not empty. Then there exists a graph $\mathcal{G} \in \mathfrak{G}$ that has exactly $k_i$ components of size $i$, for $1 \leq i \leq L + 1$. A component of size $i$ contains $i$ vertices (by definition) and at least $i - 1$ edges (otherwise it would not be connected.) Since $\mathcal{G} \in \mathfrak{G}$ it must have $n_w$ vertices and at most $L$ edges. Therefore (88) must hold.

Suppose that $\mathbf{k} \in \mathbb{N}^{L+1}$ satisfies (88). Then we can easily construct a graph $\mathcal{G}$ on $\mathcal{V}$ that has a number of edges less or equal to $L$, and that has exactly $k_i$ components of size $i$, for $1 \leq i \leq L+1$. To do this we first partition the vertices into subsets so that there are $k_i$ subsets of size $i$, for $1 \leq i \leq L+1$. We then put $i - 1$ edges on each subset of size $i$ so that they form connected components. The resulting graph has $k_i$ components of size $i$, for $1 \leq i \leq L + 1$, and $\sum_{i=1}^{L+1} (i-1) k_i$ edges. □

The previous lemma allows us to partition $\mathfrak{G}$ into non-empty subsets as follow:

$$\mathfrak{G} = \bigcup_{\mathbf{k} \in \mathcal{S}} \mathfrak{G}_{\mathbf{k}}, \qquad \mathfrak{G}_{\mathbf{k}} \neq \emptyset \text{ for all } \mathbf{k} \in \mathcal{S}, \qquad \text{and} \qquad \mathfrak{G}_{\mathbf{k}} \cap \mathfrak{G}_{\mathbf{k}'} = \emptyset \text{ if } \mathbf{k} \neq \mathbf{k}' \qquad (89)$$

$$\text{where } \mathcal{S} := \left\{ \mathbf{k} \in \mathbb{N}^{L+1} : \sum_{i=1}^{L+1} i k_i = n_w \quad \text{and} \quad \sum_{i=1}^{L+1} (i-1) k_i \leq L \right\}. \qquad (90)$$

Recall that $\mathcal{I}(\mathcal{G})$ count the number of equipartitions that do not severe edges of $\mathcal{G}$. The next lemma shows that two graphs that belongs to the same subset $\mathfrak{G}_{\mathbf{k}}$ have the same number of cut-free equipartitions, and it provides a formula for this number in term of the index $\mathbf{k}$ of the subset.

**Lemma 15.** *Suppose $\mathbf{k} \in \mathcal{S}$ and define the set of admissible assignment matrices*

$$\mathcal{A}_{\mathbf{k}} := \left\{ A \in \mathbb{N}^{(L+1) \times n_c} \quad : \quad \sum_{j=1}^{n_c} A_{ij} = k_i \text{ for all } i \quad \text{and} \quad \sum_{i=1}^{L+1} i A_{ij} = s_c \text{ for all } j \right\}. \qquad (91)$$

*Then for all $\mathcal{G} \in \mathfrak{G}_{\mathbf{k}}$, we have that*

$$\mathcal{I}(\mathcal{G}) = \sum_{A \in \mathcal{A}_{\mathbf{k}}} \prod_{i=1}^{L+1} \binom{k_i}{A_{i,1}, A_{i,2}, \ldots, A_{i,n_c}}. \qquad (92)$$

Let us remark that, since $0! = 1$, the multinomial coefficient $\binom{k_i}{A_{i,1}, A_{i,2}, \ldots, A_{i,n_c}}$ appearing in (92) is equal to 1 when $k_i$ is equal to 0.

*Poof of Lemma 15.* Let $\mathbf{k} \in \mathcal{S}$ and fix once and for all a graph $\mathcal{G} \in \mathfrak{G}_{\mathbf{k}}$. Define the set

$$\Psi = \{ \varphi \in \Phi : \varphi(v) = \varphi(v') \text{ for all edge } \{v, v'\} \text{ of the graph } \mathcal{G} \}$$

so that $\mathcal{I}(\mathcal{G}) = |\Psi|$. Note that a map $\varphi$ that belongs to $\Psi$ must map all vertices that are in a connected component to the same concept (otherwise some edges of $\mathcal{G}$ would be severed.) So a map $\varphi \in \Psi$ can be viewed as assigning connected components to concepts. Given a matrix $A \in \mathbb{N}^{(L+1) \times n_c}$ we define the set:

$$\Psi_A = \{ \varphi \in \Psi : \varphi \text{ assigns } A_{ij} \text{ components of size } i \text{ to concept } j, \text{ for all } 1 \leq i \leq L+1 \text{ and } 1 \leq j \leq n_c \}.$$

We then note that the set $\Psi_A$ is empty unless the matrix $A$ satisfies:

$$A_{i,1} + A_{i,2} + A_{i,3} + \ldots + A_{i,n_c} = k_i \qquad \text{for all } 1 \leq i \leq L+1$$
$$A_{1,j} + 2 A_{2,j} + 3 A_{3,j} + \ldots + (L+1) A_{L+1,j} = s_c \qquad \text{for all } 1 \leq j \leq n_c.$$

The first constraint states that the total number of connected components of size $i$ is equal to $k_i$ (because $\mathcal{G} \in \mathfrak{G}_{\mathbf{k}}$). The second constraint states that concept $j$ must receive a total of $s_c$ vertices (because $\varphi \in \Phi$.) The matrices that satisfy these two constraints constitute the set $\mathcal{A}_{\mathbf{k}}$ defined in (91). We therefore have the following partition of the set $\Psi$:

$$\Psi = \bigcup_{A \in \mathcal{A}_{\mathbf{k}}} \Psi_A, \qquad \Psi_A \neq \emptyset \text{ if } A \in \mathcal{A}_{\mathbf{k}}, \qquad \Psi_A \cap \Psi_B = \emptyset \text{ if } A \neq B.$$

To conclude the proof, we need to show that

$$|\Psi_A| = \prod_{i=1}^{L+1} \binom{k_i}{A_{i,1}, A_{i,2}, \ldots, A_{i,n_c}} \qquad \text{for all } A \in \mathcal{A}_{\mathbf{k}}. \qquad (93)$$

To see this, consider the $k_i$ components of size $i$. The number of ways to assign them to the $n_c$ concepts so that concept $j$ receives $A_{ij}$ of them is equal to the multinomial coefficient $\binom{k_i}{A_{i,1}, A_{i,2}, \ldots, A_{i,n_c}}$. Repeating this reasonning for the components of each size gives (93). $\square$

We now leverage the previous lemma to obtain a formula for $K^\star$. For $\mathbf{k} \in \mathcal{S}$ we define

$$\Omega_{\mathbf{k}} := \zeta^{-1}(\mathfrak{G}_{\mathbf{k}}).$$

Since $\zeta : \mathcal{X} \times \mathcal{X} \to \mathfrak{G}$ is surjective, partition (89) of $\mathfrak{G}$ induces the following partition of $\mathcal{X} \times \mathcal{X}$:

$$\mathcal{X} \times \mathcal{X} = \bigcup_{\mathbf{k} \in \mathcal{S}} \Omega_{\mathbf{k}}, \qquad \Omega_{\mathbf{k}} \neq \emptyset \text{ if } \mathbf{k} \in \mathcal{S} \qquad \text{and} \qquad \Omega_{\mathbf{k}} \cap \Omega_{\mathbf{k}'} = \emptyset \text{ if } \mathbf{k} \neq \mathbf{k}'. \tag{94}$$

Using the graph-cut formulation of the optimal kernel together with Lemma 15 we therefore have

$$K^{\star}(\mathbf{x}, \mathbf{y}) = \frac{1}{|\Phi| s_c^L} \ \mathcal{I}(\zeta(\mathbf{x}, \mathbf{y})) = \frac{1}{|\Phi| s_c^L} \ \sum_{A \in \mathcal{A}_{\mathbf{k}}} \prod_{i=1}^{L+1} \binom{k_i}{A_{i,1}, A_{i,2}, \ldots, A_{i,n_c}} \qquad \text{for all } (\mathbf{x}, \mathbf{y}) \in \Omega_{\mathbf{k}}. \tag{95}$$

The above formula is key to our analysis. We restate it in the lemma below, but in a slightly different format that will better suit the rest of the analysis. Let $\mathfrak{f} : \mathcal{S} \to \mathbb{R}$ be the function defined by

$$\mathfrak{f}(\mathbf{k}) := \frac{n_c^L \ (s_c!)^{n_c}}{n_w!} \ \sum_{A \in \mathcal{A}_{\mathbf{k}}} \prod_{i=1}^{L+1} \binom{k_i}{A_{i,1}, A_{i,2}, \ldots, A_{i,n_c}} \tag{96}$$

We then have:

**Lemma 16** (Level set decomposition of $K^{\star}$). *The kernel $K^{\star}$ is constant on each subsets $\Omega_{\mathbf{k}}$ of the partition (94). Moreover we have*

$$K^{\star}(\mathbf{x}, \mathbf{y}) = \ \mathfrak{f}(\mathbf{k})/|\mathcal{X}| \qquad \text{for all } (\mathbf{x}, \mathbf{y}) \in \Omega_{\mathbf{k}} \text{ and for all } \mathbf{k} \in \mathcal{S}.$$

*Proof.* The quantity $|\Phi|$ appearing in (95) can be interpreted as the number of ways to assign the $n_w$ words to the $n_c$ concepts so that each concept receives $s_c$ words. Therefore

$$|\Phi| = \binom{n_w}{s_c, s_c, \ldots, s_c} = \frac{n_w!}{(s_c!)^{n_c}}.$$

Combined with the fact that $|\mathcal{X}| = n_w^L$, this leads to the desired formula for $K^{\star}$. $\qquad\square$

The above lemma provides us with the level sets of the optimal kernel. Together with Lemma 3, this allows us to derive the following upper bound for the permuted moment of $K^{\star}$.

**Lemma 17.** *Let $\Omega = \mathcal{X} \times \mathcal{X}$. The inequality*

$$\frac{1}{|\mathcal{X}|} \sum_{\mathbf{x} \in \mathcal{X}} \mathcal{H}_t(K_{\mathbf{x}}^{\star}) \leq \left( 1 - \sum_{\mathbf{k} \in \mathcal{S}'} \frac{|\Omega_{\mathbf{k}}|}{|\Omega|} \ \mathfrak{f}(\mathbf{k}) \right) + \frac{1}{t+1} \left( \max_{\mathbf{k} \in \mathcal{S}'} \ \mathfrak{f}(\mathbf{k}) \right)$$

*holds for all $\mathcal{S}' \subset \mathcal{S}$.*

*Proof.* Let $\mathcal{S}' \subset \mathcal{S}$ and define:

$$\lambda = \max_{\mathbf{k} \in \mathcal{S}'} \max_{(\mathbf{x}, \mathbf{y}) \in \Omega_{\mathbf{k}}} K^{\star}(\mathbf{x}, \mathbf{y}) = \frac{1}{|\mathcal{X}|} \max_{\mathbf{k} \in \mathcal{S}'} \ \mathfrak{f}(\mathbf{k})$$

where we have used the fact that $K^{\star}$ is equal to $\mathfrak{f}(\mathbf{k})/|\mathcal{X}|$ on $\Omega_{\mathbf{k}}$. We then appeal to Lemma 3 to obtain:

$$\frac{1}{|\mathcal{X}|} \sum_{\mathbf{x} \in \mathcal{X}} \mathcal{H}_t\left(K^{\star}(\mathbf{x}, \cdot)\right) \leq \frac{1}{|\mathcal{X}|} \sum_{\mathbf{x} \in \mathcal{X}} \left( \frac{\lambda |\mathcal{X}|}{t+1} + 1 - \sum_{\mathbf{y} \in \mathcal{X}} \min\{K^{\star}(\mathbf{x}, \mathbf{y}), \lambda\} \right) \tag{97}$$

$$= \frac{\lambda |\mathcal{X}|}{t+1} + 1 - \frac{1}{|\mathcal{X}|} \sum_{\mathbf{x} \in \mathcal{X}} \sum_{\mathbf{y} \in \mathcal{X}} \min\{K^{\star}(\mathbf{x}, \mathbf{y}), \lambda\} \tag{98}$$

$$= \frac{\lambda |\mathcal{X}|}{t+1} + 1 - \frac{1}{|\mathcal{X}|} \sum_{\mathbf{k} \in \mathcal{S}} \sum_{(\mathbf{x}, \mathbf{y}) \in \Omega_{\mathbf{k}}} \min\{K^{\star}(\mathbf{x}, \mathbf{y}), \lambda\} \tag{99}$$

$$\leq \frac{\lambda |\mathcal{X}|}{t+1} + 1 - \frac{1}{|\mathcal{X}|} \sum_{\mathbf{k} \in \mathcal{S}'} \sum_{(\mathbf{x}, \mathbf{y}) \in \Omega_{\mathbf{k}}} \min\{K^{\star}(\mathbf{x}, \mathbf{y}), \lambda\} \tag{100}$$

$$= \frac{\lambda |\mathcal{X}|}{t+1} + 1 - \frac{1}{|\mathcal{X}|} \sum_{\mathbf{k} \in \mathcal{S}'} \sum_{(\mathbf{x}, \mathbf{y}) \in \Omega_{\mathbf{k}}} K^{\star}(\mathbf{x}, \mathbf{y}) \tag{101}$$

$$= \frac{\lambda |\mathcal{X}|}{t+1} + 1 - \frac{1}{|\mathcal{X}|} \sum_{\mathbf{k} \in \mathcal{S}'} |\Omega_{\mathbf{k}}| \frac{f(\mathbf{k})}{|\mathcal{X}|} \tag{102}$$

where we have use the fact that $\mathcal{X} \times \mathcal{X} = \bigcup_{\mathbf{k} \in \mathcal{S}} \Omega_{\mathbf{k}}$ to go from (98) to (99), and the fact that $K^\star(\mathbf{x}, \mathbf{y}) \leq \lambda$ on $\bigcup_{\mathbf{k} \in \mathcal{S}'} \Omega_{\mathbf{k}}$ to go from (100) to (101). To conclude the proof, we simply note that $\lambda |\mathcal{X}| = \max_{\mathbf{k} \in \mathcal{S}'} \mathfrak{f}(\mathbf{k})$ according to our definition of $\lambda$. □

The bound provided by the above lemma is not fully explicit because it involves the size of level sets $\Omega_{\mathbf{k}}$. In the next section, we appeal to Cayley's formula to obtain a lower bound for $|\Omega_{\mathbf{k}}|$.

### C.3 FOREST LOWER BOUND FOR THE SIZE OF THE LEVEL SETS

We recall that a forest is a graph whose connected components are trees (equivalently, a forest is a graph with no cycles.) Let us define:

$$\mathfrak{F} := \{\mathcal{G} \in \mathfrak{G} : \mathcal{G} \text{ is a forest}\}.$$

In other words, $\mathfrak{F}$ is the set of forests on $\mathcal{V} = \{1, 2, \ldots, n_w\}$ that have at most $L$ edges. We obviously have the following lower bound on the size of the level sets:

$$|\Omega_{\mathbf{k}}| = \left|\zeta^{-1}(\mathfrak{G}_{\mathbf{k}})\right| \geq \left|\zeta^{-1}(\mathfrak{G}_{\mathbf{k}} \cap \mathfrak{F})\right|. \tag{103}$$

In this subsection, we use Cayley's formula to derive an explicit formula for $\left|\zeta^{-1}(\mathfrak{G}_{\mathbf{k}} \cap \mathfrak{F})\right|$. We start with the following lemma:

**Lemma 18.** *Let* $\mathbf{k} \in \mathcal{S}$, *then*

$$|\mathfrak{G}_{\mathbf{k}} \cap \mathfrak{F}| = \frac{n_w!}{k_1! k_2! \cdots k_{L+1}!} \prod_{i=2}^{L+1} \left(\frac{i^{i-2}}{i!}\right)^{k_i} \tag{104}$$

*Proof.* First we note that (104) can be written as

$$|\mathfrak{G}_{\mathbf{k}} \cap \mathfrak{F}| = \binom{n_w}{k_1, 2k_2, \ldots, (L+1)k_{L+1}} \prod_{i=2}^{L+1} \frac{i^{k_i(i-2)}}{k_i!} \binom{ik_i}{i, i, \ldots, i}$$

We now explain the above formula. The set $\mathfrak{G}_{\mathbf{k}} \cap \mathfrak{F}$ consists in all the forests that have exactly $k_1$ trees of size 1, $k_2$ trees of size 2, $\ldots$, $k_{L+1}$ trees of size $L+1$. In order to construct a forest with this specific structure, we start by assigning the $n_w$ vertices to $L+1$ bins, with bin 1 receiving $k_1$ vertices, bin 2 receiving $2k_2$ vertices, $\ldots$, bin $L+1$ receiving $(L+1)k_{L+1}$ vertices. The number of ways of accomplishing this is

$$\binom{n_w}{k_1, 2k_2, \ldots, (L+1)k_{L+1}}.$$

Let us now consider the vertices in bin $i$ for some $i \geq 2$. We claim that there are

$$\frac{1}{k_i!} \binom{ik_i}{i, i, \ldots, i} i^{k_i(i-2)}$$

ways of putting edges on these $ik_i$ vertices in order to make $k_i$ trees of size $i$. Indeed, there are $\frac{1}{k_i!} \binom{ik_i}{i, i, \ldots, i}$ ways of partitioning the vertices into $k_i$ disjoint subsets of size $i$, and then, according to Cayley's formula, there are $i^{i-2}$ ways of putting edges on each of these subsets so that they form a tree. To conclude, we remark that there is obviously only one way to to make $k_1$ trees of size 1 out of the $k_1$ vertices in the first bin. □

Recall that a tree on $n$ vertices always has $n - 1$ edges. So a graph that belongs to $\mathfrak{G}_{\mathbf{k}} \cap \mathfrak{F}$ has

$$m = \sum_{i=1}^{L+1} (i-1)k_i$$

edges since it is made of $k_1$ trees of size 1, $k_2$ trees of size 2, $\ldots$, $k_{L+1}$ trees of size $(L+1)$. The fact that all graphs in $\mathfrak{G}_{\mathbf{k}} \cap \mathfrak{F}$ have the same number of edges allows us to to obtain an explicit formula for $|\zeta^{-1}(\mathfrak{G}_{\mathbf{k}} \cap \mathfrak{F})|$ by combining combine Lemma 13 and 18, namely

$$|\zeta^{-1}(\mathfrak{G}_{\mathbf{k}} \cap \mathfrak{F})| = \left(\frac{n_w!}{k_1! k_2! \cdots k_{L+1}!} \prod_{i=2}^{L+1} \left(\frac{i^{i-2}}{i!}\right)^{k_i}\right) \left(m! \sum_{\alpha=m}^{L} \binom{L}{\alpha} \begin{Bmatrix} \alpha \\ m \end{Bmatrix} 2^\alpha n_w^{L-\alpha}\right).$$

This lead us to define the function $\mathfrak{g} : \mathcal{S} \to \mathbb{R}$ by

$$\mathfrak{g}(\mathbf{k}) = \frac{1}{n_w^{2L}} \left( \frac{n_w!}{k_1! k_2! \cdots k_{L+1}!} \prod_{i=2}^{L+1} \left( \frac{i^{i-2}}{i!} \right)^{k_i} \right) \left( \gamma(\mathbf{k})! \sum_{\alpha=\gamma(\mathbf{k})}^{L} \binom{L}{\alpha} \left\{ \begin{matrix} \alpha \\ \gamma(\mathbf{k}) \end{matrix} \right\} 2^{\alpha} n_w^{L-\alpha} \right)$$

$$\text{where } \gamma(\mathbf{k}) = \sum_{i=1}^{L+1} (i-1) k_i.$$

Recalling (103) we therefore have that

$$\frac{|\Omega_{\mathbf{k}}|}{|\Omega|} \geq \mathfrak{g}(\mathbf{k}) \qquad \text{for all } \mathbf{k} \in \mathcal{S}.$$

Combining the above inequality with Lemma 17 we obtain:

**Theorem 6.** *The inequality*

$$\frac{1}{|\mathcal{X}|} \sum_{\mathbf{x} \in \mathcal{X}} \mathcal{H}_t(K_{\mathbf{x}}^{\star}) \leq \left( 1 - \sum_{\mathbf{k} \in \mathcal{S}'} \mathfrak{g}(\mathbf{k}) \, \mathfrak{f}(\mathbf{k}) \right) + \frac{1}{t+1} \left( \max_{\mathbf{k} \in \mathcal{S}'} \mathfrak{f}(\mathbf{k}) \right)$$

*holds for all $\mathcal{S}' \subset \mathcal{S}$.*

The above theorem is more general than Theorem 5 — indeed, in Theorem 5, the choice of the subset $\mathcal{S}'$ is restricted to the $L+1$ candidates:

$$\mathcal{S}_\ell := \left\{ \mathbf{k} \in \mathbb{N}^{L+1} : \sum_{i=1}^{L+1} i k_i = n_w \quad \text{and} \quad \ell \leq \sum_{i=1}^{L+1} (i-1) k_i \leq L \right\} \qquad \text{where } \ell = 0, 1, \dots, L.$$

When $L = 9$, $n_w = 150$, $n_c = 5$ and $t = 1999$ (these are the parameters used in Theorem 1), choosing $\mathcal{S}' = \mathcal{S}_7$ leads to a relatively tight upper bound. When $L = 15$, $n_w = 30$, $n_c = 5$ and $t = 5999$ (these are the parameters corresponding the the second experiment of the experimental section), choosing $\mathcal{S}' = \mathcal{S}_{11}$ gives a good upper bound.

## D  MULTIPLE UNFAMILIAR SENTENCES PER CATEGORY

In the data model depicted in Figure 1, each unfamiliar sequence of concept has a single representative in the training set. In this section we consider the more general case in which each unfamiliar sequence of concepts has $n^*$ representatives in the training set, where

$$1 \leq n^* \leq n_{\text{spl}}.$$

An example with $n^* = 2$ is depicted in Figure 6. The variables $L, n_w, n_c, R, n_{\text{spl}}$ and $n^*$ parametrize instances of this more general data model, and the associated sampling process is:

**Sampling Process DM2:**

---

(i)  Sample $\mathcal{T} = ( \, \varphi \, ; \, \mathbf{c}_1, \dots, \mathbf{c}_R \, ; \, \mathbf{c}_1', \dots, \mathbf{c}_R' \, )$ uniformly at random in $\mathfrak{T} = \Phi \times \mathcal{Z}^{2R}$.

(ii)  For $r = 1, \dots, R$:

- Sample $(\mathbf{x}_{r,1}, \dots, \mathbf{x}_{r,n^*})$ uniformly at random in $\mathring{\varphi}^{-1}(\mathbf{c}_r') \times \dots \times \mathring{\varphi}^{-1}(\mathbf{c}_r')$.
- Sample $(\mathbf{x}_{r,n^*+1}, \dots, \mathbf{x}_{r,n_{\text{spl}}})$ uniformly at random in $\mathring{\varphi}^{-1}(\mathbf{c}_r) \times \dots \times \mathring{\varphi}^{-1}(\mathbf{c}_r)$.

(iii)  Sample $\mathbf{x}^{\text{test}}$ uniformly at random in $\mathring{\varphi}^{-1}(\mathbf{c}_1')$.

---

Our analysis easily adapts to this more general case and gives:

**Theorem 7.** *Let $\mathfrak{T} = \Phi \times \mathcal{Z}^{2R}$. Then*

$$1 - \overline{err}(\mathcal{F}, \psi, \mathfrak{T}) \leq n^* \left[ \left( 1 - \sum_{\mathbf{k} \in \mathcal{S}_\ell} \mathfrak{f}(\mathbf{k}) \mathfrak{g}(\mathbf{k}) \right) + \frac{1}{2R} \left( \max_{\mathbf{k} \in \mathcal{S}_\ell} \mathfrak{f}(\mathbf{k}) \right) \right] + \frac{1}{R} \qquad (105)$$

*for all feature space $\mathcal{F}$, all feature map $\psi : \mathcal{X} \mapsto \mathcal{F}$, and all $0 \leq \ell \leq L$.*

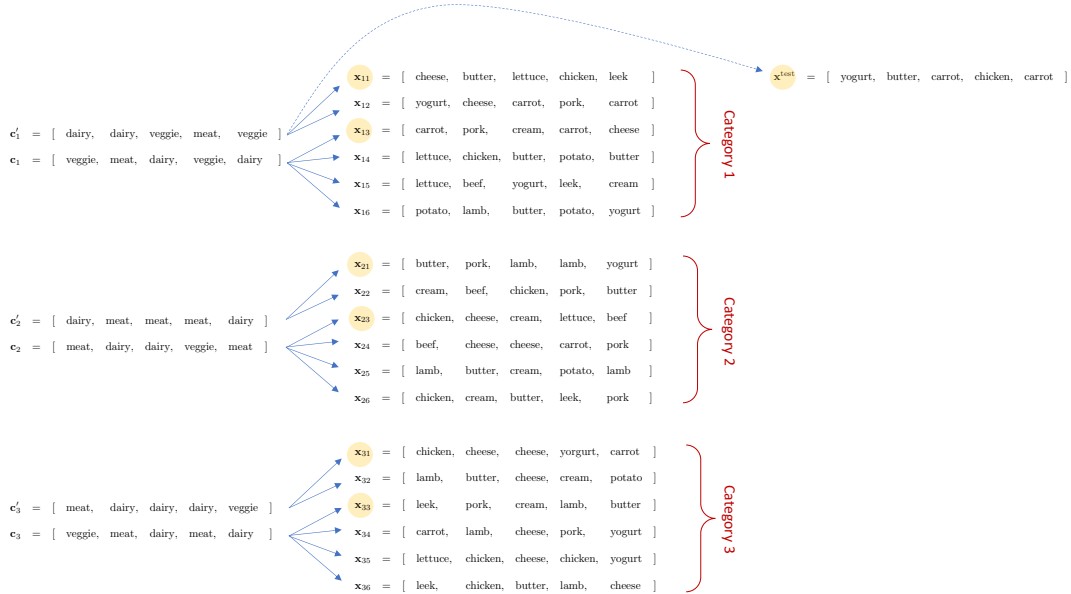

Figure 6: Data Model with $n^* = 2$ unfamiliar sentences per category. The other parameters in this example are set to $L = 5$, $n_w = 12$, $n_c = 3$, $R = 3$ and $n_{\mathrm{spl}} = 6$. The points highlighted in yellow are the ones involved in the definition of the event $A^{(1)}$, see equation (109).

Theorem 3 in the main body of the paper can be viewed as a special case of the above theorem — indeed, setting $n^* = 1$ in inequality (105) we exactly recover (9).

In order to prove Theorem 7, we simply need to revisit subsection B.3. We denote by $\Omega_{\mathrm{DM2}}$ and $\mathbb{P}_{\mathrm{DM2}}$ the sample space and probability measure associated with the sampling process DM2. As in subsection B.3, given a kernel $K \in \mathcal{K}$, we define the event

$$E_K = \Big\{ \omega \in \Omega_{\mathrm{DM2}} : \quad \text{There exists } 1 \le s^* \le n_{\mathrm{spl}} \text{ such that}$$

$$K\left(\mathbf{x}^{\mathrm{test}}, \mathbf{x}_{1,s^*}\right) > K\left(\mathbf{x}^{\mathrm{test}}, \mathbf{x}_{r,s}\right) \text{ for all } 2 \le r \le R \text{ and all } 1 \le s \le n_{\mathrm{spl}} \Big\}.$$

Such event describes all outcomes corresponding to successful classification of the test point $\mathbf{x}^{\mathrm{test}}$. For simplicity let us assume that $n^* = 2$ (therefore matching the scenario depicted in Figure 6). We further partition the event $E_K$ according to which training point from the first category is most similar to the test point:

$$E_K = E_{\mathrm{meaningful}}^{(1)} \cup E_{\mathrm{meaningful}}^{(2)} \cup E_{\mathrm{luck}} \tag{106}$$

The event $E_{\mathrm{meaningful}}^{(1)}$ consists in all the outcomes where, among the points from first category, $\mathbf{x}_{1,1}$ is the most similar to $\mathbf{x}^{\mathrm{test}}$, $E_{\mathrm{meaningful}}^{(2)}$ consists in all the outcomes where, among the points from first category, $\mathbf{x}_{1,2}$ is the most similar to $\mathbf{x}^{\mathrm{test}}$, and $E_{\mathrm{luck}}$ consists in all the remaining cases. Formally we have:

$$E_{\mathrm{meaningful}}^{(1)} = E_K \ \cap \ \Big\{ \omega \in \Omega_{\mathrm{DM2}} : \quad K\left(\mathbf{x}^{\mathrm{test}}, \mathbf{x}_{1,1}\right) > K\left(\mathbf{x}^{\mathrm{test}}, \mathbf{x}_{1,2}\right) \Big\}$$

$$\cap \ \Big\{ \omega \in \Omega_{\mathrm{DM2}} : \quad K\left(\mathbf{x}^{\mathrm{test}}, \mathbf{x}_{1,1}\right) > K\left(\mathbf{x}^{\mathrm{test}}, \mathbf{x}_{1,s}\right) \text{ for all } 3 \le s \le n_{\mathrm{spl}} \Big\}$$

$$E_{\mathrm{meaningful}}^{(2)} = E_K \ \cap \ \Big\{ \omega \in \Omega_{\mathrm{DM2}} : \quad K\left(\mathbf{x}^{\mathrm{test}}, \mathbf{x}_{1,2}\right) \ge K\left(\mathbf{x}^{\mathrm{test}}, \mathbf{x}_{1,1}\right) \Big\}$$

$$\cap \ \Big\{ \omega \in \Omega_{\mathrm{DM2}} : \quad K\left(\mathbf{x}^{\mathrm{test}}, \mathbf{x}_{1,2}\right) > K\left(\mathbf{x}^{\mathrm{test}}, \mathbf{x}_{1,s}\right) \text{ for all } 3 \le s \le n_{\mathrm{spl}} \Big\}$$

$$E_{\mathrm{luck}} = E_K \ \cap \ \Big\{ \omega \in \Omega_{\mathrm{DM2}} : \quad \text{there exists } 3 \le s^* \le n_{\mathrm{spl}} \text{ such that}$$

$$K\left(\mathbf{x}^{\mathrm{test}}, \mathbf{x}_{1,s^*}\right) \ge K\left(\mathbf{x}^{\mathrm{test}}, \mathbf{x}_{1,s}\right) \text{ for all } 1 \le s \le n_{\mathrm{spl}} \Big\}$$

Exactly as in subsection B.3, we then prove that

$$\mathbb{P}_{\text{DM2}}[E_{\text{meaningful}}^{(i)}] \le \frac{1}{|\mathcal{X}|} \sum_{\mathbf{x} \in \mathcal{X}} \mathcal{H}_{2R-1}(K_{\mathbf{x}}^{\star}) \quad \text{for } i = 1, 2 \tag{107}$$

$$\mathbb{P}_{\text{DM2}}[E_{\text{luck}}] \le \frac{1}{R}. \tag{108}$$

The proof of inequality (107) is essentially identical to the proof of Lemma 10. We define the event

$$A^{(1)} := \left\{ \omega \in \Omega_{\text{DM2}} : \quad K\left(\mathbf{x}^{\text{test}}, \mathbf{x}_{1,1}\right) > K\left(\mathbf{x}^{\text{test}}, \mathbf{x}_{r,1}\right) \text{ for all } 2 \le r \le R \right\}$$

$$\cap \left\{ \omega \in \Omega_{\text{DM2}} : \quad K\left(\mathbf{x}^{\text{test}}, \mathbf{x}_{1,1}\right) > K\left(\mathbf{x}^{\text{test}}, \mathbf{x}_{r,3}\right) \text{ for all } 1 \le r \le R \right\} \tag{109}$$

and the event

$$A^{(2)} := \left\{ \omega \in \Omega_{\text{DM2}} : \quad K\left(\mathbf{x}^{\text{test}}, \mathbf{x}_{1,2}\right) > K\left(\mathbf{x}^{\text{test}}, \mathbf{x}_{r,2}\right) \text{ for all } 2 \le r \le R \right\}$$

$$\cap \left\{ \omega \in \Omega_{\text{DM2}} : \quad K\left(\mathbf{x}^{\text{test}}, \mathbf{x}_{1,2}\right) > K\left(\mathbf{x}^{\text{test}}, \mathbf{x}_{r,3}\right) \text{ for all } 1 \le r \le R \right\}$$

The $\mathbf{x}$'s involved in the definition of the event $A^{(1)}$ are highlighted in yellow in Figure 6. Crucially they are all generated by different sequences of concepts, except for $\mathbf{x}_{1,1}$ and $\mathbf{x}^{\text{test}}$. We can therefore appeal to Theorem 4 to obtain

$$\mathbb{P}_{\text{DM2}}[A^{(1)}] \le \frac{1}{|\mathcal{X}|} \sum_{\mathbf{x} \in \mathcal{X}} \mathcal{H}_{2R-1}(K_{\mathbf{x}}^{\star})$$

since there is a total of $t = 2R - 1$ 'distractors' (the 'distractors' in Figure 6 are $\mathbf{x}_{1,3}$, $\mathbf{x}_{2,1}$, $\mathbf{x}_{2,3}$, $\mathbf{x}_{3,1}$ and $\mathbf{x}_{3,3}$). We then use the fact $E_{\text{meaningful}}^{(1)} \subset A^{(1)}$ to obtain (107) with $i = 1$. The case $i = 2$ is exactly similar.

We now prove (108). The proof is similar to the proof of Lemma 11. For $1 \le r \le R$, we define the events

$$B_r = \bigcap_{\substack{1 \le r' \le R \\ r' \ne r}} \left\{ \omega \in \Omega_{\text{DM2}} : \quad \max_{3 \le s \le n_{\text{spl}}} K(\mathbf{x}^{\text{test}}, \mathbf{x}_{r,s}) > \max_{3 \le s' \le n_{\text{spl}}} K(\mathbf{x}^{\text{test}}, \mathbf{x}_{r',s'}) \right\}$$

By symmetry, these events are equiprobable. They also are mutually disjoints, and therefore $\mathbb{P}_{\text{DM2}}[B_r] \le 1/R$. Inequality (108) then comes from the fact that $E_{\text{luck}} \subset B_1$.

Combining (106), (107), (108) then gives

$$\sup_{K \in \mathcal{K}} \mathbb{P}_{\text{DM2}}\Big[E_K\Big] \le \frac{2}{|\mathcal{X}|} \sum_{\mathbf{x} \in \mathcal{X}} \mathcal{H}_{2R-1}\left(K_{\mathbf{x}}^{\star}\right) + \frac{1}{R}. \tag{110}$$

and, going back to the general case where $n^*$ denotes the number of representatives that each sequence of unfamiliar concepts has in the training set,

$$\sup_{K \in \mathcal{K}} \mathbb{P}_{\text{DM2}}\Big[E_K\Big] \le \frac{n^*}{|\mathcal{X}|} \sum_{\mathbf{x} \in \mathcal{X}} \mathcal{H}_{2R-1}\left(K_{\mathbf{x}}^{\star}\right) + \frac{1}{R} \tag{111}$$

which in turn implies (16). Combining inequalities (15) and (16) then concludes the proof of Theorem 7.

# E   DETAILS OF THE EXPERIMENTS

In this section we provide the details of the experiments described in Section 6, as well as additional experiments. Table 4 provides the results of experiments in which the parameters $L$, $n_w$, $n_c$ and $R$ are set to

$$L = 9, \quad n_w = 150, \quad n_c = 5, \quad R = 1000.$$

Table 3: Accuracy in % on unfamiliar test points ($L = 9$, $n_w = 150$, $n_c = 5$, $R = 1000$).

| | $n^* = 1$ $n_{\text{spl}} = 6$ | $n^* = 2$ $n_{\text{spl}} = 7$ | $n^* = 3$ $n_{\text{spl}} = 8$ | $n^* = 4$ $n_{\text{spl}} = 9$ | $n^* = 5$ $n_{\text{spl}} = 10$ |
|---|---|---|---|---|---|
| Neural network | $99.8 \pm 0.3$ | $99.9 \pm 0.1$ | $99.9 \pm 0.1$ | $99.9 \pm 0.1$ | $100 \pm 0.1$ |
| NN on feat. extracted by neural net | $99.9 \pm 0.1$ | $99.9 \pm 0.1$ | $99.9 \pm 0.1$ | $99.9 \pm 0.1$ | $99.9 \pm 0.1$ |
| NN on feat. extracted by $\psi^\star$ | $0.7 \pm 0.2$ | $1.1 \pm 0.3$ | $1.5 \pm 0.3$ | $1.8 \pm 0.3$ | $2.2 \pm 0.3$ |
| NN on feat. extracted by $\psi_{\text{one-hot}}$ | $0.6 \pm 0.2$ | $1.1 \pm 0.3$ | $1.4 \pm 0.2$ | $1.7 \pm 0.3$ | $2.1 \pm 0.3$ |
| Upper bound ($0.015n^* + 1/1000$) | $1.6$ | $3.1$ | $4.6$ | $6.1$ | $7.6$ |
| SVM on feat. extracted by $\psi^\star$ | $0.6 \pm 0.3$ | $1.5 \pm 0.4$ | $2.2 \pm 0.4$ | $3.2 \pm 0.6$ | $4.2 \pm 1.0$ |
| SVM on feat. extracted by $\psi_{\text{one-hot}}$ | $0.5 \pm 0.1$ | $1.1 \pm 0.1$ | $1.9 \pm 0.1$ | $2.8 \pm 0.2$ | $3.8 \pm 0.2$ |
| SVM with Gaussian kernel | $0.6 \pm 0.1$ | $1.1 \pm 0.1$ | $2.0 \pm 0.1$ | $2.8 \pm 0.2$ | $3.6 \pm 0.2$ |

Table 4: Accuracy in % on unfamiliar test points ($L = 9$, $n_w = 50$, $n_c = 5$, $R = 1000$).

| | $n^* = 1$ $n_{\text{spl}} = 6$ | $n^* = 2$ $n_{\text{spl}} = 7$ | $n^* = 3$ $n_{\text{spl}} = 8$ | $n^* = 4$ $n_{\text{spl}} = 9$ | $n^* = 5$ $n_{\text{spl}} = 10$ |
|---|---|---|---|---|---|
| Neural network | $99.9 \pm 0.1$ | $99.9 \pm 0.1$ | $99.9 \pm 0.1$ | $99.9 \pm 0.1$ | $100 \pm 0.1$ |
| NN on feat. extracted by neural net | $99.9 \pm 0.1$ | $99.9 \pm 0.1$ | $99.9 \pm 0.1$ | $99.9 \pm 0.1$ | $99.9 \pm 0.1$ |
| NN on feat. extracted by $\psi^\star$ | $2.4 \pm 0.3$ | $4.1 \pm 0.6$ | $5.5 \pm 0.6$ | $6.9 \pm 0.8$ | $8.0 \pm 0.8$ |
| NN on feat. extracted by $\psi_{\text{one-hot}}$ | $2.0 \pm 0.3$ | $3.4 \pm 0.5$ | $4.8 \pm 0.6$ | $5.7 \pm 0.5$ | $6.7 \pm 0.7$ |
| Upper bound ($0.073n^* + 1/1000$) | $7.4$ | $14.7$ | $22.0$ | $29.3$ | $36.6$ |
| SVM on feat. extracted by $\psi^\star$ | $2.2 \pm 0.5$ | $5.2 \pm 0.9$ | $8.6 \pm 0.9$ | $11.7 \pm 0.6$ | $15.1 \pm 1.2$ |
| SVM on feat. extracted by $\psi_{\text{one-hot}}$ | $1.2 \pm 0.1$ | $3.5 \pm 0.2$ | $6.4 \pm 0.2$ | $9.9 \pm 0.3$ | $13.6 \pm 0.4$ |
| SVM with Gaussian kernel | $2.0 \pm 0.1$ | $3.7 \pm 0.2$ | $5.4 \pm 0.2$ | $8.6 \pm 0.3$ | $12.1 \pm 0.3$ |

The parameters $n_{\text{spl}}$ and $n^*$ are chosen so that the training set contains 5 familiar sentences per category, and between 1 and 5 unfamiliar sentences per category. Table 3 is identical to Table 1 in Section 6, with the exception that it contains additional information (i.e. the standard deviations of the obtained accuracies). The abbreviation NN appearing in Table 3 stands for 'Nearest Neighbor'.

Table 4 provides the results of additional experiments in which the parameters $L$, $n_w$, $n_c$ and $R$ are set to

$$L = 9, \quad n_w = 50, \quad n_c = 5, \quad R = 1000.$$

The parameters $n_{\text{spl}}$ and $n^*$ are chosen, as in the previous set experiments, so that the training set contains 5 familiar sentences per category, and between 1 and 5 unfamiliar sentences per category. The tasks considered in this set of experiments are easier due to the fact that the vocabulary is smaller ($n_w = 50$ instead of $n_w = 150$).

### E.1 NEURAL NETWORK EXPERIMENTS

We consider the neural network in Figure 2. The number of neurons in the input, hidden and output layers of the MLPs constituting the neural network are set to:

MLP 1: $d_{\text{in}} = 150, d_{\text{hidden}} = 500, d_{out} = 10$,      MLP 2: $d_{in} = 90, d_{\text{hidden}} = 2000, d_{\text{out}} = 1000$.

For each of the 10 possible parameter settings in Table 3 and Table 4, we do 104 experiments. For each experiment we generate:

- A training set containing $R \times n_{\text{spl}}$ sentences.
- A test set containing $10{,}000$ unfamiliar sentences (10 sentences per category).

We then train the neural network with stochastic gradient descent until the training loss reaches $10^{-4}$ (we use a cross entropy loss). The learning rate is set to $0.01$ (constant learning rate), and the batch size to 100. At test time, we either use the neural network to classify the test points (first row of the tables) or we use a nearest neighbor classification rule on the top of the features extracted by the neural network (second row of the tables). The mean and standard deviation of the 104 test

accuracies, for each of the 10 settings, and for each of the two evaluation strategies, is reported in the first two rows of Table 3 and Table 4.

## E.2 NEAREST-NEIGHBOR EXPERIMENTS

In these experiments we use a nearest neighbor classification rule on the top of features extracted by $\psi^\star$ (third row of Table 3 and 4) or $\psi_{\text{one}-\text{hot}}$ (fourth row). For each of the 10 possible parameter settings in Table 3 and Table 4, we do 50 experiments. For each experiment we generate:

- A training set containing $R \times n_{\text{spl}}$ sentences.
- A test set containing $1,000$ unfamiliar sentences (one sentences per category).

In order to perform the nearest neighbor classification rule on the features extracted by $\psi^\star$, one needs to evaluate the kernel $K^\star(\mathbf{x}, \mathbf{y}) = \langle \psi^\star(\mathbf{x}), \psi^\star(\mathbf{y}) \rangle_{\mathcal{F}^\star}$ for each pair of sentences. Computing $K^\star(\mathbf{x}, \mathbf{y})$ requires an expensive combinatorial calculation which is the reason why we perform fewer experiments and use a smaller test set than in E.1. In order to break ties, the values of $K^\star(\mathbf{x}, \mathbf{y})$ are perturbed according to (31).

With the parameter setting $L = 9$, $n_w = 50$, $n_c = 5$ and $R = 1000$, our theoretical lower bound for the generalization error is

$$\overline{\text{err}}(\mathcal{F}, \psi, \mathfrak{T}) \geq 1 - 0.073 \, n^* - 1/R \qquad \text{for all } \mathcal{F} \text{ and all } \psi, \tag{112}$$

which is obtained by choosing $\ell = 6$ in inequality (105). This lead to an upper bound of $0.073 \, n^* + 1/R$ on the success rate. This upper bound is evaluated for $n^*$ ranging from 1 to 5 in the fifth row of Table 4.

## E.3 SVM ON FEATURES EXTRACTED BY $\psi_{\text{one}-\text{hot}}$ AND SVM WITH GAUSSIAN KERNEL

For each of the 10 possible parameter settings in Table 3 and Table 4, we do 100 experiments. For each experiment we generate:

- A training set containing $R \times n_{\text{spl}}$ sentences.
- A test set containing $10,000$ unfamiliar sentences (10 sentences per category).

We use the feature map $\psi_{\text{one}-\text{hot}}$ (which simply concatenates the one-hot-encodings of the words composing a sentence) to extract features from each sentence. These features are further normalized according to the formula

$$\tilde{\mathbf{x}} = \frac{\psi_{\text{one}-\text{hot}}(\mathbf{x}) - p}{\sqrt{p(1 - p)}} \qquad \text{where } p = 1/n_w \tag{113}$$

so that they are centered around 0 and are $O(1)$. We then use the SVC function of Scikit-learn Pedregosa et al. (2011), which itself relies on the LIBSVM library Chang & Lin (2011), in order to run a soft multiclass SVM algorithm on these features. We tried various values for the parameter controlling the $\ell_2$ regularization in the soft-SVM formulation, and found that the algorithm, on this task, is not sensitive to this choice — so we chose $C = 1$. The results are reported in the seventh row of both tables.

We also tried a soft SVM with Gaussian Kernel

$$K(\mathbf{x}, \mathbf{y}) = e^{-\gamma \|\mathbf{x} - \mathbf{y}\|^2}$$

applied on the top of features extracted by $\psi_{\text{one}-\text{hot}}$ and normalized according to (113). We use the SVC function of Scikit-learn with $\ell_2$ regularization parameter set to $C = 1$. For the experiments in Table 3 ($n_w = 150$), the parameter $\gamma$ involved in the definition of the kernel was set to $\gamma = 0.25$ when $n^* \in \{1, 2\}$ and to $\gamma = 0.1$ when $n^* \in \{3, 4, 5\}$. For the experiments in Table 4 ($n_w = 50$), it was set to $\gamma = 0.75$ when $n^* = 1$, to $\gamma = 0.1$ when $n^* = 2$, and finally to $\gamma = 0.005$ when $n^* \in \{3, 4, 5\}$.

Table 5: Search for the hyperparameter $\alpha$

| $\alpha$ | $\lambda_{\min}(G^{\text{train}})$ | $\lambda_{\max}(G^{\text{train}})$ | Test Accuracy |
|---|---|---|---|
| 0.001 | $-90.9$ | $50,583.3$ | 6.1% |
| 0.01 | $-81.5$ | $32,334.9$ | 6.1% |
| 0.1 | $-56.8$ | $15,358.7$ | 6.6% |
| 1.0 | $-22.9$ | $4,191.5$ | 7.6% |
| 10 | $-2.5$ | $673.4$ | 7.2% |
| 15 | $-0.138$ | $471.7$ | 7.0% |
| 16 | $0.2$ | $445.5$ | 7.0% |
| 100 | $4.7$ | $86.9$ | 5.4% |
| 1000 | $4.983$ | $15.573$ | 5.0% |

### E.4 SVM ON FEATURES EXTRACTED BY $\psi^\star$

Applying a SVM on the feature extracted by $\psi^\star$ is equivalent to running a kernelized SVM with kernel $K^\star$. A naive implementation of such algorithm leads to very poor results on our data model. For such algorithm to not completely fail, it is important to carefully "rescale" $K^\star$ so that the eigenvalues of the corresponding Gram matrix are well behaved. Recall that

$$K^\star(\mathbf{x},\mathbf{y}) = \frac{n_c^L}{n_w^L} \frac{\left|\{\varphi \in \Phi : \varphi(x_\ell) = \varphi(y_\ell) \text{ for all } 1 \le \ell \le L\}\right|}{|\Phi|} \tag{114}$$

and let $\xi : \mathbb{R} \to \mathbb{R}$ be a strictly increasing function. Since the nearest neighbor classification rule works by comparing the values of $K^\star$ on various pairs of points, it is clear that using the kernel $K^{\star\star}(\mathbf{x},\mathbf{y}) = \xi(K^\star(\mathbf{x},\mathbf{y}))$ is equivalent to using the kernel $K^\star(\mathbf{x},\mathbf{y})$. In particular, choosing $\xi(x) := \log(1 + (n_w^L/\alpha)x)$ gives the following family of optimal kernels:

$$K_\alpha^{\star\star}(\mathbf{x},\mathbf{y}) = \log\left(1 + \frac{n_w^L}{\alpha}K^\star(\mathbf{x},\mathbf{y})\right) \tag{115}$$

To be clear, all these kernels are exactly equivalent to the the kernel $K^\star$ when using a nearest neighbor classification rule. However, they lead to different algorithms when used for kernelized SVM. We have experimented with various choice of the function $\xi$ and found out that this logarithmic scaling works well for kernelized SVM.

For each of the 10 possible parameter settings in Table 3 and Table 4, we do 10 experiments. For each experiment we generate:

- A training set containing $R \times n_{\text{spl}}$ sentences.
- A test set containing $1,000$ unfamiliar sentences (one sentences per category).

Let us denote by $\mathbf{x}_i^{\text{train}}$, $1 \le i \le R \times n_{\text{spl}}$, the data points in one of these training set, and by $\mathbf{x}_i^{\text{test}}$, $1 \le i \le 1000$, the data points in the corresponding test set. In order to run the kernelized SVM algorithm we need to form the Gram matrices

$$G_{ij}^{\text{train}} = K_\alpha^{\star\star}(\mathbf{x}_i^{\text{train}}, \mathbf{x}_j^{\text{train}}) \qquad \text{and} \qquad G_{ij}^{\text{test}} = K_\alpha^{\star\star}(\mathbf{x}_i^{\text{test}}, \mathbf{x}_j^{\text{train}}) \tag{116}$$

Constructing each of these Gram matrices takes a few days on CPU. We then use the SVC function of Scikit-learn to run a soft multiclass kernelized-SVM algorithm. We tried various values for the parameter controlling the $\ell_2$ and found that the algorithm is not sensitive to this choice — so we chose $C = 1$. The algorithm, on the other hand, is quite sensitive to the choice of the hyperparamater $\alpha$ defining the kernel $K_\alpha^{\star\star}$. We experimented with various choices of $\alpha$ and found that choosing the smallest $\alpha$ that makes the Gram matrix $G^{\text{train}}$ positive definite works well (note that the Gram matrix *should* be positive semidefinite for the kernelized SVM method to make sense). In Table 5 we show an example, on a specific pair of train and test set[5], of how the eigenvalues of $G^{\text{train}}$ and the test accuracy depends on $\alpha$.

---

[5] the training set and test set used in this experiment were generated by our data model with parameters $L = 9$, $n_w = 50$, $n_c = 5$, $R = 1000$, $n_{\text{spl}} = 8$, and $n^* = 3$

