# OpenReview forum: "Long-Tailed Learning Requires Feature Learning"
_ICLR.cc/2023/Conference — ICLR 2023 poster_

### Official Review · Reviewer_3imt · 2022-10-25

**Confidence:** 4
**Correctness:** 4
**Technical Novelty And Significance:** 3
**Empirical Novelty And Significance:** 3
**Recommendation:** 8

**Clarity, Quality, Novelty And Reproducibility:**

**Clarity** The paper is pretty clear although some aforementioned points can be improved.

**Quality** The quality of paper is above average but there are some problems which I brought up above.

**Novelty** The paper as far as I can tell is very novel but it resembles previous works in some of the long-tail assumptions.

**Reproducibility** The paper contains sufficient details for the experiments to be reproduced.


**Strength And Weaknesses:**

### Strength
1. The paper proposes a fairly novel model for deep learning in certain long-tailed distributions that argues for the importance of feature learning
2. The theoretical result is sound as far as I can tell and the proof techniques are novel for this area (although I did not have the time to go through all the details of the appendix given the short period of review).
3. The story is convincing, and I can tell that the authors have taken great care to make the paper as clear as possible in spite of its highly complicated nature and novelty (although there are still parts of the paper that are unclear)

### Weakness
1. It is unclear how well the proposed model reflects real world problems. Some empirical evidence would strengthen the paper.
2. There seems to be a disconnect between the theoretical part and empirical verification (please correct me if I am wrong).
     - Based on my understanding, the theoretical result precludes **any** feature map $\psi$ that maps the sentence to features, which presumably includes those learned by neural networks so it is weird that the experiments suggest something otherwise. In a similar vein, I find the claim to be too strong to be useful.
    - I spent a lot of time trying to understand the previous point and I came to the conclusion that (I believe) the tasks used in the experiments only have a single $\varphi$. Is this correct? If that is the case, how does the experimental section fits into the theoretical framework which uses $\Phi$ that include **all** partition functions.
    - How does the theory change if I have only **one** partition function $\varphi$?
    - I am concerned that this disconnect makes the theoretical results less useful than it appears to be.
3. The test error is not iid but in practice the model would presumably still get most of the data correct if the test distribution is the same as training distribution.
4. The paper doesn’t have a conclusion.

### Questions
1. Can you provide more explanation on how one should interpret the optimal feature map $\psi^\star$?

**Summary Of The Paper:**

The paper proposes a novel model of data, and shows that under the proposed model, a model can only succeed if it performs feature learning. Specifically, the primary data model proposed in the paper involves a set of concepts, each with an equal number of vocabularies. There is an $R$ number of categories (or classes). Each category contains two sequences of concepts. The first concept sequence, $c_r'$, is “unfamiliar” which accounts for a small portion of the generated data (i.e., long tail of the distribution) and the second concept sequence, $c_r$, is “familiar” which accounts for the majority of generated data (i.e., the head of the distribution). At training time, for each sample, the model first samples a concept sequence, and then generates a sequence of vocabularies corresponding to the concept sequence uniformly. At test time, the model only samples from the tail of the distribution. The paper then proceeds to show that on this data model, generalization error on the test distribution be lower bounded by a combinatorial quantity and using this lower bound, the paper shows in certain configurations of the data distribution, the test error can be arbitrarily high. Finally, the paper demonstrates, through simple experiments on an instantiation of the proposed data model, that the test error of models which do not learn features is much higher than neural networks which learn features.

**Summary Of The Review:**

The paper proposes a very interesting data model that advocates for the importance of learning features and shows models that do not learn features cannot do well on the tail of the distribution. However, currently, there seems to be a gap between the theoretical results and empirical results which I could not resolve after reading the paper multiple times. As such, my current assessment of the paper is slightly below the threshold and look forward to hearing the authors’ response.


## Update
The paper is much clearer after the revision and I think the paper is definitely interesting enough for the theoretical community. I decide to increased my score to 8.

---

> ### Author Response · Authors · 2022-11-14
> **Authors' response**
>
> Thanks for the time you spent carefully reviewing our work. We answer your questions and concerns below.
>
> > It is unclear how well the proposed model reflects real world problems.
>
> The model is somewhat synthetic, and necessarily so. Theoretical work in just about any field will frequently study idealizations of real problems. The relevant inquiry is whether the idealization captures something of importance, and whether it does so in a way that allows for practical lessons to be learned from it. In our specific case, it is already very difficult to even give a precise definition of what `learning' actually means without immediately running into problems. While vague definitions are easy to come by, it is impossible to draw mathematically meaningful distinctions from ill-defined notions. Moreover, overly complicated models tend to introduce confounds. That makes it difficult to ascertain whether it is the phenomenon of interest (feature learning) or one of the confounds that explains a performance gap. The data model we study strikes the appropriate balance in our view; it exhibits characteristics, such as a hierarchical, patch-level structure and a long-tailed distribution, that occur in applications yet it remains analytically tractable. In short, it is simple enough to study in depth but has a rich enough structure that many further papers can and will be written based on it.
>
> >  Based on my understanding, the theoretical result precludes any feature map  that maps the sentence to features, which presumably includes those learned by neural networks so it is weird that the experiments suggest something otherwise.
>
> This is a bit of a subtle point, but it all depends on the order of operations.
> Consider the following scenario:
> 1. Choose a feature map $\psi$.
> 2. Randomly select a task (that is, a partition $\phi$ and sequences of concepts $c_1$,...,$c_R$ and $c_1'$,..., $c_R'$).
> 3.  Generate a train and test set from the task.
> 4. Classify the test points using a nearest neighbor algorithm on the features extracted by $\psi$.
>
> The main theorem states that it is not possible to choose a feature map in step 1 that will be successful in step 4.
>
> Compare it with the alternative scenario:
> 1. Randomly select a task (that is, a partition $\phi$ and sequences of concepts $c_1$,...,$c_R$ and $c_1'$,..., $c_R'$).
> 2. Generate a train and test  set from the task.
> 3. Train a neural net on the train set and let $\psi$ be the feature map associated with this trained neural net.
> 4. Classify the test points using a nearest neighbor algorithm on the features extracted by $\psi$.
>
> Our experiments show that the feature map $\psi$ learned by the network in step 3 works almost perfectly in step 4.
>
> In the first scenario the feature map is chosen **before** the random selection of the task ($\psi$ is task independent). In the second scenario, the neural network learns the feature map from the training set, and since the training set is generated by the task, this process takes place **after** the random selection of the task ($\psi$ is task dependent). The map $\psi$ learned by the neural network works almost perfectly for the specific task randomly selected in step 1 of the second scenario, but by our main theorem, it would fail for most other tasks.
>
> The point here is to separate **hand crafted** or **fixed** features from **learned features** via the order of operations. Hand crafting a feature map that works across a broad range of problems is hard since the feature map must work for tasks which are a-priori unknown. Indeed, we mathematically prove that this cannot be done (this is a bit similar to the no-free-lunch theorem from learning theory). On the other hand, a neural network will have no difficulty since it will tailor its features to the task, and therefore perform perfectly on data drawn from that task, but it too must fail if the task changes but the features do not. The discussion on page 6, titled “Interpretation”, provides a more detailed explanation of our learning framework.
>
> > How does the theory change if I have only **one** partition function $\phi$?
>
> If there is a single partition function $\phi$ (or a single task to be more precise), then the best possible feature map succeeds 100% of the time. This is Theorem 2 in the paper. If the collection of tasks reduces to a single task then the problem becomes trivial; we know with 100% certainty the only problem we could ever face, so we solve it perfectly. Once again, see the discussion on page 6.

---

> > ### Author Response · Authors · 2022-11-14
> > **Continued response**
> >
> > > The test error is not iid but in practice the model would presumably still get most of the data correct if the test distribution is the same as training distribution.
> >
> > Consider the data model with $L = 9$, $n_w = 150$, $n_c = 5$, $R = 1000$, $n_{\rm spl} =10$ but take $n^* = 5$, which corresponds to a situation where each category contains 5 familiar sentences and 5 unfamiliar sentences. This is the "constant samples per class" realm; the distinction between familiar and unfamiliar sentences disappears and the train/test distributions are essentially the same. Even in this scenario the best possible feature map will fail at least 92.4% of the time (taking $n^* = 5$ in inequality (10) shows this). We will provide experiments and discussions corresponding to the above setting (i.e. train and test distribution are the same) to help clarify this important point. Thanks for raising it.
> >
> > > Can you provide more explanation on how one should interpret the optimal feature map $\psi^\star$?
> >
> > The optimal feature map $\psi^\star$ is better understood through its associated optimal kernel $K^\star({\bf x},{\bf y})$ that measures the similarity between a pair $({\bf x}, {\bf y})$ of sentences. To design the **best possible** kernel we need to leverage the only information we have. That is, we know the general structure of the problem (words are partitioned into concepts) but not the partition $\phi$ itself. So if we get two sentences ${\bf x}$ and ${\bf y}$ and try to determine if they were generated by the same sequence of concepts, intuitively the best we can do is the following: try all possible partitions of the vocabulary and count how many of them lead to sentences ${\bf x}$ and ${\bf y}$ having the same underlying sequence concept (that is: the first word of both sentences belong to the same concept, the second word of both sentences belong to the same concept, and so forth). If we obtain a high count, then it is likely that these two sentences have been generated by the same sequence of concepts.
> >
> > The kernel $K^\star$ does exactly this: as can be seen from formula (12), it counts the number of partitions of the vocabulary for which sentences $\bf x$ and ${\bf y}$ have the same underlying sequences of concepts.

---

> > > ### Comment · Reviewer_3imt · 2022-11-14
> > > **Thanks for the response.**
> > >
> > > I appreciate the authors for the response. My questions have been answered. This explanation is more or less aligned with what I had in mind but much more intuitive as I would say it was not clear in the text. I would encourage the authors to include something to this effect in the draft to emphasize it. I believe part of my confusion arises from the definition of the optimal kernel which is much clearer after the clarification, so I would also encourage the authors to include them in the text.
> > >
> > > I will increase my score after the authors update the draft with these changes as well as the new experiments.

---

> > > > ### Author Response · Authors · 2022-11-16
> > > > **Paper updated**
> > > >
> > > > * We have added a paragraph (page 8 in blue) that provides some intuition on how to interpret and understand the formula defining the optimal kernel $K^\star$.
> > > > * On page 9, in blue, we now explain intuitively why the empirical success of the neural network does not contradict our theoretical result.
> > > >
> > > > We would like to thank you for raising these two points --- the second one in particular is quite subtle and clarifying it has certainly improved the paper. Similarly, the optimal kernel $K^\star$ is at the heart of our proof, and an intuitive description of it was missing from the first version.

---

> > > > > ### Comment · Reviewer_3imt · 2022-11-18
> > > > > **Score updated.**
> > > > >
> > > > > Thank you for updating the paper. I have decided to increase my score to 8. Please finish all the promises you made to other reviewers.

---

> > > > > > ### Author Response · Authors · 2022-11-18
> > > > > > **Thanks**
> > > > > >
> > > > > >  Thank you for taking the time to provide a thoughtful and careful review. Your suggestions have improved the overall clarity of paper: the theoretical results are now more intuitive and easier to digest. We appreciate your contribution.

---

### Official Review · Reviewer_gVke · 2022-10-26

**Confidence:** 3
**Correctness:** 3
**Technical Novelty And Significance:** 3
**Empirical Novelty And Significance:** 1
**Recommendation:** 6

**Clarity, Quality, Novelty And Reproducibility:**

The paper is clearly-written and technically sound. The idea of using a data model that is simple enough that can be analyzed in a combinatorial way is simple. But I don't know many papers that are doing that. So in this sense, I find this work original and more interesting to read than more common works about a new algorithm improvement that brings another eta improvement on some benchmark. The experiments should be easy to reproduce from the description + the code is provided by the authors.


**Strength And Weaknesses:**

Strengths:
+ The paper is well-written, mostly easy to understand, and technically sound.
+ The theoretical findings are nicely confirmed by empirical experiments.
+ The very comprehensive appendix nicely explains the proofs and some other details (unfortunately, I haven't yet had time to go through everything).
+ The paper matches the main themes of the conference very well.

Weaknesses:
- The main conclusions of the work are not surprising, I believe this is the well-known fact that tail-distributions require better feature learning and was kind of confirmed empirically in many domains. While I find this as a nice theoretical confirmation, at the moment, I don't see any other application/implications of this work. The derived error bound is just for the proposed model, which is too simple to be used as a model for any real dataset.
- Based on the abstract I was hoping that the authors will use the model to try to answer some more interesting questions like: what error can we expect if we will use the same feature for a set of similar tasks?
- Among all the nice proofs and details in the appendix, I lack the proof for Theorem 2.
- Some important details are in the appendix. e.g., the experimental section is much clearer after reading the corresponding appendix section. I would suggest moving the description of parameters to the appendix and describing the better experimental protocol in the main paper.

NITs:
- In section 2, the example with 3 concepts could use different colors to mark words from the same concept, I believe this would make it even easier to read/understand quickly.
- Some small language mistakes I noticed:
  - from a different point view -> from a different point of view
  - intances -> instances
  - The former represent/the latter represent -> represents?
  - data generative process itself more coherent -> data generative process itself is more coherent
  - We would like to emphasis one more time -> We would like to emphasize one more time
  - one just need -> one just needs
  - Our experiments shows -> Our experiments show
  - There are missing comas in phrases like "In ...," in some places.

**Summary Of The Paper:**

The authors propose a simple data model of a classification task (a model that generates a training and testing set), and use it to demonstrate the theoretical importance of learning task-specific features (transformation of input features) to achieve good generalization performance on a specific task. The model relates to the commonly encountered long-tail distribution of data, by assuming a low number of examples in the train set for each category/class and test set containing unobserved before examples (not present in train set), what is common for a long-tail (the unbalanced that is also a characteristic for long-trail distributions is not touched here). In such a setting, intuitively learning features is required to achieve a good result. First, the authors use their model to confirm that by showing that on a given task, the nearest neighbor classifier achieves a perfect score when given optimal features representation results using experimental comparison. Later the authors show that it's not possible to find feature representation that will generalize well for all tasks from a set of all possible (with the assumed data model with the same parameters) when using the nearest neighbor classification. The authors derive the bond on the expected error on a task from a set of all possible ones. Finally, these findings are confirmed by empirical experiments, where the authors compare the performance of two settings. In the first, a neural network was used to learn features for a specific task, and mean performance on these tasks is reported, as expected, this gives low mean error. In the second setting, one general feature map is used for all tasks, as expected, resulting in a very high mean error.

**Summary Of The Review:**

I find this paper very difficult to rate, I think it is interesting, from the point of view of the technique and the proof, maybe because I'm not very familiar with similar works. On the other hand, I'm not sure how this work may impact further research. Convinced by the high quality of the writing, at the moment, I'm leaning toward accepting this paper.

---

> ### Author Response · Authors · 2022-11-14
> **Authors' response**
>
> Thanks for the time you spent carefully reviewing our work. We answer your questions and address your concerns below.
>
> > The main conclusions of the work are not surprising, I believe this is the well-known fact that tail-distributions require better feature learning and was kind of confirmed empirically in many domains. While I find this as a nice theoretical confirmation, at the moment, I don't see any other application/implications of this work. The derived error bound is just for the proposed model, which is too simple to be used as a model for any real dataset.
>
> As our general reply mentions, this is certainly a common intuition that has been around for a long time. Yet it is a wholly separate matter, of course, to try and give a vague intuition precise mathematical content. The past decade of research has witnessed several "empirically confirmed" intuitions in deep learning that later fell apart under further scrutiny. Additionally, some skeptics question whether neural networks actually *learn* anything at all. This makes it quite valuable to have a well-defined setting where the precise boundaries of an important intuitive concept can be explored with precision.
>
> A number of practical inferences flow from our results. For example, they indicate that "bigger architectures are better" does not always hold in the context of NLP tasks (It is well-known from the NTK literature that as the size of an architecture grows its predictions reduce to those of a fixed kernel method, which provably fails by our main theorem). We may also infer the importance of using a non-convex, non-linear optimization procedure even for a linearly separable problem (It is necessary to learn features, and this cannot happen with a convex model). In our view this makes the paper more practically relevant than, say, universal approximation theorems that hold regardless of the data under consideration and so cannot draw meaningful distinctions.
>
> > Some important details are in the appendix. e.g., the experimental section is much clearer after reading the corresponding appendix section. I would suggest moving the description of parameters to the appendix and describing the better experimental protocol in the main paper.
>
> We are currently working on adding to the experimental section. In addition to making this suggested change, the revision will include additional experiments requested by other reviewers.
>
> > Among all the nice proofs and details in the appendix, I lack the proof for Theorem 2.
>
> The proof of this theorem is on page 5, right before the statement of the theorem. Equation (8) is the feature map for which the generalization error vanishes.
>
> Finally, thanks for the NITs: we will address them.

---

> > ### Author Response · Authors · 2022-11-18
> > **Color coding the figures**
> >
> > The figures are now color coded (words belonging to the same concept have the same color). It indeed makes them easier to read --- thanks for the suggestion.

---

> > ### Author Response · Authors · 2022-11-18
> > **Empirical section**
> >
> > We have followed your suggestion: The description of the parameters was moved to the appendix, and the experimental protocol is now described more carefully in the main body of the paper. We hope the empirical section is now clearer. Thank you for this feedback.

---

### Official Review · Reviewer_LVLa · 2022-10-28

**Confidence:** 3
**Correctness:** 3
**Technical Novelty And Significance:** 3
**Empirical Novelty And Significance:** 3
**Recommendation:** 5

**Clarity, Quality, Novelty And Reproducibility:**

The clarity and quality of the text are ordinary. Specifically, the logic of the proof and reasoning process in the text is not clear, and the necessary reasoning process is lacking in the main text
However, the novelty and originality are good, the idea given is inspiring and novel.


**Strength And Weaknesses:**

Strength:
1. The effectiveness of the method has been proved by mathematical analysis and empirical experience.
2. The method can be used for long-tailed problems in the field of both CV and NLP.
3. The paper puts novel theoretical results that can inspire future theoretical research on long-tail problems.

Weaknesses:
1. The experimental results are too few to illustrate the effectiveness of the method.
2. Although the detailed proof process is given in the appendix, the idea of the text is a bit jumpy.
3. The method in this paper is based on the nearest neighbor or SVM, which requires neural networks to extract features for learning. However, the extraction process is a black-box model, and it is difficult to combine the features extracted by the deep network with the method in this paper without mathematics analysis and experimental verification in this regard.


**Summary Of The Paper:**

This paper derives generalization error bounds that are non-asymptotic and relatively tight against long-tailed problems, within the context of our data model, through mathematical analysis and partial empirical evidence.

**Summary Of The Review:**

Being a novel research in the field of CV and NLP, especially some parts do not have strict mathematical analysis, but rely on experience, and more experimental results should be provided to support the conclusion. In addition, the legends in the article are not rich, which is not conducive to reading and understanding. More graphical explanations should be added appropriately. Finally, as a very mathematical article, important reasoning procedures should be included in the main text rather than all listed in the appendix.

---

> ### Author Response · Authors · 2022-11-14
> **Author's response**
>
> Thanks for your review. We now address the following concern:
>
> > The method in this paper is based on the nearest neighbor or SVM, which requires neural networks to extract features for learning. However, the extraction process is a black-box model, and it is difficult to combine the features extracted by the deep network with the method in this paper without mathematics analysis and experimental verification in this regard.
>
> There seems to be some confusion on this point. To clarify, the purpose of the work is not to introduce new methods or new algorithms. Rather, we provide theoretical results that quantify the penalty that one must pay for not learning features. The experiments are not meant to validate the success or failure of a proposed algorithm. They serve an illustrative role instead, and were designed to confirm that success or failure depends only on the ability to identify the correct features and not on the underlying classification rule. It is better to think of them as the picture that illustrates thousands of theoretical words. A quick glance at Table 1 shows a dramatic performance gap between those algorithms that learn features compared to those that do not; that’s our theoretical result in a nutshell. Our goal was to provide easily reproducible experiments that clearly illustrate our theoretical results.

---

> > ### Author Response · Authors · 2022-11-18
> > **About reasonings being in the appendix rather than in the main text**
> >
> > > Important reasoning procedures should be included in the main text rather than all listed in the appendix.
> >
> > > The necessary reasoning process is lacking in the main text.
> >
> > The 9 pages format of ICLR required us to make some choices. Our priority, in the main text, was to:
> > 1. clearly and precisely state our theoretical results;
> > 2. explain why we think these theoretical results are important;
> > 3. provide some ideas of the kind of techniques we use to prove the theoretical results, and describe intuitively the main mathematical objects involved in the proof (i.e., optimal feature map and permuted moment);
> > 4. provide some empirical confirmations of our theoretical results, and explore empirically questions that we couldn't resolve analytically.
> >
> > While we would have liked to include more of the actual proof in the main text, making as clearly as possible the four points above was more important to us.
> >
> > Nonetheless, we could have done a better job with point 3). In the new version of the paper, we have added intuitive explanations for one of the main mathematical objects considered in our work, namely the optimal kernel $K^\star$. The formula defining it is now explained at an intuitive level (see paragraph in blue on page 8).
> >
> > We have also updated the empirical section with the goal of better highlighting our theoretical results (for example one row in the table is now devoted to our theoretical bound). We hope this revised empirical section helps in providing intuition and context for our theoretical results.
> >
> > > The experimental results are too few to illustrate the effectiveness of the method.
> >
> > The new version of the paper contains many additional experiments that better show how the theoretical lower bound (10) on the generalization error compares with the empirically evaluated generalization error.

---

### Official Review · Reviewer_VYVD · 2022-11-01

**Confidence:** 4
**Correctness:** 4
**Technical Novelty And Significance:** 3
**Empirical Novelty And Significance:** 3
**Recommendation:** 5

**Clarity, Quality, Novelty And Reproducibility:**

The motivation and data model of this paper are very clear, while the formal claims and derivations could be improved.

To the best of my knowledge, the data model and results are novel, and the results seem reproduceable.

**Details Of Ethics Concerns:**

No ethics concerns

**Strength And Weaknesses:**

Strengths

 - The model is relatively straightforward to understand, and it is relatively intuitive to understand why it is critical to learn the features.
 - The performance differences between learning and not learning the features are  very dramatic.

Weaknesses

 - The theoretical section of the paper is somewhat difficult to follow. There are many variables that I had to look back to recall, and the derivations sometimes lack descriptive text around it to help guide the reader.
 - The model is somewhat synthetic. While it does highlight the importance of learning categories in this particular data model, it is relatively unclear to me how directly this generalizes to other NLP problems and image problems.


**Summary Of The Paper:**

This paper studies machine learning in settings where where classes have rare subcategories (potentially as small as one example in the training set).

The paper creates a data model, and shows that while a model that properly learns features can achieve very high performance, models that do not learn these features have very low accuracy. Error bounds are derived for this data model, and experiments showing similar behavior with other models are performed.

**Summary Of The Review:**

The paper introduces a data model where the performance of machine learning heavily depends on learning the categories of each word. While the results do highlight the importance of feature learning in this data model, my impression is that the data model is still relatively far from real problems.

---

> ### Author Response · Authors · 2022-11-14
> **Authors' response**
>
> Thanks for your review. We now address the following concern:
>
> > The model is somewhat synthetic. While it does highlight the importance of learning categories in this particular data model, it is relatively unclear to me how directly this generalizes to other NLP problems and image problems.
>
> The model is somewhat synthetic, and necessarily so. Theoretical work in just about any field will frequently study idealizations of real problems. The relevant inquiry is whether the idealization captures something of importance, and whether it does so in a way that allows for practical lessons to be learned from it. In our specific case, it is already very difficult to even give a precise definition of what *learning* actually means without immediately running into problems. While vague definitions are easy to come by, it is impossible to draw mathematically meaningful distinctions from ill-defined notions. Moreover, overly complicated models tend to introduce confounds. That makes it difficult to ascertain whether it is the phenomenon of interest (feature learning) or one of the confounds that explains a performance gap. The data model we study strikes the appropriate balance in our view; it exhibits characteristics, such as a hierarchical, patch-level structure and a long-tailed distribution, that occur in applications yet it remains analytically tractable. In short, it is simple enough to study in depth but has a rich enough structure that many further papers can and will be written based on it.

---

> > ### Author Response · Authors · 2022-11-18
> > **Theoretical section**
> >
> > > The theoretical section of the paper is somewhat difficult to follow.
> >
> > We have added some intuitive explanations in the theoretical section (Section 5): The formula for the optimal kernel $K^\star$ is now explained at an intuitive level (see paragraph in blue on page 8). We hope the section is a bit easier to follow.

---

### Author Response · Authors · 2022-11-14
**To all Reviewers**

We gladly thank the referees for their feedback and for the time they spent carefully reviewing our work. As all reviewers echo a concern that applies to any theoretical work, i.e. the extent to which it bears on empirical practice, we thought it would be helpful if we explained our motivations for the paper and situated it in the broader context of theoretical deep learning.

The intuitive idea that learning features leads to better generalization has permeated the deep learning folklore since its inception, but the extent to which this intuition plays out in practice remains debatable. For example, a common critique of deep learning posits that neural networks do not actually learn anything at all and function instead via memorization. Unfortunately, most (although certainly not all) theoretical work does not bear directly on this question for one of two reasons. First, more traditional "black-box" learning guarantees only apply in an asymptotic, large-sample limit. The "constant number of samples per class" realm or the long-tailed realm of empirical practice simply does not fit into these black boxes. Second, theoretical works often invoke overly synthetic data models that fail to capture the types of structures we expect networks to learn in applications. This makes the mechanisms behind feature learning and generalization poorly understood from a theoretical perspective, so it remains quite difficult to know with any mathematical precision the extent to which intuition is correct and the critique is wrong, or vice-versa.

The paper arose out of our own frustration with this state of affairs. To put it succinctly, it is very difficult to give a precise, non-asymptotic notion of what *learning* actually means in a way that leads to mathematically meaningful distinctions. It is the nature of the scientific endeavor, of course, that theoretical work frequently deals in idealizations. In our mind, a good idealization should elucidate the phenomenon of study as clearly as possible. The model should prove rich enough to have a clear concept of *feature-learning* as a fundamental mechanism; it should bear some resemblance to the types of data (e.g. long-tailed) that occur in applications; and it should have no extraneous but irrelevant properties that only muddy the waters. Finally, it must meet these goals while remaining analytically tractable. We believe we achieved this goal.

---

### Author Response · Authors · 2022-11-16
**New experiments added to the empirical section**

To address some of the reviewers' questions, and also to clarify our overall message, we have made the following minor revisions to the empirical section (Section 6 in the paper):
* We describe the experimental protocol more carefully.
* We added experiments with multiple unfamiliar sentences per category. In particular, these experiments reveal how the theoretical lower bound (10) on the generalization error compares with the empirically evaluated generalization error for various values of $n^*$.
* We added experiments with the trivial feature map $\psi_{\rm one-hot}$ (c.f. the updated experimental section for a formula for this feature map). The reader can now see the empirically evaluated success rate of the nearest neighbor classification rule applied on the top of two distinct feature maps ($\psi^\star$ and $\psi_{\rm one-hot}$). Our theoretical bound on the success rate applies to *any* feature map.

We also added a set of experiments with different model parameters to the appendix (see table 4 p 37).

The paragraphs in blue answer specific questions raised by a reviewer.

---

### Decision · Program_Chairs · 2023-01-20

**Decision:**

Accept: poster

**Justification For Why Not Higher Score:**

Reviewers had some reservations of the impact on wider ML community, which may make this less appropriate for a spotlight/oral.

**Justification For Why Not Lower Score:**

Solid theoretical contribution about a known empirically observed phenomenon. In the discussions, all reviewers were willing to accept (although some of them may not have updated their scores).

**Metareview: Summary, Strengths And Weaknesses:**

The paper studies the importance of learning features to achieve good generalization in the setting where the data distribution has a long tail.

The reviewers agreed the paper provides a detailed theoretical study, under a simple long-tailed data model, to study the importance of learning features to achieve good generalization. Such an intuition seems to have been known empirically, and is highly practically relevant, and this appears to be the first work to give a framework in which to formalize this intuition.

If accepted, we encourage the authors take the reviewers' feedback into consideration, especially with regards to clarity and further discussion around the relevance and impact of these theoretical results.


**Note From Pc:**

if the above contains the word "oral" or "spotlight" please see: "oral" presentation means -> notable-top-5% and "spotlight" means -> notable-top-25%. As stated in our emails, we are disassociating presentation type from AC recommendations